# High-dimensional limit theorems for SGD: Momentum and Adaptive Step-sizes

**Aukosh Jagannath**
Department of Statistics and Actuarial Science, Department of Applied Mathematics
Cheriton School of Computer Science
University of Waterloo, Canada
`a.jagannath@uwaterloo.ca`

**Taj Jones-McCormick, Varnan Sarangian**
Department of Statistics and Actuarial Science
University of Waterloo, Canada
`{tejonesm, v3sarang}@uwaterloo.ca`

## Abstract

We develop a high-dimensional scaling limit for Stochastic Gradient Descent with Polyak Momentum (SGD-M) and adaptive step-sizes. This provides a framework to rigourously compare online SGD with some of its popular variants. We show that the scaling limits of SGD-M coincide with those of online SGD after an appropriate time rescaling and a specific choice of step-size. However, if the step-size is kept the same between the two algorithms, SGD-M will amplify high-dimensional effects, potentially degrading performance relative to online SGD. We demonstrate our framework on two popular learning problems: Spiked Tensor PCA and Single Index Models. In both cases, we also examine online SGD with an adaptive step-size based on normalized gradients. In the high-dimensional regime, this algorithm yields multiple benefits: its dynamics admit fixed points closer to the population minimum and widens the range of admissible step-sizes for which the iterates converge to such solutions. These examples provide a rigorous account, aligning with empirical motivation, of how early preconditioners can stabilize and improve dynamics in settings where online SGD fails.

## 1 Introduction

Stochastic gradient descent (SGD) and its variants are central to large-scale optimization in modern machine learning. A significant theoretical interest lies in characterizing and comparing the performance of these algorithms in the high-dimensional setting, where data and compute are limited relative to the data dimension and model complexity.

The fixed-dimensional asymptotic theory of SGD is classical, dating back to Robbins & Monro (1951); McLeish (1976); Kushner (1984). In this regime, the small step-size limit of online SGD converges to the gradient flow on the population loss. More recently, in the high dimensional regime, there have been numerous works on scaling limits where the dimension tends to infinity for specific problems, including linear regression (Wang et al., 2017; Paquette et al., 2021; 2024), online PCA (Li et al., 2016; Wang et al., 2017), single-index models (Saad & Solla, 1995a;b; Goldt et al., 2019; Veiga et al., 2022), and multi-index models (Collins-Woodfin et al., 2024). A unifying framework for proving such scaling limits was introduced by Ben Arous et al. (2022; 2024), including an extension to study diffusive dynamics. A key insight from the high-dimensional perspective is that when the dimension scales up as the step-size tends to zero, there exists a critical scaling regime of the step-size in which high-dimensional effects yield distinct dynamics. The fixed-dimensional dynamics are recovered with sub-critical step-size, but at the critical scaling, there are additional corrections, such as the "population corrector," in addition to the gradient flow drift.

For SGD with momentum (Polyak, 1964; Nesterov, 1983; Sutskever et al., 2013), recent works in the fixed dimensional setting consider similar continuous-time ballistic limits (Su et al., 2016; Kovachki

& Stuart, 2021; Wilson et al., 2021; Feng et al., 2023). There is also a well-established body of literature characterizing the benefits of SGD-M over online SGD, such as improved convergence rates, reduced escape times from saddle points, and implicit regularization effects (Liu et al., 2021; Wang et al., 2021; Cowsik et al., 2022; Gess & Kassing, 2023; Ghosh et al., 2023; Dang et al., 2025). Conversely, in the high-dimensional setting, Paquette & Paquette (2021); Ferbach et al. (2025) study the limiting dynamics of several momentum-based stochastic algorithms in the setting where they scale the momentum parameter accordingly.

Preconditioned SGD methods incorporating adaptive step-sizes (Duchi et al., 2011; Hinton, 2012; Zeiler, 2012; Kingma & Ba, 2014; Yu et al., 2017; Shazeer & Stern, 2018) are also of theoretical interest, particularly for explaining their empirical success in high-dimensional, non-convex optimization (Zhang et al., 2020b). In fixed-dimensional settings, notable works addressed similar questions regarding convergence behaviour and continuous-time dynamics (da Silva & Gazeau, 2018; Reddi et al., 2019; Barakat & Bianchi, 2021; Ma et al., 2022; Malladi et al., 2022; Ghosh et al., 2023). The preconditioner we consider in Section 3, gradient normalization, was first introduced to mitigate exploding gradients in deep sequence models (Mikolov, 2012; Pascanu et al., 2013). Since then, there have been several works looking at the effectiveness of "clipped" gradient control (Li et al., 2017; Chen et al., 2020; Zhang et al., 2020a; Koloskova et al., 2023) as well as high-dimensional scaling limits for random least squares regression (Marshall et al., 2024).

## 1.1 CONTRIBUTIONS

We extend the effective dynamics framework of Ben Arous et al. (2024) to derive the high-dimensional limiting dynamics of SGD with Polyak momentum (SGD-M) and of SGD with adaptive step-sizes. We find that the critical step-size scalings are the same between these dynamics. Furthermore, we find that in the critical step-size regime, SGD-M amplifies the high-dimensional effects as compared to SGD, potentially steering the dynamics even further from the population gradient. More precisely, for any sequence of step-sizes in the critical scaling regime, the effective dynamics of SGD-M accentuates the aforementioned emergent "population corrector" observed in online SGD. However, for any given step-size of SGD-M, there exists a corresponding step-size for online SGD that yields the same effective dynamics, up to a time rescaling, establishing an equivalence between the algorithms.

In contrast, we find that training can be improved by incorporating pre-conditioning. To this end, we make use of our framework to compare SGD-M with a version of online SGD equipped with a simple adaptive step-size that restricts the gradients to have unit norm (Mikolov, 2012; Pascanu et al., 2013); we refer to this variant as "SGD-U" in subsequent sections. We prove, in two popular classes of learning tasks, that the SGD-U can exhibit dynamics with not only superior fixed points (i.e. closer to the population minimum) but can significantly broaden the range of admissible step-sizes that ensure convergence to these solutions. The admissibility of larger step-sizes that maintain stable dynamics and yield high-quality solutions provides a rigorous demonstration of how pre-conditioners can mitigate high-dimensional effects and empirical phenomena such as exploding or vanishing gradients. Thus, our framework offers rigorous justification for the empirical motivations underlying the design of early preconditioners.

While our framework is broadly applicable, we illustrate our results on two problems in high dimensional inference: spiked tensor PCA (Johnstone, 2001; Péché, 2006; Montanari & Richard, 2014) and single index models (Bishop & Nasrabadi, 2006; Hastie et al., 2009; Barbier et al., 2019).

## 2 MAIN RESULT

In this section, we present our main results, starting with the scaling limits of SGD-M, followed by a discussion of how the same methodology can be used to study effective dynamics for scalar preconditioners. All proofs are deferred to the appendix.

We consider the following online learning setup. Suppose we are given an i.i.d. data sequence $\boldsymbol{y}_1, \boldsymbol{y}_2, \ldots$ taking values in $\mathcal{Y} \subseteq \mathbb{R}^{d_n}$ with law $P_n \in \mathcal{M}_1(\mathbb{R}^{d_n})$ and loss $L : \mathcal{X}_n \times \mathcal{Y}_n \to \mathbb{R}$ where $\mathcal{X}_n \subseteq \mathbb{R}^{p_n}$ is the parameter space. Consider online SGD with learning rate $\delta_n > 0$ and momentum

rate $\beta \in [0, 1)$, given by

$$\begin{aligned}
\boldsymbol{p}_\ell &= \beta \boldsymbol{p}_{\ell-1} - \delta_n \, \nabla L_n(\boldsymbol{x}_{\ell-1}, \boldsymbol{y}_\ell) \\
\boldsymbol{x}_\ell &= \boldsymbol{x}_{\ell-1} + \boldsymbol{p}_\ell
\end{aligned} \tag{1}$$

with possibly random initialization $\boldsymbol{x}_0 \sim \mu_n \in \mathcal{M}_1(\mathcal{X}_n)$. Our interest is understanding the evolution of a finite collection of summary statistics $\boldsymbol{u}_n(\boldsymbol{x}) = (u_1^n(\boldsymbol{x}), \dots, u_k^n(\boldsymbol{x}))$ in the regime where both $p_n$ and $d_n$ may grow with $n$, and $\delta_n \to 0$ as $n \to \infty$.

We define the functions

$$\nabla H(x, Y) = \nabla L_n(x, Y) - \nabla \Phi(x) \quad \text{where} \quad \nabla \Phi(x) = \mathbb{E}[\nabla L_n(x, Y)]$$

In the following, we suppress the dependence of $H$ on $Y$ and let $V(x) = \mathbb{E}[\nabla H(x) \otimes \nabla H(x)]$ be the covariance matrix for $\nabla H$ at $x$.

To proceed, we state two assumptions: $\delta_n$-localizability and asymptotic closability. The first, which we present next, provides an upper bound on the learning rate in terms of the regularity of the summary statistics and data distribution. These bounds ensure tightness of trajectories of the summary statistics.

**Definition 2.1.** A tuple $(\boldsymbol{u}_n, \nabla L_n, P_n, \beta)$ is $\delta_n$-**localizable** with localizing sequence $(E_K)_K$ if there is an exhaustion by compacts $(E_K)_K$ of $\mathbb{R}^k$, and constants $C_K$ such that for any $i \le k$,

1. $\sup_{x \in \boldsymbol{u}_n^{-1}(E_K)} \|\nabla^2 u_i^n\|_{\text{op}} \le C_K \cdot \delta_n^{-1/2}$, and $\sup_{x \in \boldsymbol{u}_n^{-1}(E_K)} \|\nabla^3 u_i^n\|_{\text{op}} \le C_K$;

2. $\sup_{x \in \boldsymbol{u}_n^{-1}(E_K)} \|\nabla \Phi\| \le C_K$, and $\sup_{x \in \boldsymbol{u}_n^{-1}(E_K)} \mathbb{E}[\|\nabla H\|^8] \le C_K \delta_n^{-4}$;

3. $\sup_{x \in \boldsymbol{u}_n^{-1}(E_K)} \mathbb{E}[\langle \nabla H, \nabla u_i^n \rangle^4] \le C_K \delta_n^{-2}$, and
   $\sup_{x \in \boldsymbol{u}_n^{-1}(E_K)} \mathbb{E}[\langle \nabla^2 u_i^n(x), \nabla H(x, y) \otimes \nabla H(x, y) - V(x) \rangle^2] = o(\delta_n^{-3})$, and
   $\sup_{(x_i)_1^2 \in \boldsymbol{u}_n^{-1}(E_K)} \mathbb{E}_{y_1 \perp y_2}[\langle \nabla^2 u_i^n(x_1), \nabla H(x_1, y_1) \otimes \nabla H(x_2, y_2) \rangle^2] = o(\delta_n^{-3})$ for $\beta > 0$

The second assumption ensures that the limiting coefficients for the evolution of the statistics close. To this end, we define the first and second-order differential operators,

$$\mathcal{A}_n = \langle \nabla \Phi, \nabla \rangle \quad \text{and} \quad \mathcal{L}_n = \frac{1}{2} \langle V, \nabla^2 \rangle$$

**Definition 2.2.** A family of summary statistics $(\boldsymbol{u}_n)$ are **asymptotically closable** for learning and momentum rates $((\delta_n)_n, \beta)$ if $(\boldsymbol{u}_n, \nabla L_n, P_n, \beta)$ are $\delta_n$-localizable with localizing sequence $(E_K)_K$, and furthermore there exist locally Lipschitz functions $\boldsymbol{h} : \mathbb{R}^k \times \mathbb{R} \to \mathbb{R}^k$ and $\boldsymbol{\Sigma} : \mathbb{R}^k \to \mathbb{R}^{k \times k}$, such that

$$\sup_{x \in \boldsymbol{u}_n^{-1}(E_K)} \left\| \left( -\frac{1}{1-\beta} \mathcal{A}_n + \frac{1}{(1-\beta)^2} \delta_n \mathcal{L}_n \right) \boldsymbol{u}_n(x) - \boldsymbol{h}\left(\beta, \boldsymbol{u}_n(x)\right) \right\| \to 0$$

$$\sup_{x \in \boldsymbol{u}_n^{-1}(E_K)} \|\delta_n J_n V J_n^T - \boldsymbol{\Sigma}(\boldsymbol{u}_n(x))\| \to 0$$

with $J_n$ the Jacobian of $\boldsymbol{u}_n$. We call $\boldsymbol{h}$ the *effective drift* and $\boldsymbol{\Sigma}$ the *effective volatility* of $\boldsymbol{u}$ respectively.

We refer the reader to Ben Arous et al. (2024) for a detailed discussion regarding the interpretations of Definition 2.1. In particular, these scaling assumptions are more general than the regularity assumptions commonly imposed in the literature, e.g., $L$-smoothness and convexity of the loss. These assumptions can be checked in a broad class of models, such as those studied here or in (Ben Arous et al., 2024) and related works. For example, one can verify that Definition 2.1 holds for common models such as binary classification and Gaussian XOR classification with two-layer neural networks with ReLU activations (Ben Arous et al., 2024).

Let us briefly compare the notions of localizability and asymptotically closability to the related notions in Ben Arous et al. (2024). Compared to online SGD, we now require an additional item in Definition 2.1-(3) to control the correlation between the random fluctuations $\nabla H(x_1, Y_i), \nabla H(x_2, Y_j)$ when $i \ne j$. Additionally, for Definition 2.2, we note that the two operators scale differently in $\beta$. We revisit this point in Remark 2.4 after presenting our main result below.

**Theorem 2.3.** Let $(X_\ell^{\delta_n})_\ell$ be SGD initialized from $X_0 \sim \mu_n$ for $\mu_n \in \mathcal{M}_1(\mathbb{R}^{p_n})$ with learning rate $\delta_n$ and fixed momentum parameter $\beta \in [0, 1)$ for the loss $L_n(\cdot, \cdot)$ and data distribution $P_n$. For a family of summary statistics $\boldsymbol{u}_n = (u_i^n)_{i=1}^k$, let $(\boldsymbol{u}_n(t))_t$ be the linear interpolation of $(\boldsymbol{u}_n(X_{\lfloor t\delta_n^{-1} \rfloor}^{\delta_n}))_t$.

Suppose that $\boldsymbol{u}_n$ are asymptotically closable with learning rate $\delta_n$, effective drift $\boldsymbol{h}$, and effective volatility $\boldsymbol{\Sigma}$, and that the pushforward of the initial data has $(\boldsymbol{u}_n)_* \mu_n \to \nu$ weakly for some $\nu \in \mathcal{M}_1(\mathbb{R}^k)$. Then $(\boldsymbol{u}_n(t))_t \to (\boldsymbol{u}_t)_t$ weakly as $n \to \infty$, where $\boldsymbol{u}_t$ solves:

$$d\boldsymbol{u}_t = \boldsymbol{h}(\beta, \boldsymbol{u}_t)\,dt + \frac{1}{1-\beta}\sqrt{\boldsymbol{\Sigma}(\boldsymbol{u}_t)}\,d\mathbf{B}_t$$

initialized from $\nu$, where $\mathbf{B}_t$ is a standard Brownian motion in $\mathbb{R}^k$.

Taking $\beta = 0$ recovers the original result due to Ben Arous et al. (2024) for online SGD. The proof is presented in Appendix B.

**Remark 2.4.** It is beneficial to compare SGD-M with online SGD as well as compare the fixed and high dimensional regimes in light of Theorem 2.3. To do so, suppose that we further impose the individual limits

$$\sup_{x \in \boldsymbol{u}_n^{-1}(E_K)} \|\mathcal{A}_n \boldsymbol{u}_n(x) - \boldsymbol{f}(\boldsymbol{u}_n(x))\| \to 0, \qquad \sup_{x \in \boldsymbol{u}_n^{-1}(E_K)} \|\delta_n \mathcal{L}_n \boldsymbol{u}_n(x) - \boldsymbol{g}(\boldsymbol{u}_n(x))\| \to 0$$

for some functions $\boldsymbol{f}, \boldsymbol{g} : \mathbb{R}^k \to \mathbb{R}^k$ such that the revised dynamics are spelt out as

$$d\boldsymbol{u}_t = \left[ -\frac{1}{1-\beta}\boldsymbol{f}(\boldsymbol{u}_t) + \frac{1}{(1-\beta)^2}\boldsymbol{g}(\boldsymbol{u}_t) \right] dt + \frac{1}{1-\beta}\sqrt{\boldsymbol{\Sigma}(\boldsymbol{u}_t)}\,d\mathbf{B}_t \qquad (2)$$

One can loosely interpret the *population drift* $\boldsymbol{f}$ as the "learning" or "signal" term which pushes the dynamics along the descent direction of the population loss $\Phi$. The *population corrector* $\boldsymbol{g}$ is then an emergent "variance" term in the critical scaling regime.

To compare the behaviour of SGD-M with online SGD, consider the dynamics of the latter with step-size $\hat{\delta}_n = \delta_n/(1-\beta)$. The limiting dynamics are then:

$$d\boldsymbol{u}_t = \left[ -\boldsymbol{f}(\boldsymbol{u}_t) + \frac{1}{1-\beta}\boldsymbol{g}(\boldsymbol{u}_t) \right] dt + \sqrt{\frac{1}{1-\beta}\boldsymbol{\Sigma}(\boldsymbol{u}_t)}\,d\mathbf{B}_t$$

which coincide with equation 2 after an additional time-rescaling $t \mapsto t/(1-\beta)$. In particular, the relative reweighting between $\boldsymbol{f}$ and $\boldsymbol{g}$ are modified so that the corrector poses a stronger influence on the dynamics, potentially overwhelming the underlying signal. Evidently, for the case of SGD-M, we see that $\boldsymbol{g}$ becomes more prominent as $\beta$ tends to one unless we adjust the step-size accordingly.

As a consequence of Remark 2.4, any trajectory produced by SGD-M over some time interval $[0, T]$ can be replicated by online SGD after a time change where both the step-size and time horizon are rescaled by $(1-\beta)^{-1}$. The effective number of iterations $T/\delta$ remains unchanged, confirming a similar observation to both Kovachki & Stuart (2021), who considers the fixed-dimensional deterministic regime, and Paquette & Paquette (2021), who considers the high-dimensional regime for the case of linear regression.

Finally, we note that our results subsume the classical ODE theory for fixed-dimensional SGD. That is, the high-dimensional dynamics reduce to the ballistic phase

$$d\boldsymbol{u}_t = -\frac{1}{1-\beta}\boldsymbol{f}(\boldsymbol{u}_t)\,dt$$

by taking $\delta_n$ below the critical scaling threshold. In this subcritical regime, the relative scaling of the population drift and population corrector becomes moot as $\boldsymbol{g}$ is negligible.

## 2.1 SCALING LIMITS OF ADAPTIVE STEP-SIZES

Following the setup of Theorem 2.3, consider the modified online SGD update step with a scalar-valued preconditioner $\eta_n : \mathcal{X}_n \times \mathcal{Y}_n \to \mathbb{R}^+$, given by

$$\boldsymbol{x}_\ell = \boldsymbol{x}_{\ell-1} - \delta \cdot \eta(\boldsymbol{x}_{\ell-1}, \boldsymbol{y}_\ell) \nabla L(\boldsymbol{x}_{\ell-1}, \boldsymbol{y}_\ell) \qquad (3)$$

Under the same approach of online SGD, we can develop a scaling limit by decoupling $\delta$ from the data-dependent component of $\eta$, i.e. we define

$$\nabla \tilde{H}(x, Y) = \eta_n(x, Y) \nabla L_n(x, Y) - \nabla \tilde{\Phi}(x) \quad \text{where} \quad \nabla \tilde{\Phi}(x) = \mathbb{E}[\eta_n(x, Y) \cdot \nabla L_n(x, Y)]$$

which serve as the analogue to $\nabla \Phi$ and $\nabla H$ as defined before. Provided $\nabla \tilde{H}$ and $\nabla \tilde{\Phi}$ satisfy Definition 2.1 and 2.2, then Theorem 2.3 with $\beta = 0$ extends naturally to the preconditioned case (see Appendix A for an extended discussion). Hence, we can rigorously derive scaling limits for this family of pre-conditioners under this same framework, which we demonstrate next.

## 3 EXAMPLES

In the remaining subsections, we demonstrate Theorem 2.3 and the followup discussion in Section 2.1 for the spiked tensor model and single index model. In particular, we consider the dynamics of SGD-M as well as the dynamics for a version of online SGD with,

$$\eta(x, Y) = \frac{\sqrt{n}}{\|\nabla L(x, Y)\|} \tag{4}$$

We impose the additional dimension factor $\sqrt{n}$ (see notation below) when normalizing the gradient to preserve the scaling relationships presented in Definition 2.1. Since $\|\nabla L\| = O(\sqrt{n})$ in both of our examples, we require the rescaling to ensure that the limiting dynamics are non-trivial. We abbreviate this version of SGD equipped with $\eta$ as "SGD-U" in subsequent sections.

### 3.1 SPIKED MATRIX AND TENSOR PCA

Suppose we are given i.i.d. samples of data of the form $Y^\ell = \lambda v^{\otimes k} + W^\ell$ where $W^\ell$ are i.i.d. $k$-tensors with Gaussian entries, $v \in \mathbb{R}^n$ is a unit vector, and $\lambda = \lambda_n > 0$ is the signal-to-noise ratio. Our goal is to infer the direction $v$ from the noisy sample.

We take as loss the (negative) log-likelihood,

$$L(x, Y) = \|Y - x^{\otimes k}\|^2$$

to optimize parameter vector $x$. We consider the pair of summary statistics $\boldsymbol{u}_n = (m, r^2)$ with

$$m = m(x) := \langle x, v \rangle \quad \text{and} \quad r^2 = r^2(x) := \|x - mv\|^2 = \|x\|^2 - m^2$$

We also define $R^2 = m^2 + r^2 = \|x\|^2$. For both cases we take the critical scaling of $\delta_n = c_\delta/n$ where $c_\delta > 0$ is a constant. Since our goal is to estimate the direction $v$, we wish to optimize the ratio $|\frac{m}{R}|$, representing the magnitude of the cosine ratio between $x$ and $v$.

**Proposition 3.1.** Fix $k \geq 2, \lambda > 0, c_\delta > 0$, let $\delta_n = c_\delta/n$. For $\beta \in [0, 1)$, $\boldsymbol{u}_n(t)$ following the dynamics induced by SGD-M converges as $n \to \infty$ to the solution of the following ODE initialized from $\lim_{n\to\infty} (\boldsymbol{u}_n)_* \mu_n$:

$$\mathrm{d}m = \frac{1}{1-\beta} 2m \left(\lambda k m^{k-2} - k R^{2(k-1)}\right) \mathrm{d}t, \quad \mathrm{d}r^2 = -\frac{4k R^{2(k-1)}}{(1-\beta)^2} \left(r^2 (1-\beta) - c_\delta\right) \mathrm{d}t \tag{5}$$

Likewise for SGD-U, $\boldsymbol{u}_n(t)$ converges weakly to the solution of the system

$$\mathrm{d}m = \sqrt{k} \left(\lambda \left(\frac{m}{R}\right)^{k-1} - R^{k-1} m\right) \mathrm{d}t, \quad \mathrm{d}r^2 = -2\sqrt{k} \left(R^{k-1} r^2 - \frac{c_\delta}{2\sqrt{k}}\right) \mathrm{d}t \tag{6}$$

We verify localizability and closability for both SGD-M and SGD-U in Sections C.1.1, C.1.2 as part of the proof of Proposition 3.1. We analyze the fixed points for both systems in the presented order. We find that the dynamics yield fixed points away from the axis $m = 0$ only if $\lambda$ exceeds a critical value, say $\lambda_{\text{crit}}(k, \beta, c_\delta)$, that depends on the tensor order, $k$, step-size $c_\delta$ and $\beta$. For both SGD-M and SGD-U, we can show (see Appendix C.2) that the critical $\lambda$ for the respective methods are

$$\lambda_{\text{crit}}^M(k, \beta, c_\delta) = \left(\frac{c_\delta}{k(1-\beta)}\right)^{k/2} \left[\frac{[2(k-1)]^{k-1}}{[k-2]^{(k-2)/2}}\right]$$

$$\lambda_{\text{crit}}^U(k, c_\delta) = \left[ \frac{\left( \frac{c_\delta}{2\sqrt{k}} \right)^k \left( (k-1)(k+2) \right)^{\frac{(k-2)(k+1)}{2}+k}}{(2k)^k \left( (k-2)(k+1) \right)^{\frac{(k-2)(k+1)}{2}}} \right]^{1/(k+1)}$$

We fix $k = 2$ in the following to determine the stability of the fixed points.

**Proposition 3.2** (Fixed Points - SGD-M). Let $k = 2$, $\lambda > 0$, $c_\delta > 0$, $\beta \in [0, 1)$ and denote $c_\delta^*$ by $c_\delta^* = c_\delta/(1-\beta)$ with $\lambda_{\text{crit}}^M(2, \beta, c_\delta) = c_\delta^*$. We then have the following fixed points:

1. An unstable fixed point at $(0, 0)$ and a fixed point at $(0, c_\delta^*)$ that is stable if $\lambda < \lambda_{\text{crit}}^M(2, \beta, c_\delta)$ and unstable if $\lambda > \lambda_{\text{crit}}^M(2, \beta, c_\delta)$

2. If $\lambda > \lambda_{\text{crit}}^M(2, \beta, c_\delta)$: two stable fixed points at $(\pm m_\star, c_\delta^*)$ where $m_\star = \sqrt{\lambda - c_\delta^*}$.

**Proposition 3.3** (Fixed Points - SGD-U). Let $k = 2$, $\lambda > 0$, $c_\delta > 0$, and denote $c_\delta^\dagger$ by $c_\delta^\dagger = c_\delta/(2\sqrt{2})$ with $\lambda_{\text{crit}}^U(2, \beta, c_\delta) = (c_\delta^\dagger)^{2/3}$. We then have the following fixed points:

1. An unstable fixed point at $(0, 0)$ and a fixed point at $(0, (c_\delta^\dagger)^{2/3})$ that is stable if $\lambda < \lambda_{\text{crit}}^U(2, c_\delta)$ and unstable if $\lambda > \lambda_{\text{crit}}^U(2, c_\delta)$

2. If $\lambda > \lambda_{\text{crit}}^U(2, c_\delta)$: two stable fixed points $(\pm m_\star, c_\delta^\dagger/\sqrt{\lambda})$ with $m_\star = \sqrt{\lambda - (c_\delta^\dagger/\sqrt{\lambda})}$.

In Figure 1 (top row), we provide simulations for Matrix PCA (i.e. $k = 2$) with fixed $c_\delta = 1$ to demonstrate how SGD-U and SGD-M with different values for $\beta$ and $\lambda$ behave in the ballistic phase. We also overlay the algorithmic trajectories with the predicted ODEs (presented as dashed lines in the figure) given in Proposition 3.1. We see that as $\lambda_{\text{crit}}^U(2, c_\delta) < \lambda_{\text{crit}}^M(2, \beta, c_\delta)$ for $\beta \geq 0$, there are values of $\lambda$ for which SGD-U is *supercritical* when SGD-M and online SGD are *subcritical*. This means that SGD-U is able to converge towards one of the fixed points away from $m = 0$ for values of $\lambda$ where the other methods fail to do so. A careful analysis reveals different settings of $c_\delta$ and $\lambda$ for which SGD-M may converge to desirable critical points instead. For example, fixing $c_\delta$, the fixed points suggest that SGD-U is preferred whenever $\lambda > (1-\beta)^2/8$. Similarly, the emergence of fixed points away from $m = 0$ depends on the choice of $c_\delta$ relative to $\lambda$. Indeed, one can verify that the largest $c_\delta$ for which non-trivial fixed points emerge is larger under SGD-U than SGD-M whenever $\lambda > (1-\beta)^2/8$. This suggests that there is a problem-specific critical $\lambda = 1/8$ dictating which of online SGD or SGD-U is preferred.

In practice, the choice of $c_\delta$ is specified by the user. Thus one may be inclined to simply take $c_\delta$ small enough such that the non-trivial fixed points emerge. However, the number of iterations (and thus the requisite samples and algorithm run-time) scales as $Tn/c_\delta$, i.e., inversely with $c_\delta$, which naturally constrains the step-size from being arbitrarily small. Consequently, taking $c_\delta$ to be too small may prevent one from running the algorithm long enough for the dynamics to even reach the basin of stability for the emergent fixed point.

### 3.1.1 DIFFUSIVE LIMITS AT THE EQUATOR ($m = 0$)

Here, we demonstrate an example of diffusive dynamics for Tensor PCA around $m = 0$ for SGD-U. We consider the rescaled observables $\tilde{u}_n(t) = (\tilde{u}_1, \tilde{u}_2) = (\sqrt{n}m, r^2)$.

**Proposition 3.4.** Fix $k \geq 2$, $\lambda > 0$ and $\delta_n = c_\delta/n$. Then for SGD-U, $\tilde{u}_n(t)$ converges as $n \to \infty$ to the solution of the following SDE initialized from $\tilde{\nu} = \lim_n (\tilde{u}_n)_* \mu_n = \mathcal{N}(0, 1) \otimes \delta_1$:

$$d\tilde{m} = \left( \lambda \left( \frac{\tilde{m}}{r} \right)^{k-1} \mathbb{1}_{k=2} - r^{k-1} \tilde{m} \right) dt + \sqrt{c_\delta} \left( (k-1)\frac{\tilde{m}^2}{r^2} + 1 \right)^{1/2} dB_t \tag{7}$$

$$dr^2 = -2 \left( r^{k+1} - \frac{c_\delta}{2} \right) dt \tag{8}$$

We see that $r^2$ now solves an autonomous ODE which converges exponentially to $[c_\delta/2]^{2/(k+1)}$. In particular, when $k = 2$ and as $t$ tends to $\infty$, the equation for $\tilde{m}$ behaves like

$$d\tilde{m} = \frac{2^{1/3}\tilde{m}}{c_\delta^{1/3}} \left( \lambda - \left[ \frac{c_\delta}{2} \right]^{2/3} \right) dt + \left( \frac{2^{2/3}\tilde{m}^2}{c_\delta^{1/3}} + c_\delta \right)^{1/2} dB_t$$

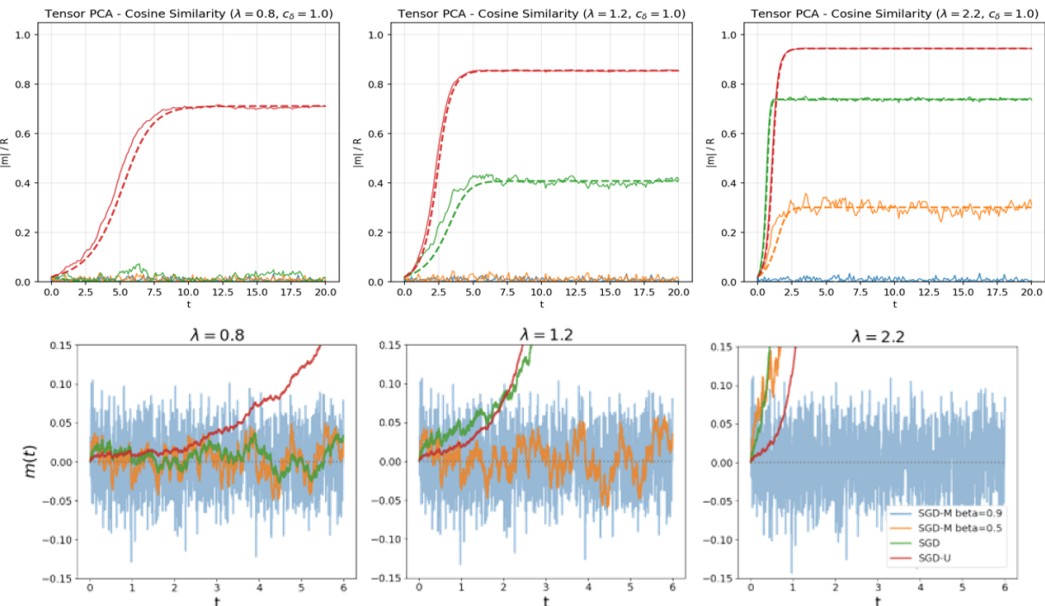

Figure 1: Matrix PCA in dimension $n = 10000$ for $\lambda = 0.8$ (left figure), $\lambda = 1.2$ (middle figure), and $\lambda = 2.2$ (right figure) with $c_\delta = 1$. Top row depicts the evolution of the statistic $|m(t)|/R(t)$ for $20n$ steps with random initialization with predicted ODEs overlayed as dashed lines. Bottom row depicts the evolution of the rescaled statistic $\tilde{u}_1 = \sqrt{n}\, m(t)$ for $6n$ steps around a fixed window about $m = 0$. We note the increased volatility of the diffusive limits for SGD-M as $\beta$ increases.

For the ballistic phase (top row), we note that in the left figure, only SGD-U is supercritical. In the middle figure, both SGD-U and online SGD are supercritical, however the former attains better alignment with the direction vector $v$. The rightmost figure shows only SGD-M with $\beta = 0.9$ as subcritical, however the alignment order is consistent. For the diffusive phase (bottom row), we note the increased volatility of the diffusive limits for SGD-M as $\beta$ increases. We also observe that SGD-U becomes mean repellent for smaller values of $\lambda$ than SGD-M.

Interestingly, in contrast to the dynamics for online SGD (Ben Arous et al., 2024, Proposition 3.4), equation 7 is no longer a standard OU process as the diffusion coefficient explicitly depends on $\tilde{m}$. However it is still reminiscent of one since, for example, the drift is mean-reverting or mean-repellent depending on the choice of $c_\delta$ relative to $\lambda$. Figure 1 (bottom row) presents several simulations of the diffusive limits for $\tilde{u}_1$ under different setups of SGD-M and SGD-U.

## 3.2 SINGLE INDEX MODELS

Let $v \in \mathbb{S}^{n-1}$ be a fixed direction and suppose we are given i.i.d. samples of data $(y^\ell, a^\ell)_{\ell \geq 1}$ under the model

$$y^\ell = f(a^\ell \cdot v) + \epsilon^\ell$$

where $f : \mathbb{R} \to \mathbb{R}$ is a possibly non-linear *link* function (which is assumed to be known) and $(\epsilon^\ell)$ are zero-mean additive errors with $\mathbb{E}[\epsilon^2] = \sigma^2 < \infty$. Our goal is to infer $v$ from $(y^\ell, a^\ell)$ where $a^\ell$ are i.i.d. standard Gaussian feature vectors by optimizing the quadratic loss,

$$L(x, Y) = \left(y^\ell - f(a^\ell \cdot x)\right)^2 = (f(a^\ell \cdot v) - f(a^\ell \cdot x) + \epsilon^\ell)^2$$

Similar to the previous example, we determine the evolution of the pair of summary statistics $\boldsymbol{u}_n = (m, r^2)$. The scaling limit for this problem was analyzed for online SGD by Saad & Solla (1995a;b); Goldt et al. (2019); Veiga et al. (2022); Rangriz (2025). The critical scaling is determined by the step-size $\delta_n = c_\delta/n$.

**Proposition 3.5.** Fix $c_\delta > 0$, non-constant link function $f : \mathbb{R} \to \mathbb{R}$, let $\delta_n = c_\delta/n$. For $\beta \in [0, 1)$, $\boldsymbol{u}_n(t)$ following the dynamics induced by SGD-M converges as $n \to \infty$ to the solution of the

following ODE initialized from $\lim_{n\to\infty} (\boldsymbol{u}_n)_* \mu_n$:

$$\mathrm{d}m = \frac{-2}{(1-\beta)}\mathbb{E}\left[a_1 f'(a_1 m + a_2 r)(f(a_1 m + a_2 r) - f(a_1))\right] \mathrm{d}t, \tag{9}$$

$$\mathrm{d}r^2 = \frac{-4}{(1-\beta)}\left\{\mathbb{E}\left[a_2 r f'(a_1 m + a_2 r)(f(a_1 m + a_2 r) - f(a_1))\right]\right.$$
$$\left. - \frac{c_\delta}{1-\beta}\mathbb{E}\left[f'(a_1 m + a_2 r)^2\left((f(a_1 m + a_2 r) - f(a_1))^2 + \sigma^2\right)\right]\right\} \mathrm{d}t \tag{10}$$

where $a_1, a_2 \sim \mathcal{N}(0, 1)$ are i.i.d. standard Gaussian variables. Likewise for SGD-U, $\boldsymbol{u}_n(t)$ converges weakly to the solution of the system,

$$\mathrm{d}m = -\mathbb{E}[a_1 \cdot \mathrm{sgn}(f'(ma_1 + ra_2)(f(ma_1 + ra_2) - f(a_1) + \epsilon))] \mathrm{d}t, \tag{11}$$

$$\mathrm{d}r^2 = -2\left\{r\mathbb{E}[a_2 \cdot \mathrm{sgn}(f'(ma_1 + ra_2)(f(ma_1 + ra_2) - f(a_1) + \epsilon))] - \frac{c_\delta}{2}\right\} \mathrm{d}t \tag{12}$$

where the sign function satisfies $\mathrm{sgn}(x) = \mathbb{1}_{x>0} - \mathbb{1}_{x<0}$.

We verify localizability and closability for both SGD-M and SGD-U as part of the proof of Proposition 3.5 in Section D. In what follows, we consider $f(x) = x^k$ for $k \geq 1$. This family of link functions includes well-studied examples such as phase retrieval ($k = 2$). In the small noise regime $\sigma^2 \to 0$, the population optimum $(m, r^2) = (1, 0)$ is a fixed point for SGD-M (see Proposition 3.8), but not for SGD-U. Despite this, we show that SGD-U is able to converge to fixed points that are still close to the optimum in settings where SGD-M diverges (i.e. $\mathrm{d}r^2 > 0, \forall t$).

For concreteness, we present some specified choices of $f$ to gauge the exact limiting dynamics and fixed points for SGD-M. We obtain closed-form dynamics and fixed points for SGD-U in the next section where we consider the small noise regime.

**Proposition 3.6.** If we take $f(x) = x$, then the continuous-time dynamics for SGD-M in Proposition 3.5 with $\beta \in [0, 1)$ reduce to

$$\mathrm{d}m = -\frac{2}{1-\beta}(m-1)\,\mathrm{d}t, \quad \mathrm{d}r^2 = -\frac{4}{1-\beta}\left(r^2 - \frac{c_\delta}{1-\beta}\left((m-1)^2 + r^2\right) + \sigma^2\right)\mathrm{d}t$$

with $\sigma^2 > 0$. In particular, the dynamics admit a unique fixed point $(m, r^2) = (1, c_\delta \sigma^2/(1-\beta-c_\delta))$ if and only if $c_\delta < 1 - \beta$.

**Proposition 3.7.** If we take $f(x) = x^2$, then the continuous-time dynamics for SGD-M in Proposition 3.5 with $\beta \in [0, 1)$ reduce to

$$\mathrm{d}m = -\frac{12}{1-\beta}m(R^2 - 1)\,\mathrm{d}t, \quad \mathrm{d}r^2 = -\frac{4}{1-\beta}\left(2r^2(3R^2 - 1) - \frac{4c_\delta}{1-\beta}\left(P(m,r) + \sigma^2 R^2\right)\right)\mathrm{d}t$$

with $\sigma^2 > 0$ and where we define the polynomial

$$P(m, r) = 15m^6 + 45m^4 r^2 + 45m^2 r^4 + 15r^6 - 30m^4 - 36m^2 r^2 - 6r^4 + 15m^2 + 3r^2$$

In particular, the dynamics admit the following fixed points:

1. For $c_\delta > (1 - \beta)/12$, we have a fixed point at $(m, r^2) = (0, 0)$ and additional fixed points $(m, r^2) = (0, r_p^2)$ for $r_p^2 > 0$ where $r_p^2$ are roots (assuming they exist) to the equation,

$$6r^2 - 2 = \frac{4c_\delta}{1-\beta}\left(15r^4 - 6r^2 + 3 + \sigma^2\right)$$

2. For $c_\delta < (1 - \beta)/12$, we obtain the pair of fixed points $(m_*, r_*^2)$ for $r_*^2 > 0$ where $r_*^2 = c_\delta \sigma^2/(1 - \beta - 12c_\delta)$ and $m_* = \pm\sqrt{1 - r_*^2}$.

The proof for both results is provided in Appendix D.2. These results inform us of the inherent difficulty for the SGD-M (and consequently online SGD) dynamics to converge to fixed points near the population minimum. To attain these fixed points, we require taking a smaller $c_\delta$, thus increasing number of iterations to run the algorithm as per our discussion in Section 3.1. We capture this phenomena more generally in the small noise limit next.

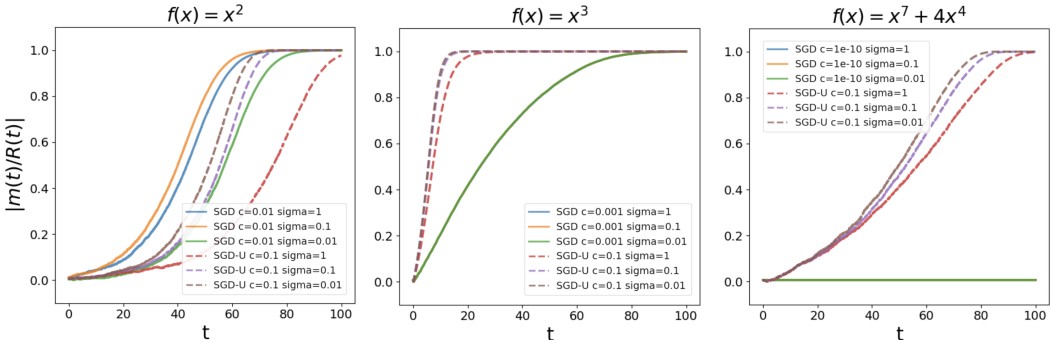

Figure 2: We plot the value of $|m/R|$ over the course of training for independent runs of SGD (full lines) and SGD-U (dashed lines) for various functions $f$ with different amounts of additive noise. We set $n = 10,000$, $\delta = c_\delta/n$ and the total number of steps is taken as one million. We consider $f(x) = x^2$, $f(x) = x^3$ and $f(x) = x^7 + 4x^4$. In each case we choose $c_\delta = 10^{-k}$ for the smallest integer k such that the dynamics are not effected by exploding gradients.

### 3.2.1 SMALL NOISE $\sigma^2 \to 0$ LIMIT

In this section, we consider the regime where we take $\sigma^2 \to 0$ after $n \to \infty$ to obtain closed-form dynamics and fixed points for SGD-U. We also generalize our observations from Propositions 3.6, 3.7 to the family of monomial link functions $f(x) = x^k$, $k \geq 1$. To do so, we determine the existence of stable fixed points along the axis $m = \pm 1$. The following result considers the fixed point along $m = 1$, however the same holds for $m = -1$ for even-powered monomials by symmetry.

**Proposition 3.8.** Let $k \geq 1$, $\beta \in [0, 1)$ and take $\sigma^2 \to 0$. For the link function $f(x) = x^k$, the dynamics of SGD-M, as given in Proposition 3.5, admit a locally stable fixed point at $(m_*, r_*^2) = (1, 0)$ if and only if we take $c_\delta$ such that

$$0 < c_\delta < \frac{1 - \beta}{k^2} \cdot \frac{(2k - 3)!!}{(4k - 5)!!} \tag{13}$$

For SGD-M, increasing the degree of the monomial link restricts the admissible values of $c_\delta$ for which the dynamics converge to fixed points away from $m = 0$. In particular, one can show (e.g. see Appendix D.2) that the rate for which the upper bound decreases is of the order $k^{-k-2}$. In particular, we observe that even though $(1, 0)$ is a fixed point for any $c_\delta > 0$ when $\sigma^2 = 0$, it is not stable unless $c_\delta$ satisfies equation 13. We next determine the dynamics for a large collection of link functions for SGD-U in the small-noise regime.

**Proposition 3.9.** If $\sigma \to 0$, then for every strictly increasing link function $f$, SGD-U admits the following dynamics,

$$\mathrm{d}m = -\sqrt{\frac{2}{\pi}} \frac{m - 1}{\sqrt{(m - 1)^2 + r^2}} \, \mathrm{d}t, \quad \mathrm{d}r^2 = \left[ -\frac{2\sqrt{2/\pi} r^2}{\sqrt{(m - 1)^2 + r^2}} + c_\delta \right] \mathrm{d}t$$

These dynamics yield a unique fixed point at $(m, r^2) = (1, c_\delta^2 \pi/8)$.

This can be viewed as a type of "universality" result in the link function. For odd-powered monomials, equation 13 illustrated how $c_\delta$ must be super-exponentially small in $k$ for the dynamics of SGD-M to converge to a fixed point about $m = 1$. However, in Proposition 3.9, we see that regardless of the degree, the dynamics under SGD-U always converge to a fixed point along $m = 1$ with $r^2 > 0$. Setting $c_\delta$ arbitrarily small thus recovers fixed points near the population optimum. We can view this as a rigorous justification for how using normalized gradients, as in SGD-U, allows for choosing larger step-sizes for which the learning dynamics remain reasonable. This example validates some of the heuristics and intuition behind using unit-norm gradients to control exploding and vanishing gradients, causing instability for SGD (Mikolov, 2012; Pascanu et al., 2013).

To complement Proposition 3.9, we provide additional analysis for even-powered monomials in Appendix D.5. We show a similar universality result where the dynamics are again invariant with

respect to the monomial's degree. To summarize our results, we show that there also exists a threshold for $c_\delta$ that determines the existence of fixed points away from $m = 0$. In particular, we find this threshold to be $4/\sqrt{10\pi} \approx 0.71$, which is larger than, for example, the $1/12$ threshold observed for phase retrieval in Proposition 3.7. We suspect a similar phenomenon to occur for more general polynomials.

In Figure 2, we provide simulations for online SGD and SGD-U with varying choices of $f$ and $\sigma^2 > 0$ where $\epsilon^\ell \sim \mathcal{N}(0, \sigma^2)$ are i.i.d. Gaussian variables. For each algorithm, we choose $c_\delta$ such that the dynamics converge to a fixed point. We see in the first two figures that both algorithms recover the solution, although the step-size required to do so is smaller by one and two orders of magnitude respectively for SGD compared to SGD-U. For the third figure, we take an arbitrary large degree polynomial and see that we require $c_\delta \leq 10^{-10}$ for SGD not to be effected by exploding gradients. As a result, the dynamics fail to progress within the specified number of iterations. In contrast, SGD-U is able to recover the true direction with the same step-size in all our examples within the same timescale.

### 3.2.2 DIFFUSIVE LIMITS AT FIXED POINTS

Using our results, one can also obtain diffusive limits around appropriate fixed points for rescaled observables, similar to Section 3.1.1. We illustrate this by considering the diffusive limit for phase retrieval at $m = 0$ (i.e. at initialization).

**Proposition 3.10.** For the link function $f(x) = x^2$, The dynamics for the rescaled statistics $\tilde{\boldsymbol{u}}_n(t) = (\tilde{m}, r^2) = (\sqrt{n}m, r^2)$ converges to the solution of,

$$\mathrm{d}\tilde{m} = -\frac{12(r^2 - 1)}{1 - \beta}\tilde{m}\,\mathrm{d}t + \frac{4r}{1 - \beta}\sqrt{c_\delta(15r^4 - 18r^2 + 15 + \sigma^2)}\,\mathrm{d}B_t$$

$$\mathrm{d}r^2 = -\frac{8r^2}{1 - \beta}\left[3r^2 - 1 + \frac{2c_\delta(15r^4 - 6r^2 + 3 + \sigma^2)}{1 - \beta}\right]\mathrm{d}t \qquad (14)$$

The proof is provided in Appendix D.3. We note that the evolution of $r^2$ is independent of $\tilde{m}$. Furthermore, the equator $\tilde{m} = 0$ is unstable when $r^2 < 1$, providing similar insight to the ballistic phase where we observed the emergence of stable fixed points away from the equator for sufficiently small $c_\delta$ (which indirectly forces $r^2 < 1$ at such point(s)). One could also obtain a similar diffusive limit for the SGD-U dynamics obtained in Proposition 3.9, however as the fixed point is unique and stable we find that these dynamics may not be as informative.

## 4 CONCLUDING REMARKS

We develop a general theory for the high-dimensional scaling limits of online SGD with Polyak momentum and scalar Markovian pre-conditioning under the effective dynamics framework of Ben Arous et al. (2024). We illustrate our results on two canonical models in high-dimensional inference: Spiked Tensor PCA and Single Index models. For both problems, we consider both the ballistic and diffusive limits of SGD-M and SGD-U, with the latter setting the preconditioner as the inverse gradient norm. In doing so, we provide rigorous examples of how SGD-U is able to stabilize training dynamics and high-dimensional effects leading to more desirable training outcomes as observed empirically. In particular, we illustrate how the effective dynamics framework can sharply capture empirical phenomenon such as exploding/vanishing gradients in the limiting dynamics.

Given these results, there are several interesting directions to explore for future work. Of particular interest is the problem of developing limits of more general scalar preconditioners and develop a systematic analysis comparing and contrasting preconditoning and momentum based methods on broader classes of statistical problems. It would also be interesting to determine the limiting dynamics of popular matrix-based preconditioners (Duchi et al., 2011; Hinton, 2012; Kingma & Ba, 2014) and whether they mitigate the high-dimensional corrections.

ACKNOWLEDGMENTS

A.J. acknowledges the support of the Natural Sciences and Engineering Research Council of Canada (NSERC), the Canada Research Chairs program,the Canadian Foundation for Innovation- John Evan's Leaders fund, and the Ontario Research Fund. T.J and V.S. acknowledge the support of the Natural Sciences and Engineering Research Council of Canada (NSERC). Cette recherche a été enterprise grâce, en partie, au soutien financier du Conseil de Recherches en Sciences Naturelles et en Génie du Canada (CRSNG), du Programme des chaires de recherche du Canada, et du Foundation canadienne pour l'innovation FLJE, et les Fonds pour la recerce en Ontario. [RGPIN-2020-04597, DGECR-2020-00199,CRC-2022-00142,CFI-JELF Project 43994]

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

## A  GENERALIZATION OF THEOREM 2.3 FOR PRE-CONDITIONING

Here, we provide a generalized theorem (with adequate conditions) that allows one to prove similar scaling limits in the setting where we adopt scalar Markovian preconditioners for SGD.

Consider the following online learning setup. Suppose we are given an i.i.d. data sequence $\boldsymbol{y}_1, \boldsymbol{y}_2, \ldots$ taking values in $\mathcal{Y} \subseteq \mathbb{R}^{d_n}$ with law $P_n \in \mathcal{M}_1(\mathbb{R}^{d_n})$ and loss $L : \mathcal{X}_n \times \mathcal{Y}_n \to \mathbb{R}$ where $\mathcal{X}_n \subseteq \mathbb{R}^{p_n}$ is the parameter space. Consider online SGD with learning rate $\delta_n > 0$, momentum rate $\beta \in [0, 1)$ and the scalar and Markov preconditioner $\eta_n(x, y) : \mathcal{X}_n \times \mathcal{Y}_n \to \mathbb{R}^+$, given by:

$$\boldsymbol{p}_\ell = \beta \boldsymbol{p}_{\ell-1} - \delta_n \, \eta_n(\boldsymbol{x}_{\ell-1}, \boldsymbol{y}_\ell) \, \nabla L_n(\boldsymbol{x}_{\ell-1}, \boldsymbol{y}_\ell)$$
$$\boldsymbol{x}_\ell = \boldsymbol{x}_{\ell-1} + \boldsymbol{p}_\ell$$

with possibly random initialization $\boldsymbol{x}_0 \sim \mu_n \in \mathcal{M}_1(\mathcal{X}_n)$. Our interest is understanding the evolution of a finite collection of summary statistics $\boldsymbol{u}_n(\boldsymbol{x}) = (u_1^n(\boldsymbol{x}), \ldots, u_k^n(\boldsymbol{x}))$ in the regime where both $p_n$ and $d_n$ may grow with $n$, and $\delta_n \to 0$ as $n \to \infty$.

We define the functions

$$\nabla \tilde{H}(x, Y) = \eta_n(x, Y) \nabla L_n(x, Y) - \nabla \tilde{\Phi}(x) \quad \text{where} \quad \nabla \tilde{\Phi}(x) = \mathbb{E}[\eta_n(x, Y) \nabla L_n(x, Y)] \quad (15)$$

Let $\tilde{V}(x) = \mathbb{E}[\nabla \tilde{H}(x) \otimes \nabla \tilde{H}(x)]$ be the covariance matrix for $\nabla \tilde{H}$ at $x$.

We state two assumptions: $(\delta_n, \eta_n)$-localizability and asymptotic-closability.

**Definition A.1.** A tuple $(\boldsymbol{u}_n, \nabla L_n, P_n, \eta_n, \beta)$ is $(\boldsymbol{\delta_n}, \boldsymbol{\eta_n})$**-localizable** with localizing sequence $(E_K)_K$ if there is an exhaustion by compacts $(E_K)_K$ of $\mathbb{R}^k$, and constants $C_K$ (independent of $n$) such that

1. $\sup_{x \in \boldsymbol{u}_n^{-1}(E_K)} \|\nabla^2 u_i^n\|_{\text{op}} \leq C_K \cdot \delta_n^{-1/2}$, and $\sup_{x \in \boldsymbol{u}_n^{-1}(E_K)} \|\nabla^3 u_i^n\|_{\text{op}} \leq C_K$;

2. $\sup_{x \in \boldsymbol{u}_n^{-1}(E_K)} \|\nabla \tilde{\Phi}\| \leq C_K$, and $\sup_{x \in \boldsymbol{u}_n^{-1}(E_K)} \mathbb{E}[\|\nabla \tilde{H}\|^8] \leq C_K \delta_n^{-4}$;

3. $\sup_{x \in \boldsymbol{u}_n^{-1}(E_K)} \mathbb{E}[\langle \nabla \tilde{H}, \nabla u_i^n \rangle^4] \leq C_K \delta_n^{-2}$, and
   $\sup_{x \in \boldsymbol{u}_n^{-1}(E_K)} \mathbb{E}[\langle \nabla^2 u_i^n(x), \nabla \tilde{H}(x, y) \otimes \nabla \tilde{H}(x, y) - V(x) \rangle^2] = o(\delta_n^{-3})$, and
   $\sup_{(x_i)_1^2 \in \boldsymbol{u}_n^{-1}(E_K)} \mathbb{E}_{y_1 \perp y_2}[\langle \nabla^2 u_i^n(x_1), \nabla \tilde{H}(x_1, y_1) \otimes \nabla \tilde{H}(x_2, y_2) \rangle^2] = o(\delta_n^{-3})$ for $\beta > 0$

The second assumption ensures that the limiting coefficients for the evolution of the statistics close. To this end, we define the first and second-order differential operators,

$$\tilde{\mathcal{A}}_n = \langle \nabla \tilde{\Phi}, \nabla \rangle \quad \text{and} \quad \tilde{\mathcal{L}}_n = \frac{1}{2} \langle \tilde{V}, \nabla^2 \rangle$$

**Definition A.2.** A family of summary statistics $(\boldsymbol{u}_n)$ are **asymptotically closable** for learning and momentum rates $((\delta_n)_n, \beta)$, and preconditioners $(\eta_n)_n$ if $(\boldsymbol{u}_n, \nabla L_n, P_n, \eta_n, \beta)$ are $(\delta_n, \eta_n)$-localizable with localizing sequence $(E_K)_K$, and furthermore there exist locally Lipschitz functions $\tilde{\boldsymbol{h}} : \mathbb{R}^k \times \mathbb{R} \to \mathbb{R}^k$ and $\tilde{\boldsymbol{\Sigma}} : \mathbb{R}^k \to \mathbb{R}^{k \times k}$, such that

$$\sup_{x \in \boldsymbol{u}_n^{-1}(E_K)} \left\| \left( -\frac{1}{1-\beta} \tilde{\mathcal{A}}_n + \frac{\delta_n}{(1-\beta)^2} \tilde{\mathcal{L}}_n \right) \boldsymbol{u}_n(x) - \tilde{\boldsymbol{h}}(\beta, \boldsymbol{u}_n(x)) \right\| \to 0$$

$$\sup_{x \in \boldsymbol{u}_n^{-1}(E_K)} \|\delta_n J_n \tilde{V} J_n^T - \tilde{\boldsymbol{\Sigma}}(\boldsymbol{u}_n(x))\| \to 0$$

with $J_n$ the Jacobian of $\boldsymbol{u}_n$. We call $\boldsymbol{h}$ the *effective drift* and $\boldsymbol{\Sigma}$ the *effective volatility* of $\boldsymbol{u}$ respectively.

**Theorem A.3.** Let $(X_\ell^{\delta_n})_\ell$ be SGD initialized from $X_0 \sim \mu_n$ for $\mu_n \in \mathcal{M}_1(\mathbb{R}^{p_n})$ with learning rate $\delta_n$, momentum rate $\beta \in [0, 1)$ and pre-conditioner $\eta_n(x, y)$ for the loss $L_n(\cdot, \cdot)$ and data distribution $P_n$. For a family of summary statistics $\boldsymbol{u}_n = (u_i^n)_{i=1}^k$, let $(\boldsymbol{u}_n(t))_t$ be the linear interpolation of $(\boldsymbol{u}_n(X_{\lfloor t\delta_n^{-1} \rfloor}^{\delta_n}))_t$.

Suppose that $\boldsymbol{u}_n$ are asymptotically closable with learning rate $\delta_n$, momentum rate $\beta$, preconditioners $(\eta_n)_n$, effective drift $\tilde{\boldsymbol{h}}$, and effective volatility $\tilde{\boldsymbol{\Sigma}}$, and that the pushforward of the initial data has $(\boldsymbol{u}_n)_* \mu_n \to \nu$ weakly for some $\nu \in \mathscr{M}_1\left(\mathbb{R}^k\right)$. Then $(\boldsymbol{u}_n(t))_t \to (\boldsymbol{u}_t)_t$ weakly as $n \to \infty$, where $\boldsymbol{u}_t$ solves:

$$\mathrm{d}\boldsymbol{u}_t = \tilde{\boldsymbol{h}}(\beta, \boldsymbol{u}_t)\,\mathrm{d}t + \frac{1}{1 - \beta}\sqrt{\tilde{\boldsymbol{\Sigma}}\left(\boldsymbol{u}_t\right)}\,\mathrm{d}\mathbf{B}_t$$

initialized from $\nu$, where $\mathbf{B}_t$ is a standard Brownian motion in $\mathbb{R}^k$.

This result is essentially identical to Theorem 2.3 under the revised constructions of $\nabla\Phi$, $\nabla H$, to $\nabla\tilde{\Phi}$ and $\nabla\tilde{H}$. That is, by augmenting the loss gradient with the preconditioner to obtain the coupled gradient $\eta_n \nabla L$, we can leverage the same mechanics as those for SGD-M to immediately obtain scaling limits for SGD-M equipped with Markovian scalar preconditioners. To understand why this holds, we note that $\nabla L_n$ is treated as an arbitrary vector field satisfying specific regularity conditions in the proof of Theorem 2.3. The fact that $\nabla L_n$ is the gradient is irrelevant, and in principle can be replaced with any arbitrary estimator satisfying Definitions A.1, A.2. In this case, we replace $\nabla L$ with the "estimator" $\eta \nabla L$.

## B    PROOF OF THEOREM 2.3

Our goal is to establish the weak convergence $\boldsymbol{u}_n \Rightarrow \boldsymbol{u}$ as random variables on the space of continuous functions $C([0, \infty))$, where $\boldsymbol{u}$ solves the limiting SDE. It suffices to show the same on $C([0, T])$, endowed with the uniform (i.e. sup-norm) topology, for every $T > 0$. The proof proceeds by constructing an auxiliary process $\boldsymbol{z}_n$, defined in equation 16, such that

$$\sup_{x \in E_{K,n}^*} \mathbb{E}[\|\boldsymbol{u}_n - \boldsymbol{z}_n\|] \to 0$$

Let $\tau_K^n$ denote the exit time of the interpolated process $\boldsymbol{u}_n(t)$ from the compact set $E_K^n$. Furthermore, let $E_{K,n}^* := \boldsymbol{u}_n^{-1}(E_K^n)$ denote the corresponding pre-image, and $L_{K,n}^\infty := L^\infty(E_{K,n}^*)$ the associated space of localized bounded functions. The subsequent analysis relies on several technical lemmas, which hold under the aysmptotic-closability and -$\delta_n$-localizability assumptions detailed in Definition 2.1, 2.2. We assume $\boldsymbol{p}_0 = 0$ unless stated otherwise.

To simplify notation, we use the subscript notation $f_j := f(\boldsymbol{x}_j)$ for quantities that depend on $\boldsymbol{x}_j$. In particular, we write $g_j := g(\boldsymbol{x}_j, Y_{j+1})$ for functions that further depend on the random data $Y_{j+1}$. Finally, we say that $x \lesssim y$ if there is some constant $C > 0$ such that $f \leq Cg$ and that $f \lesssim_a g$ is there is some constant $C(a) > 0$ depending only on $a$ such that $f \leq C(a)g$.

**Lemma B.1.** For any integers $0 \leq i < j \leq \ell - 1$, the momentum variable satisfies the recurrence

$$\boldsymbol{p}_{\ell-1-i} = \beta^{j-i}\boldsymbol{p}_{\ell-1-j} - \delta \sum_{m=0}^{j-i-1} \beta^m \nabla L_{\ell-2-i-m}.$$

**Lemma B.2.** For any $0 \leq i \leq \ell - 1$, the position variable admits the representation

$$\boldsymbol{x}_{\ell-1} = \boldsymbol{x}_{\ell-1-i} - \delta \sum_{m=0}^{i-1} \frac{1 - \beta^{m+1}}{1 - \beta} \nabla L_{\ell-2-m} - \delta \frac{1 + \beta^i}{1 - \beta} \sum_{m=0}^{\ell-2-i} \beta^{m+1} \nabla L_{\ell-2-i-m}.$$

**Lemma B.3.** For any $0 \leq i < \ell - 1$, the Hessian difference is bounded uniformly in time in $L^1$ by

$$\|\nabla^2 u(\boldsymbol{x}_{\ell-1}) - \nabla^2 u(\boldsymbol{x}_{\ell-1-i})\|_{\mathrm{op}} \leq O(i\sqrt{\delta}).$$

**Lemma B.4.** For any $0 \leq i < \ell - 1$, the gradient admits the expansion

$$\nabla u(\boldsymbol{x}_{\ell-1}) - \nabla u(\boldsymbol{x}_{\ell-1-i}) = -\delta \sum_{m=0}^{i-1} \frac{1 - \beta^{m+1}}{1 - \beta} \nabla^2 u(\boldsymbol{x}_{\ell-2-m}) \nabla H_{\ell-2-m}$$

$$- \delta \frac{1 + \beta^i}{1 - \beta} \sum_{m=0}^{\ell-2-i} \beta^{m+1} \nabla^2 u(\boldsymbol{x}_{\ell-2-i-m}) \nabla H_{\ell-2-i-m} + O(i^2\sqrt{\delta}),$$

where the trailing term holds uniformly in time in $L^1$.

We proceed by analyzing the process coordinate-wise. Let $u := u^j$ denote the $j$-th coordinate function of $\boldsymbol{u}$, and we again adopt the notation $u_\ell := u(\boldsymbol{x}_\ell)$. For any $\ell \leq \tau_K^n/\delta$, a second-order Taylor expansion yields

$$u(\boldsymbol{x}_\ell) = u(\boldsymbol{x}_{\ell-1} + \boldsymbol{p}_\ell) = u_{\ell-1} + \langle \boldsymbol{p}_\ell, \nabla u_{\ell-1} \rangle + \frac{1}{2} \langle \boldsymbol{p}_\ell \otimes \boldsymbol{p}_\ell, \nabla^2 u_{\ell-1} \rangle + \mathcal{R}_\ell,$$

where by $\delta$-localizability, the remainder term is bounded in $L^1$ by

$$\sup_{x \in E_{K,n}^*} \mathbb{E}\left[|\mathcal{R}_\ell|\right] \lesssim \sup_{x \in E_{K,n}^*} \mathbb{E}\left[\|\nabla^3 u\|_{L_{K,n}^\infty} \cdot \|\boldsymbol{p}_\ell\|_{L_{K,n}^\infty}^3\right]$$

$$\lesssim \delta^3 \|\nabla^3 u\|_{L_{K,n}^\infty} \left(\sum_{i \geq 0} \beta^i (\|\nabla \Phi\|_{L_{K,n}^\infty} + \|\nabla H\|_{L_{K,n}^\infty})\right)^3 \lesssim_K \delta^{3/2}.$$

This bound holds uniformly for $\ell \leq \tau_K^n/\delta$. Substituting the definition of $\boldsymbol{p}_\ell$ and rearranging the expansion provides the explicit form of the increment:

$$u_\ell - u_{\ell-1} = -\delta \sum_{i=0}^{\ell-1} \beta^i \langle \nabla \Phi._{-i}, \nabla u \rangle_{\ell-1} - \delta \sum_{i=0}^{\ell-1} \beta^i \langle \nabla H._{-i}, \nabla u \rangle_{\ell-1}$$

$$+ \frac{\delta^2}{2} \sum_{i,j=0}^{\ell-1} \beta^{i+j} \langle \nabla \Phi._{-i} \otimes \nabla \Phi._{-j}, \nabla^2 u \rangle_{\ell-1}$$

$$+ \delta^2 \sum_{i,j=0}^{\ell-1} \beta^{i+j} \langle \nabla \Phi._{-i} \otimes \nabla H._{-j}, \nabla^2 u \rangle_{\ell-1}$$

$$+ \frac{\delta^2}{2} \sum_{i,j=0}^{\ell-1} \beta^{i+j} \langle \nabla H._{-i} \otimes \nabla H._{-j} - \mathbb{E}[\nabla H._{-i} \otimes \nabla H._{-j}], \nabla^2 u \rangle_{\ell-1}$$

$$+ \frac{\delta^2}{2} \sum_{i,j=0}^{\ell-1} \beta^{i+j} \langle \mathbb{E}[\nabla H._{-i} \otimes \nabla H._{-j}], \nabla^2 u \rangle_{\ell-1} + O(\delta^{3/2}).$$

Here, we employ the compact subscript notation $\langle a, b \rangle_\ell = \langle a_\ell, b_\ell \rangle$, $\langle a._{-i}, b \rangle_\ell = \langle a_{\ell-i}, b_\ell \rangle$ and $\langle a._{-i} \otimes c._{-j}, b \rangle_\ell = \langle a_{\ell-i} \otimes c_{\ell-j}, b_\ell \rangle$, assuming that the indexed quantities at $\ell$, $\ell - i$, $\ell - j$ are all well-defined.

The next several lemmas summarize how we construct the auxillary process $\boldsymbol{z}_n$ by identifying the terms in the expansion contributing to the drift and quadratic variation of the limiting SDE. We show that all other terms in the expansion vanish in $L^2$.

**Lemma B.5.** The evolution of the $j$-th coordinate of $u$, which we denote $u_j$ and $\nabla u^j$ for the respective gradient, admits the decomposition:

$$u_j(\boldsymbol{x}_\ell) - u_j(\boldsymbol{x}_0) = -\delta \sum_{\ell' \leq \ell} \sum_{i < \ell'} \beta^i \langle \nabla \Phi_{\ell'-1-i}, \nabla u_{\ell'-1}^j \rangle + \frac{\delta^2}{2} \sum_{\ell' \leq \ell} \sum_{i < \ell'} \beta^{2i} \langle V_{\ell'-1-i}, \nabla^2 u_{\ell'-1}^j \rangle$$

$$- \delta \sum_{\ell' \leq \ell} \sum_{i < \ell'} \beta^i \langle \nabla H_{\ell'-1-i}, \nabla u_{\ell'-1}^j \rangle + o(1).$$

where the remainder term is $o(1)$ in $L^1$, uniformly for $\ell \leq \tau_K^n/\delta$. In particular, we have that the first two series contribute to the limiting drift and the final term contributes to the limiting quadratic variation.

**Lemma B.6.** The martingale component, i.e. the third series in the expansion due to Lemma B.5, admits the asymptotic expansion

$$\delta \sum_{\ell=1}^{\lfloor t/\delta \rfloor} \sum_{i=1}^{\ell-1} \beta^i \langle \nabla H._{-i}, \nabla u \rangle_{\ell-1} = \frac{\delta}{1-\beta} \sum_{\ell=1}^{\lfloor t/\delta \rfloor} \langle \nabla H, \nabla u \rangle_{\ell-1}$$

$$- \delta^2 \sum_{\ell=1}^{\lfloor t/\delta \rfloor} \sum_{i=1}^{\ell-1} \beta^i \frac{1-\beta^i}{1-\beta} \langle V, \nabla^2 u \rangle_{\ell-1-i} + o(1).$$

where remainder is $o(1)$ in $L^1$, uniformly for $t \leq \tau_K^n$.

This final lemma obtains a similar asymptotic representation for the remaining two series in Lemma B.5.

**Lemma B.7.** The following asymptotic equivalences hold in $L^1$, uniformly for $\ell \leq \tau_K^n/\delta$:

$$\sum_{i=0}^{\ell-1} \beta^i \langle \nabla \Phi_{\cdot - i}, \nabla u \rangle_{\ell-1} = \frac{1}{1-\beta} \langle \nabla \Phi, \nabla u \rangle_{\ell-1} + o(\delta),$$

$$\frac{1}{2} \sum_{i=0}^{\ell-1} \left( \beta^{2i} + 2\beta^i \frac{1-\beta^i}{1-\beta} \right) \langle V_{\cdot-i}, \nabla^2 u \rangle_{\ell-1} = \frac{1}{2(1-\beta)^2} \langle V, \nabla^2 u \rangle_{\ell-1} + o(\delta).$$

By consolidating Lemmas B.5, B.6, and B.7, we construct the auxiliary process $z_n$ as follows. Consider the $j$-th component $z^j$, the discrete-time process is defined by the Doob decomposition:

$$z_\ell^j = z_0^j + \sum_{k=1}^{\ell} (A_k^u - A_{k-1}^u) + \sum_{k=1}^{\ell} (M_k^u - M_{k-1}^u). \tag{16}$$

where the previsible increments $(A_\ell^u - A_{\ell-1}^u)$ and martingale differences $(M_\ell^u - M_{\ell-1}^u)$ are defined as

$$A_\ell^u - A_{\ell-1}^u = \delta \left( -\frac{1}{1-\beta} \mathcal{A}_n + \frac{\delta}{(1-\beta)^2} \mathcal{L}_n \right) u_{\ell-1},$$

$$M_\ell^u - M_{\ell-1}^u = -\frac{\delta}{1-\beta} \langle \nabla H, \nabla u \rangle_{\ell-1}.$$

If we write $z_n$ as the corresponding continuous-time interpolated process of equation 16, then the difference $\|u_n - z_n\|$ is bounded by terms which vanish in $L^1$ uniformly in time. In particular, we have that this construction satisfies the requisite limit:

$$\sup_{0 \leq s \leq \tau_K^n} \|u_n(s) - z_n(s)\| \to 0 \quad \text{in } L^1.$$

The remainder of the proof closely follows (Ben Arous et al., 2024, Theorem 2.3); we provide the details for completeness. Let $z_n(s)$ be the linear interpolation of the discrete process $(z_{\lfloor s/\delta \rfloor})$ given in equation 16. We decompose this process as

$$z_n(s) = z_n(0) + a_n(s) + b_n(s),$$

where $a_n(s)$ and $b_n(s)$ are the continuous-time interpolations of the cumulative predictable and martingale parts, respectively. More precisely, we define for $s \in [0, T]$,

$$a_j'(s) = A_{[s/\delta]}^u - A_{[s/\delta]-1}^u \quad \text{and} \quad b_j'(s) = M_{[s/\delta]}^u - M_{[s/\delta]-1}^u$$

and write the $j$-th coordinate of $a_n(s)$ by

$$a_j(s) = \int_0^s a_j'(s') \, ds' = a_j(\delta[s/\delta]) + (s - \delta[s/\delta]) \left( A_{[s/\delta]}^u - A_{[s/\delta]-1}^u \right)$$

and similarly for $b_j(s) = \int_0^s b_j'(s') \, ds'$.

We now establish tightness for the sequence of stopped processes $(z_n(s \wedge \tau_K^n))_{s \in [0,T]}$ in $C([0,T])$ for each $K$. By Kolmogorov's continuity criterion, it suffices to verify the moment condition: for all $0 \leq s < t \leq T$,

$$\mathbb{E} \|z_n(t \wedge \tau_K^n) - z_n(s \wedge \tau_K^n)\|^4 \lesssim_{K,T} (t-s)^2.$$

This implies that the limit points are $(1/4)$-Hölder continuous. We control the drift and martingale components separately. Fix $j$ and let $u = u_j, a = a_j, b = b_j$ to simplify subsequent notation.

For the previsible (drift) term, we have

$$\mathbb{E}\left|a\left(t \wedge \tau_K^n\right) - a\left(s \wedge \tau_K^n\right)\right|^4 \lesssim \mathbb{E}\left|\sum_\ell \left(-\frac{\delta}{1-\beta}\mathcal{A}_n + \frac{\delta^2}{(1-\beta)^2}\mathcal{L}_n\right)u_{\ell-1}\right|^4$$

$$\lesssim \mathbb{E}\left|\delta\sum_\ell h_j(\beta, \boldsymbol{u}_n)_{\ell-1}\right|^4 + o((t-s)^4)$$

$$\lesssim (t-s)^4\left(\|h_j\|_{L^\infty_{K,n}}^4 + o(1)\right) \lesssim_K (t-s)^4.$$

The summation ranges over $\ell$ from $\lfloor s/\delta \rfloor + 1$ to $\lfloor t/\delta \rfloor$, stopped at $\tau_K^n/\delta$. The bound follows from the continuity of $\boldsymbol{h}$ as stipulated by asymptotic closability, i.e. Definition 2.2.

For the martingale term, applying the Burkholder-Davis-Gundy inequality yields

$$\mathbb{E}\left|b\left(t \wedge \tau_K^n\right) - b\left(s \wedge \tau_K^n\right)\right|^4 = \mathbb{E}\left[\left(\sum_\ell(M_\ell^u - M_{\ell-1}^u)\right)^4\right] \lesssim \mathbb{E}\left[\left(\sum_\ell(M_\ell^u - M_{\ell-1}^u)^2\right)^2\right]$$

Substituting the martingale increment and applying Cauchy-Schwarz yields,

$$\mathbb{E}\left[\left(\frac{\delta^2}{(1-\beta)^2}\sum_\ell \langle\nabla H_{\ell-1}, \nabla u_{\ell-1}\rangle^2\right)^2\right] \lesssim_\beta \delta^4 \sum_{\ell,\ell'}\mathbb{E}\left[\langle\nabla H_{\ell-1}, \nabla u_{\ell-1}\rangle^2 \langle\nabla H_{\ell'-1}, \nabla u_{\ell'-1}\rangle^2\right]$$

$$\leq \delta^4\left(\sum_\ell\left(\mathbb{E}\langle\nabla H_{\ell-1}, \nabla u_{\ell-1}\rangle^4\right)^{1/2}\right)^2 \lesssim_K (t-s)^2$$

where the final inequality follows by condition (3) of $\delta_n$-localizability.

Since both terms are $O\left((t-s)^2\right)$ for $0 \leq s, t \leq T$, we apply Kolmogorov's continuity theorem to conclude that the processes $(\boldsymbol{z}_n(s \wedge \tau_K^n))_s$ are uniformly $(1/4)$-Hölder continuous, thus establishing tightness. Consequently, the sequence of martingale components $(\boldsymbol{b}_n(t \wedge \tau_K^n))_t$ is also tight, and its limit points are continuous martingales.

We conclude the proof by showing that the limit points are solutions to the limiting SDE by the martingale problem. Let $\boldsymbol{z}_n^K(t) := \boldsymbol{z}_n(t \wedge \tau_K^n)$, and define $\boldsymbol{a}_n^K(t)$ and $\boldsymbol{b}_n^K(t)$ analogously. Denote their respective limits by $\boldsymbol{z}^K(t)$, $\boldsymbol{a}^K(t)$, and $\boldsymbol{b}^K(t)$.

We first compute the limiting quadratic variation of $\boldsymbol{b}^K(t)$, and thus $\boldsymbol{z}^K(t)$. Let $\Delta M_\ell^{z_i} := M_\ell^{z_i} - M_{\ell-1}^{z_i}$ and notice that for $1 \leq i, j \leq k$, the limiting predictable quadratic covariation is given by the integral

$$\int_0^t \delta\mathbb{E}\left[\Delta M_{[s/\delta]\wedge\tau_K}^{u_i}\Delta M_{[s/\delta]\wedge\tau_K}^{u_j}\right]\mathrm{d}s = \frac{1}{(1-\beta)^2}\int_0^t \delta\langle\nabla u_i, V\nabla u_j\rangle_{[s/\delta]\wedge\tau_K}\,\mathrm{d}s$$

By asymptotic-closability, we ensure that as $n \to \infty$,

$$\sup_{t\leq T}\left|\int_0^t \delta\langle\nabla u_i, V\nabla u_j\rangle_{[s/\delta]\wedge\tau_K} - \Sigma_{ij}\left(\mathbf{z}_n^K(s)\right)ds\right|$$

$$\leq T\sup_{x\in E_{K,n}^*}\left|\delta\langle\nabla u_i, V\nabla u_j\rangle(x) - \Sigma_{ij}\left(\mathbf{z}_n(x)\right)\right|$$

goes to zero as $n \to \infty$. Thus, the angle bracket of the limiting continuous martingale $\mathbf{b}^K(t)$ is by definition,

$$\left\langle\mathbf{b}^K\right\rangle_t = \int_0^t \boldsymbol{\Sigma}\left(\mathbf{z}^K(s)\right)ds$$

By Itô's formula for continuous semimartingales (Stroock & Varadhan, 2006, Theorem 6.3.4), for any $f \in C_0^\infty(\mathbb{R}^k)$, the process $f(\boldsymbol{z}_t^K) - \int_0^t \mathsf{L}f(\boldsymbol{z}_s^K)ds$ is a martingale, where $\mathsf{L}$ is the infinitesimal generator

$$\mathsf{L} = \frac{1}{2}\sum_{i,j=1}^k \Sigma_{ij}\partial_i\partial_j + \sum_{i=1}^k h_i\partial_i$$

Given the assumption that $\boldsymbol{h}$ and $\sqrt{\Sigma}$ are locally Lipschitz (and thus Lipschitz continuous on $E_K$), the martingale problem associated with L is well-posed. This uniquely characterizes the limit $\boldsymbol{z}^K$ as the solution to the SDE

$$d\boldsymbol{u}(t) = \boldsymbol{h}(\beta, \boldsymbol{u}(t))\, dt + \frac{1}{1-\beta}\sqrt{\Sigma(\boldsymbol{u}(t))}\, \mathrm{d}\boldsymbol{B}_t,$$

stopped at the exit time $\tau_K$. By employing a standard localization argument (see, e.g. (Stroock & Varadhan, 2006, Chapter 11)), we conclude that every limit point $\boldsymbol{z}(t)$ of $\boldsymbol{z}_n(t)$ is a solution to the limiting SDE. □

## B.1 Proof of Lemmas B.1-B.3

In this section, we prove the preparatory lemmas that characterize the behaviour of the momentum variable $\boldsymbol{p}_\ell$ as well as the summary statistic $\boldsymbol{u}_n$. In subsequent subsections we treat each of the lemmas dedicated to constructing the auxillary process $\boldsymbol{z}_n$ separately.

*Proof of Lemma B.1.* The momentum variable evolves according to the recurrence $\boldsymbol{p}_t = \beta \boldsymbol{p}_{t-1} - \delta \nabla L_{t-1}$. Iterating this relation from time $t$ down to time $s < t$ yields

$$\boldsymbol{p}_t = \beta^{t-s}\boldsymbol{p}_s - \delta \sum_{j=s}^{t-1} \beta^{t-1-j}\nabla L_j.$$

Reindexing the summation via $m = t - 1 - j$, we obtain the equivalent representation

$$\boldsymbol{p}_t = \beta^{t-s}\boldsymbol{p}_s - \delta \sum_{m=0}^{t-s-1} \beta^m \nabla L_{t-1-m}.$$

To recover the first claim, we set $t = \ell - 1 - i$ and $s = \ell - 1 - j$, which satisfies $t > s$ and $t - s = j - i$ for $0 \le i < j \le \ell - 1$. □

*Proof of Lemma B.2.* We expand the position variable recursively: $\boldsymbol{x}_{\ell-1} = \boldsymbol{x}_0 + \sum_{k=1}^{\ell-1} \boldsymbol{p}_k$. Utilizing the expression for $\boldsymbol{p}_k$ derived from Lemma B.1, we have

$$\boldsymbol{x}_{\ell-1} = \boldsymbol{x}_0 - \delta \sum_{k=1}^{\ell-1}\sum_{m=0}^{k-1} \beta^m \nabla L_{k-1-m}.$$

Interchanging the order of summation and evaluating the finite geometric series yields,

$$\boldsymbol{x}_{\ell-1} = \boldsymbol{x}_0 - \delta \sum_{m=0}^{\ell-2} \left(\sum_{j=0}^{m} \beta^j\right) \nabla L_{\ell-2-m} = \boldsymbol{x}_0 - \delta \sum_{m=0}^{\ell-2} \frac{1-\beta^{m+1}}{1-\beta}\nabla L_{\ell-2-m}.$$

To establish the relation between $\boldsymbol{x}_{\ell-1}$ and $\boldsymbol{x}_{\ell-1-i}$, we partition the summation:

$$\boldsymbol{x}_{\ell-1} = \boldsymbol{x}_0 - \delta \sum_{m=0}^{i-1} \frac{1-\beta^{m+1}}{1-\beta}\nabla L_{\ell-2-m} - \delta \sum_{m=i}^{\ell-2} \frac{1-\beta^{m+1}}{1-\beta}\nabla L_{\ell-2-m}. \tag{17}$$

We reindex the final summation by setting $k = m - i$:

$$\sum_{m=i}^{\ell-2} \frac{1-\beta^{m+1}}{1-\beta}\nabla L_{\ell-2-m} = \sum_{k=0}^{\ell-2-i} \frac{1-\beta^{k+i+1}}{1-\beta}\nabla L_{\ell-2-i-k}.$$

Decomposing the numerator as $1 - \beta^{k+i+1} = (1 - \beta^{k+1}) + (\beta^{k+1} - \beta^{k+i+1})$, we recognize that the first component, combined with $\boldsymbol{x}_0$, constitutes $\boldsymbol{x}_{\ell-1-i}$. Substituting this identity back into the expression for $\boldsymbol{x}_{\ell-1}$ and consolidating the remaining terms yields the desired result. □

*Proof of Lemma B.3.* By the integral form of the Mean Value Theorem, we have

$$\nabla^2 u(\boldsymbol{x}_{\ell-1}) - \nabla^2 u(\boldsymbol{x}_{\ell-1-i}) = \int_0^1 \nabla^3 u\left(t\boldsymbol{x}_{\ell-1} + (1-t)\boldsymbol{x}_{\ell-1-i}\right)[\boldsymbol{x}_{\ell-1} - \boldsymbol{x}_{\ell-1-i}]\,\mathrm{d}t.$$

Taking the operator norm (with respect to the tensor $\nabla^3 u$) and bounding the integral on the right-hand side, we obtain

$$\left\|\nabla^2 u(\boldsymbol{x}_{\ell-1}) - \nabla^2 u(\boldsymbol{x}_{\ell-1-i})\right\|_{\mathrm{op}} \leq \sup_{t\in[0,1]} \left\|\nabla^3 u\left(t\boldsymbol{x}_{\ell-1} + (1-t)\boldsymbol{x}_{\ell-1-i}\right)\right\|_{\mathrm{op}} \|\boldsymbol{x}_{\ell-1} - \boldsymbol{x}_{\ell-1-i}\|$$

$$\leq \sup_{x\in E_K^n} \|\nabla^3 u(x)\|_{\mathrm{op}} \|\boldsymbol{x}_{\ell-1} - \boldsymbol{x}_{\ell-1-i}\|$$

By $\delta_n$-localizability, $\|\nabla^3 u\|_{\mathrm{op}} \leq C_K$ and by Lemma B.2, the displacement vector $(\boldsymbol{x}_{\ell-1} - \boldsymbol{x}_{\ell-1-i})$ satisfies

$$\|\boldsymbol{x}_{\ell-1} - \boldsymbol{x}_{\ell-1-i}\|_{L_{K,n}^\infty} \lesssim_\beta \delta(\|\nabla\Phi\|_{L_{K,n}^\infty} + \|\nabla H\|_{L_{K,n}^\infty})\left(i + 2\sum_{m=0}^{l-2-i}\beta^{m+1}\right) \lesssim i\sqrt{\delta}$$

which is a quantity of order $O(i\sqrt{\delta})$ in $L^1$ uniformly in time. Consequently, combining both bounds gives the desired $L^1$ norm bound for the Hessian difference. $\square$

*Proof of Lemma B.4.* Take the first-order Taylor expansion of $\nabla u(\boldsymbol{x}_{\ell-1})$ around $\boldsymbol{x}_{\ell-1-i}$:

$$\nabla u(\boldsymbol{x}_{\ell-1}) = \nabla u(\boldsymbol{x}_{\ell-1-i}) + \nabla^2 u(\boldsymbol{x}_{\ell-1-i})[\boldsymbol{x}_{\ell-1} - \boldsymbol{x}_{\ell-1-i}] + R_1$$

where the remainder $R_1$ satisfies in $L^1$,

$$\|R_1\|_{L_{K,n}^\infty} \leq \sup_{x\in E_{K,n}^*} \|\nabla^3 u(x)\|_{\mathrm{op}} \|\boldsymbol{x}_{\ell-1} - \boldsymbol{x}_{\ell-1-i}\|^2 = O(i^2\delta)$$

using the bounds derived for the displacement vector in Lemma B.3 and $\delta$-localizability.

Substituting the expression for the displacement vector using Lemma B.2:

$$\nabla u(\boldsymbol{x}_{\ell-1}) = \nabla u(\boldsymbol{x}_{\ell-1-i}) - \delta\sum_{m=0}^{i-1}\frac{1-\beta^{m+1}}{1-\beta}\nabla^2 u(\boldsymbol{x}_{\ell-1-i})\nabla L_{\ell-2-m}$$

$$- \delta\frac{1+\beta^i}{1-\beta}\sum_{m=0}^{\ell-2-i}\beta^{m+1}\nabla^2 u(\boldsymbol{x}_{\ell-1-i})\nabla L_{\ell-2-i-m} + O(i^2\delta).$$

We proceed by aligning the arguments of the Hessian with those of the gradients with respect to $\boldsymbol{x}$. For the first summation, we use Lemma B.3 to bound the error introduced by realigning indices to get

$$\delta\sum_{m=0}^{i-1}\frac{1-\beta^{m+1}}{1-\beta}\nabla^2 u(\boldsymbol{x}_{\ell-1-i})\nabla L_{\ell-2-m}$$

$$= \delta\sum_{m=0}^{i-1}\frac{1-\beta^{m+1}}{1-\beta}\nabla^2 u(\boldsymbol{x}_{l-2-m})\nabla L_{l-2-m} + O\left(i\cdot\delta\cdot\|\nabla L\|_{L_{K,n}^\infty}\cdot i\sqrt{\delta}\right)$$

$$= \delta\sum_{m=0}^{i-1}\frac{1-\beta^{m+1}}{1-\beta}\nabla^2 u(\boldsymbol{x}_{l-2-m})\nabla H_{l-2-m} + O(i\sqrt{\delta} + i^2\delta)$$

where the additional error term in the last line follows by expanding $\nabla L = \nabla\Phi + \nabla H$ and bounding the series involving $\nabla\Phi$ by $\delta$-localizability. The remainder term can be further bounded as $O(i^2\sqrt{\delta})$ in $L^1$. A similar analysis applies to the second summation which we may write as

$$\sum_{m=0}^{\ell-2-i}\beta^{m+1}\nabla^2 u(\boldsymbol{x}_{\ell-1-i})\nabla L_{\ell-2-i-m} = \sum_{m=0}^{l-2-i}\beta^{m+1}\nabla^2 u(\boldsymbol{x}_{l-2-i-m})\nabla H_{l-2-i-m} + O(\delta^{-1/2})$$

Substituting the expansions and noting that the error terms are all dominated by $O(i^2\sqrt{\delta})$, completes the proof. $\square$

## B.2 PROOF OF LEMMA B.5

*Proof of Lemma B.5.* The proof proceeds by demonstrating that the extraneous second-order terms in the expansion of $u_j(\boldsymbol{x}_\ell)$, when accumulated over the trajectory, vanish asymptotically in $L^1$.

First, by $\delta_n$-localizability, the contribution at step $\ell$ for the deterministic quadratic term is bounded in $L^1$ by

$$\mathbb{E} \left| \frac{\delta^2}{2} \sum_{i,j=0}^{\ell-1} \beta^{i+j} \langle \nabla\Phi_{\ell-1-i} \otimes \nabla\Phi_{\ell-1-j}, \nabla^2 u_{\ell-1} \rangle \right| \lesssim_K \delta^{3/2}$$

Summing over $\ell = 1, \ldots, \lfloor t/\delta \rfloor$ yields a total contribution of $O(\sqrt{\delta}) = o(1)$.

For terms involving the fluctuations $\nabla H$, use Lemma B.3 to align the indices of the Hessian $\nabla^2 u$ with the fluctuation term. For example, we write the cross term as

$$\delta^2 \sum_{i,j=0}^{\ell-1} \beta^{i+j} \langle \nabla\Phi_{.-i} \otimes \nabla H_{.-j}, \nabla^2 u \rangle_{\ell-1} = \delta^2 \sum_{i,j=0}^{\ell-1} \beta^{i+j} \langle \nabla\Phi_{.-i} \otimes \nabla H_{.-j}, \nabla^2 u_{.-j} \rangle_{\ell-1} + R_\ell^{(1)}$$

The error $R_\ell^{(1)}$ arises from the Hessian difference over $j$ steps and is of the order

$$\|R_\ell^{(1)}\|_{L_{K,n}^\infty} = O\left( \delta^{5/2} \cdot \|\nabla\Phi\|_{L_{K,n}^\infty} \cdot \|\nabla H\|_{L_{K,n}^\infty} \right) = O(\delta^2)$$

uniformly in $L^1$. Similarly, for the centered quadratic fluctuation term:

$$\frac{\delta^2}{2} \sum_{i,j=0}^{\ell-1} \beta^{i+j} \langle \nabla H_{.-i} \otimes \nabla H_{.-j} - \mathbb{E}[\nabla H_{.-i} \otimes \nabla H_{.-j}], \nabla^2 u \rangle_{\ell-1}$$

$$= \frac{\delta^2}{2} \sum_{i,j=0}^{\ell-1} \beta^{i+j} \langle \nabla H_{.-i} \otimes \nabla H_{.-j} - \mathbb{E}[\nabla H_{.-i} \otimes \nabla H_{.-j}], \nabla^2 u_{.-i} \rangle_{\ell-1} + o(\delta) \qquad (18)$$

where the error follows by item (3) of $\delta$-localizability. Accumulating the errors over the trajectory yields a $o(1)$-remainder.

It remains to demonstrate that the accumulated aligned stochastic series vanish in $L^1$. We show this by proving the stronger condition that these terms vanish in $L^2$. We first introduce the notation:

$$\mathsf{P}_{ij} = \langle \nabla\Phi_i \otimes \nabla H_j, \nabla^2 u_j \rangle, \quad \mathsf{H}_{ij} = \langle \nabla H_i \otimes \nabla H_j - \mathbb{E}[\nabla H_i \otimes \nabla H_j], \nabla^2 u_i \rangle$$

Now consider equation 18 which is further accumulated over $\ell = 1, \ldots \lfloor t/\delta \rfloor$ steps and set $T = \lfloor t/\delta \rfloor$. In particular, we want to bound the term

$$S_T^P := \delta^2 \sum_{\ell=1}^T \sum_{i,j=0}^{\ell-1} \beta^{i+j} \mathsf{P}_{\ell-1-i, \ell-1-j} = \delta^2 \sum_{k,m=0}^{T-1} \mathsf{P}_{k,m} \left( \sum_{\ell=\max(k,m)+1}^{T} \beta^{2(\ell-1)-k-m} \right)$$

where the last equality follows by reindexing with $k = \ell - 1 - i$ and $m = \ell - 1 - j$ and interchanging the order of summation. The inner geometric series is uniformly bounded by

$$\sum_{\ell=\max(k,m)+1}^{T} \beta^{2(\ell-1)-k-m} \leq \frac{\beta^{2\max(k,m)-k-m}}{1-\beta^2} = \frac{\beta^{|k-m|}}{1-\beta^2}.$$

By Lemma B.8, it follows that the $L^2$-norm vanishes: $\mathbb{E}[(S_T^P)^2] = o(1)$, uniformly in $t$. An identical argument applies to the series involving $\mathsf{H}_{k,m}$ since we can write

$$\delta^2 \sum_{\ell=1}^{\lfloor t/\delta \rfloor} \sum_{i,j=0}^{\ell-1} \beta^{i+j} \mathsf{H}_{\ell-1-i, \ell-1-j} = \delta^2 \sum_{i,j=0}^{\lfloor t/\delta \rfloor - 1} \mathsf{H}_{i,j} \left( \sum_{\ell=\max(i,j)+1}^{\lfloor t/\delta \rfloor} \beta^{2\ell-i-j} \right)$$

thus completing the proof. $\qquad\square$

## B.3 PROOF OF LEMMA B.6

*Proof.* We first analyze the inner series by applying the gradient expansion provided by Lemma B.4 to $\nabla u_{\ell-1}$, centered at $\ell - 1 - i$.

$$\delta \sum_{i=1}^{\ell-1} \beta^i \langle \nabla H_{\cdot-i}, \nabla u \rangle_{\ell-1} = \delta \sum_{i=1}^{\ell-1} \beta^i \langle \nabla H, \nabla u \rangle_{\ell-1-i} + S_\ell^{(1)} + S_\ell^{(2)} + O(\delta^{3/2})$$

where the terms $S_\ell^{(1)}$ and $S_\ell^{(2)}$ are the second-order components:

$$S_\ell^{(1)} = -\delta^2 \sum_{i=1}^{\ell-1} \beta^i \sum_{m=0}^{i-1} \frac{1-\beta^{m+1}}{1-\beta} \langle \nabla H_{\cdot-i+1} \otimes \nabla H_{\cdot-m}, \nabla^2 u_{\cdot-m} \rangle_{\ell-2},$$

$$S_\ell^{(2)} = -\delta^2 \sum_{i=1}^{\ell-1} \beta^i \frac{1+\beta^i}{1-\beta} \sum_{m=0}^{\ell-2-i} \beta^{m+1} \langle \nabla H_{\cdot+1} \otimes \nabla H_{\cdot-m}, \nabla^2 u_{\cdot-m} \rangle_{\ell-2-i}.$$

We decompose these terms into centered fluctuations and deterministic correctors. In $S_\ell^{(1)}$, the diagonal contribution occurs when $\ell - 1 - i = \ell - 2 - m$, i.e., $m = i - 1$. For off-diagonal terms, we note that $\mathbb{E}[\nabla H_i \otimes \nabla H_j]$ for $i \neq j$. This yields the decomposition:

$$S_\ell^{(1)} = -\delta^2 \sum_{i=1}^{\ell-1} \beta^i \frac{1-\beta^i}{1-\beta} \langle V, \nabla^2 u \rangle_{\ell-1-i}$$

$$- \delta^2 \sum_{i=1}^{\ell-1} \beta^i \sum_{m=0}^{i-1} \frac{1-\beta^{m+1}}{1-\beta} \langle \nabla H_{\cdot-i+1} \otimes \nabla H_{\cdot-m} - \mathbb{E}[\nabla H_{\cdot-i+1} \otimes \nabla H_{\cdot-m}], \nabla^2 u_{\cdot-m} \rangle_{\ell-2}$$

From here, all of the remaining terms constitute fluctuations, which we show vanish in $L^2$ after accumulating over $\ell = 1, \ldots, T = \lfloor t/\delta \rfloor$ as in Lemma B.5. Adopting the same notation $H_{n,m}$, we reindex with $n = \ell - 1 - i$, $m = \ell - 2 - j$, and reorganize the first summation to obtain

$$\sum_{\ell=1}^{[t/\delta]} \sum_{i=1}^{\ell-1} \sum_{j=0}^{i-1} \beta^i \frac{1-\beta^{j+1}}{1-\beta} H_{\ell-1-i,\ell-2-j} = \sum_{n=0}^{[t/\delta]-2} \sum_{m=n}^{[t/\delta]-2} H_{n,m} \left( \sum_{\ell=m+1}^{[t/\delta]-1} \beta^{\ell-1-n} \frac{1-\beta^{\ell-1-m}}{1-\beta} \right)$$

For $m \geq n$, the inner geometric series is bounded by

$$\sum_{\ell=m+1}^{[t/\delta]-1} \beta^{\ell-1-n} \frac{1-\beta^{\ell-1-m}}{1-\beta} \leq \frac{\beta^{m-n}}{(1-\beta)^2} - \frac{\beta^{m-n+1}}{(1-\beta)(1-\beta^2)} \leq C_\beta \beta^{|m-n|}$$

where $C_\beta > 0$ is some constant that only depends on $\beta$. Setting $w(n, m) = 0$ for $m < n$, we apply Lemma B.8 to conclude that the $L^2$-norm vanishes as $\delta \to 0$. An identical argument applies to $S_\ell^{(2)}$ as we can rewrite the series as

$$\sum_{\ell=1}^{[t/\delta]} \sum_{i=1}^{\ell-1} \beta^{i+1} \frac{1+\beta^i}{1-\beta} \sum_{j=0}^{\ell-2-i} \beta^j H_{\ell-1-i,\ell-2-j-i}$$

$$= \sum_{m=0}^{[t/\delta]-2} \sum_{n=m+1}^{[t/\delta]-2} H_{n,m} \left( \sum_{\ell=n+2}^{[t/\delta]} \beta^{\ell-1-m} \frac{1+\beta^{\ell-1-n}}{1-\beta} \right)$$

To conclude the proof, we rewrite the leading first-order term by first reindexing the summation by $j = \ell - 1 - i$,

$$\delta \sum_{\ell=1}^{\lfloor t/\delta \rfloor} \sum_{i=1}^{\ell-1} \beta^i \langle \nabla H, \nabla u \rangle_{\ell-1-i} = \delta \sum_{j=0}^{\lfloor t/\delta \rfloor-1} \frac{1-\beta^{\lfloor t/\delta \rfloor-j}}{1-\beta} \langle \nabla H, \nabla u \rangle_j$$

$$= \frac{\delta}{1-\beta} \sum_{\ell=1}^{\lfloor t/\delta \rfloor} \langle \nabla H, \nabla u \rangle_{\ell-1} + O(\delta^{1/2})$$

Putting all the terms together shows that the series converges to a rescaled martingale sum as desired. $\square$

## B.4 PROOF OF LEMMA B.7

*Proof.* We analyze the asymptotic behavior of the first-order (drift) and second-order (diffusion) components separately. Let $T = \lfloor t/\delta \rfloor$. Starting with the second-order drift, we align the Hessian with the covariance matrix by Lemma B.3 to write

$$
\frac{\delta^2}{2} \sum_{\ell=1}^{\lfloor t/\delta \rfloor} \sum_{i=0}^{\ell-1} \left[ \beta^{2i} + 2\beta^i \frac{1-\beta^i}{1-\beta} \right] \langle V_{\ell-1-i}, \nabla^2 u_{\ell-1} \rangle
$$

$$
= \frac{\delta^2}{2} \sum_{\ell=1}^{\lfloor t/\delta \rfloor} \sum_{i=0}^{\ell-1} \left[ \beta^{2i} + 2\beta^i \frac{1-\beta^i}{1-\beta} \right] \langle V, \nabla^2 u \rangle_{\ell-1-i} + O\left( \delta^{5/2} \|\nabla H\|_{L^\infty_{K,n}}^2 (t/\delta) \right)
$$

$$
= \frac{\delta^2}{2} \sum_{j=0}^{\lfloor t/\delta \rfloor - 1} \left[ \sum_{i=0}^{\lfloor t/\delta \rfloor - 1 - j} \frac{(1-\beta^{i+1})^2 - (1-\beta^i)^2}{(1-\beta)^2} \right] \langle V, \nabla^2 u \rangle_j + o(1)
$$

$$
= \frac{\delta^2}{2(1-\beta)^2} \sum_{\ell=1}^{\lfloor t/\delta \rfloor} \langle V, \nabla^2 u \rangle_{\ell-1} + o(1)
$$

where the simplification in the $\beta$-weights follows since

$$
(1-\beta^{i+1})^2 - (1-\beta^i)^2 = 2\beta^i(1-\beta) - \beta^{2i}(1-\beta^2) = (1-\beta)\left[ 2\beta^i - \beta^{2i}(1+\beta) \right]
$$

For the first-order (drift) component, the analysis mirrors that of Lemma B.6. As a reminder, we apply Lemma B.4 to expand $\nabla u_{\ell-1}$ around $\nabla u_{\ell-1-i}$ yields

$$
\delta \sum_{i=1}^{\ell-1} \beta^i \langle \nabla \Phi_{\ell-1-i}, \nabla u_{\ell-1} \rangle
$$

$$
= \delta \sum_{i=1}^{\ell-1} \beta^i \langle \nabla \Phi, \nabla u \rangle_{\ell-1-i} - \delta^2 \sum_{i=1}^{\ell-1} \beta^i \sum_{m=0}^{i-1} \frac{1-\beta^{m+1}}{1-\beta} \mathsf{P}_{\ell-1-i,\ell-2-m}
$$

$$
- \delta^2 \sum_{i=1}^{\ell-1} \beta^i \frac{1+\beta^i}{1-\beta} \sum_{m=0}^{\ell-2-i} \beta^{m+1} \mathsf{P}_{\ell-1-i,\ell-2-m-i} + O(\delta^{3/2})
$$

where the two series involving $\mathsf{P}_{i,j}$ vanish in $L^2$ by Lemma B.8, following identical arguments as in Lemma B.6.

It remains to analyze the leading first-order term which mirrors how we handled the first-order martingale sum in Lemma B.6. Using the same arguments, we find that

$$
\delta \sum_{\ell=1}^{T} \sum_{i=0}^{\ell-1} \beta^i \langle \nabla \Phi, \nabla u \rangle_{\ell-1-i} = \frac{\delta}{1-\beta} \sum_{\ell=1}^{T} \langle \nabla \Phi, \nabla u \rangle_{\ell-1} + O(\delta^{1/2}).
$$

which completes the proof. □

## B.5 AUXILIARY $L^2$-BOUND

We establish a technical lemma essential for controlling the accumulation of second-order stochastic errors in Lemma B.8, B.6, B.7. We recall the following random variables representing the stochastic interactions:

$$
\mathsf{P}_{ij} := \langle \nabla \Phi_i \otimes \nabla H_j, \nabla^2 u_j \rangle, \quad \mathsf{H}_{ij} := \langle \nabla H_i \otimes \nabla H_j - \mathbb{E}[\nabla H_i \otimes \nabla H_j], \nabla^2 u_i \rangle.
$$

We observe the following orthogonality properties derived from the martingale difference structure of $\nabla H$ with respect to the filtration generated by the process.

- If $j \neq l$, then $\mathbb{E}[\mathsf{P}_{ij}\mathsf{P}_{kl}] = 0$ by towering the expectation with $X_{j \vee l}$. Furthermore, by $\delta$-localizability we have the correlation bound

$$
\mathbb{E}[\mathsf{P}_{ij}\mathsf{P}_{kl}] \leq \sup_{E^*_{K,n}} \|\nabla \Phi\|^2 \|\nabla^2 u\|_{\mathrm{op}}^2 \mathbb{E}[\|\nabla H\|^2] \lesssim_K \delta^{-1} \left( \delta^{-2} \right)^{1/2} = O(\delta^{-2})
$$

- If $\max\{i,j\} \neq \max\{k,l\}$, then $\mathbb{E}[\mathsf{H}_{ij}\mathsf{H}_{kl}] = 0$ by towering the expectation with $X_{i \vee j \vee k \vee l}$. Furthermore, by $\delta$-localizability we have the correlation bound

$$\mathbb{E}[\mathsf{H}_{ij}\mathsf{H}_{kl}] \leq \left(\mathbb{E}[\mathsf{H}_{ij}^2]\mathbb{E}[\mathsf{H}_{kl}]\right)^{1/2} \leq \sup_{x,y \in E_{K,n}^*} \langle \nabla H(x) \otimes \nabla H(y), \nabla^2 u(y) \rangle^2 \leq o(\delta^{-3})$$

With this observation, we construct the following lemma that controls the correlation bound over an accumulated memory.

**Lemma B.8.** Let $\{\mathsf{X}\}_{i,j}$ be a collection of r.v.s where we take either $\mathsf{X}_{i,j} = \mathsf{P}_{i,j}$ or $\mathsf{X}_{i,j} = \mathsf{H}_{i,j}$ as defined above. Then for the series constructed by

$$S_\delta = \delta^2 \sum_{i,j=0}^{N} w(i,j) \mathsf{X}_{i,j}$$

where $w(i,j)$ are deterministic weights such that $|w(i,j)| \leq C_\beta \beta^{|i-j|}$ for any $i,j$ with $C_\beta > 0$ an absolute constant that only depends on $\beta$, then the $L^2$ norm of $S_\delta$ is bounded by

$$\mathbb{E}[S_\delta^2] \lesssim_{\beta,K} o(\delta N)$$

In particular, if we take $N = O(\delta^{-1})$ then $\mathbb{E}[S_\delta^2] = o(1)$ as $\delta \to 0$ when we take $\mathsf{X}$ as either $\mathsf{P}$ or $\mathsf{H}$.

*Proof.* We first note that the decay condition on the weights $w(i,j)$ imply a uniform bound over single-index sums. That is,

$$\sup_j \sum_{i=0}^{N} |w(i,j)| \leq \sup_j \sum_{i=-\infty}^{\infty} C_\beta \beta^{|i-j|} \leq C_\beta \frac{1+\beta}{1-\beta}$$

and similarly for the uniform bound over index $i$. We now determine the $L^2$-norm,

$$\mathbb{E}[S_\delta^2] = \delta^4 \sum_{i,j,k,l} w(i,j)w(k,l)\mathbb{E}[\mathsf{X}_{i,j}\mathsf{X}_{k,l}]$$

If we set $\mathsf{X} = \mathsf{P}$, then the summation is restricted to $j = l$ which yields the bound

$$\mathbb{E}[S_\delta^2] = \delta^4 \sum_{i,j,k,l} w(i,j)w(k,l)\mathbb{E}[\mathsf{P}_{i,j}\mathsf{P}_{k,l}] \leq \delta^4 \sup_{E_{K,n}^*} |\mathbb{E}[\mathsf{P}_{ij}\mathsf{P}_{kl}]| \sum_j \left(\sum_i |w(i,j)|\right)^2 \tag{19}$$

On the other hand, for $\mathsf{X} = \mathsf{H}$ the summation is restricted to $\max(i,j) = \max(k,l) = m$ so that,

$$\mathbb{E}[S_\delta^2] = \delta^4 \sum_m \sum_{\substack{\max(i,j)=m \\ \max(k,l)=m}} w(i,j)w(k,l)\mathbb{E}[\mathsf{H}_{i,j}\mathsf{H}_{k,l}]$$

$$\leq \delta^4 \sup_{E_{K,n}^*} |\mathbb{E}[\mathsf{H}_{ij}\mathsf{H}_{kl}]| \sum_m \left(\sum_{\max(i,j)=m} |w(i,j)|\right)^2 \tag{20}$$

where the inner series involving the geometric coefficients is bounded by

$$\sum_{\max(i,j)=m} |w(i,j)| = w(m,m) + \sum_{i=0}^{m-1} |w(i,m)| + \sum_{j=0}^{m-1} |w(i,j)| \leq C_\beta + \frac{2C_\beta}{1-\beta}$$

which is an absolute constant that only depends on $\beta$. In either case, we see that both Equation equation 19 and equation 20 are uniformly bounded by

$$\mathbb{E}[S_\delta^2] \lesssim_\beta \delta^4 o(\delta^{-3})N \leq o(\delta N)$$

where the $o(\delta^{-3})$ error is attained from the previous correlation bounds. Thus the $L^2$-norm vanishes for $N \leq O(\delta^{-1})$ steps as desired. $\qquad\square$

## C    PROOFS FOR TENSOR PCA

### C.1    PROOF OF PROPOSITION 3.1

The first part of the proof pertaining to SGD-M follows from Theorem 2.3 and Section 7 of Ben Arous et al. (2024) along with the verification of the final item of (3) in $\delta_n$-localizability (which is verified by the same bounds as the previous item). For example,

$$\sup_{x \in E_{K,n}^*} \mathbb{E}_{y_1 \perp y_2} \left[ \langle \nabla^2 u(x), \nabla H(x, y_1) \otimes \nabla H(x, y_2) \rangle^2 \right] \lesssim \sup_{x \in E_{K,n}^*} \mathbb{E} \|\nabla H(x)\|^4 \lesssim n^2$$

for $u = (m, r^2)$. For our diffusive limits about $m = 0$, we have $\nabla^2 u = 0$ so that the bound is trivial. Moving on to SGD-U, we split the proof into two parts. The proof follows from the preconditioning theorem provided in appendix A.3. Thus we simply must verify asymptotic closability and $(\delta_n, \eta_n)$-localizability. We prove these two assumptions separately below starting with asymptotic closability.

#### C.1.1    PROOF OF ASYMPTOTIC CLOSABILITY

We check asymptotic closability starting with the pre-limits for $\nabla \tilde{\Phi}$ and $\tilde{V}$ under the normalized gradient pre-conditioner. Recall that from (Ben Arous et al., 2024, Section 7), $\nabla L$ is a Gaussian vector with mean $\nabla \Phi$ and covariance $V$ given by

$$\nabla \Phi = -2\lambda k m^{k-1} v + 2k R^{2k-2} m v + 2k R^{2k-2}(x - mv) = -2\lambda k m^{k-1} v + 2k R^{2k-2} x$$

$$V = 4k R^{2k-2} \boldsymbol{I} + 4k(k-1) R^{2k-4} x x^\top$$

prior to the action of the preconditioner. Furthermore, the distribution of $\nabla L$ only depends on the summary statistics $(m, r^2)$.

Before proceeding, we compute the expected norm of the gradient vector,

$$\mathbb{E}[\|\nabla L\|^2] = \|\nabla \Phi\|^2 + \mathbb{E}[\|\nabla H\|^2] + 2\mathbb{E}[\langle \nabla \Phi, \nabla H \rangle] = \|\nabla \Phi\|^2 + \operatorname{tr}(V)$$

Substituting the values yields

$$D^2 = \mathbb{E}[\|\nabla L\|^2] = \|\nabla \Phi\|^2 + \operatorname{tr}(V)$$

$$= \left( 4\lambda^2 k^2 m^{2k-2} + 4k^2 R^{4k-2} - 4\lambda k^2 m^k R^{2k-2} \right) + \left( 4nk R^{2k-2} + 4k(k-1) R^{2k-2} \right)$$

$$= 4k^2 (\lambda^2 m^{2k-2} + R^{4k-2} - \lambda m^k R^{2k-2}) + 4(n + k - 1) k R^{2k-2}$$

We compute the first and second-order generators separately.

**Lemma C.1.** In the asymptotic limit where $n \to \infty$ and $\nabla L$ is defined as above, we have the asymptotic expansion for the first-order drift for SGD-U,

$$\nabla \tilde{\Phi} = \sqrt{n} \, \mathbb{E} \left[ \frac{\nabla L}{\|\nabla L\|} \right] = \frac{\sqrt{n} \, \nabla \Phi}{D} + O(n^{-1})$$

*Proof.* For $h(x) = x^{-1/2}$, we take the Taylor series centered at $x = D^2$

$$h(x) = h(D^2) + h'(D^2)(x - D^2) + \frac{1}{2} h''(D^2)(x - D^2)^2 + \frac{1}{6} h'''(\xi)(x - D^2)^3$$

$$= \frac{1}{D} - \frac{1}{2D^3}(x - D^2) + \frac{3}{8D^5}(x - D^2)^2 - \frac{15}{64\xi^7}(x - D^2)^3$$

where $\xi \in (x, D^2)$ attains the remainder in Lagrange form. Using $h(\|\nabla L\|^2)$ we write

$$\mathbb{E} \left[ \frac{\nabla L}{\|\nabla L\|} \right] = \mathbb{E}[h(\|\nabla L\|^2) \nabla L]$$

$$= \underbrace{\frac{\nabla \Phi}{D} - \frac{1}{2D^3} \mathbb{E}[\nabla L(\|\nabla L\|^2 - D^2)] + \frac{3}{8D^5} \mathbb{E}[\nabla L(\|\nabla L\|^2 - D^2)^2]}_{T(\|\nabla L\|^2)}$$

$$\underbrace{-\frac{15}{64\xi^7}\mathbb{E}[\nabla L(\|\nabla L\|^2 - D^2)^3]}_{R(\|\nabla L\|^2)}$$

It follows that the remainder term satisfies,

$$\left\|\frac{15}{64\xi^7}\mathbb{E}[\nabla L(\|\nabla L\|^2 - D^2)^3]\right\| \lesssim \mathbb{E}\left[\left|1 - \frac{D^2}{\|\nabla L\|^2}\right|^3\right]$$

which is $O(n^{-2})$ by Lemma E.4. To conclude, we analyze the remaining polynomial

$$\mathbb{E}[T(\|\nabla L\|^2)\nabla L] = \frac{\nabla\Phi}{D} - \frac{1}{2D^3}\mathbb{E}[\nabla L(\|\nabla L\|^2 - D^2)] + \frac{3}{8D^5}\mathbb{E}[\nabla L(\|\nabla L\|^2 - D^2)^2]$$

To do so, we use Stein's lemma: $\mathbb{E}[(Z - \mu)F(Z)] = \Sigma\mathbb{E}[\nabla F(Z)]$ for a Gaussian vector $Z \sim \mathcal{N}(\mu, \Sigma)$. We bound each term individually by considering the respective numerators.

- For $\mathbb{E}[\nabla L(\|\nabla L\|^2 - D^2)]$, set $f(z) = \|z\|^2$ and center the variable:

$$\mathbb{E}[\nabla L(\|\nabla L\|^2 - D^2)] = \mathbb{E}[(\nabla L - \nabla\Phi)(\|\nabla L\|^2 - D^2)] + \nabla\Phi\mathbb{E}[\|\nabla L\|^2 - D^2]$$

$$= V\mathbb{E}[2\nabla L] = 2V\nabla\Phi$$

  which is of the normed order (with respect to the Frobenius norm) $\|V \ \nabla\Phi\| \leq \|V\|_{\mathrm{op}}\|\nabla\Phi\| = O(1)$.

- For $\mathbb{E}[\nabla L(\|\nabla L\|^2 - D^2)^2]$, use $f(z) = (\|z\|^2 - c)^2$ so $\nabla f = 4(\|z\|^2 - c)z$.

$$\mathbb{E}[\nabla L(\|\nabla L\|^2 - D^2)^2] = \mathbb{E}[(\nabla L - \nabla\Phi)(\|\nabla L\|^2 - D^2)^2] + \nabla\Phi\mathbb{E}[(\|\nabla L\|^2 - D^2)^2]$$

$$= V\mathbb{E}[4(\|\nabla L\|^2 - D^2)\nabla L] + \nabla\Phi\mathrm{Var}(\|\nabla L\|^2)$$

$$= 8V^2\nabla\Phi + \nabla\Phi\mathrm{Var}(\|\nabla L\|^2) \lesssim_K n$$

  with respect to the Frobenius norm since $\mathrm{Var}(\|\nabla L\|^2) \sim n$.

Putting the full expansion together, we conclude that

$$\mathbb{E}\left[\frac{\nabla L}{\|\nabla L\|}\right] = \frac{\nabla\Phi}{D} - \frac{1}{2D^3}O(1) + O(d^{-2}) = \frac{\nabla\Phi}{D} + O(n^{-3/2})$$

as desired. $\qquad\square$

**Lemma C.2.** In the asymptotic limit where $n \to \infty$ and $\nabla L$ is defined as above, we have the asymptotic expansion for the second-order drift for SGD-U,

$$\tilde{V} = \frac{\sqrt{n} V}{D^2} + O(n^{-1})$$

*Proof.* The arguments for the proof follow those of Lemma C.1. First, we write

$$\tilde{V} = \mathbb{E}\left[\nabla\tilde{H} \otimes \nabla\tilde{H}\right] = \mathbb{E}\left[\nabla\tilde{L} \otimes \nabla\tilde{L}\right] - \nabla\tilde{\Phi} \otimes \nabla\tilde{\Phi}$$

$$= \mathbb{E}\left[\frac{\nabla L \otimes \nabla L}{\|\nabla L\|^2}\right] - \frac{\nabla\Phi \otimes \nabla\Phi}{D^2} + O\left(n^{-2}\right)$$

where the last term follows by the trailing $O(n^{-3/2})$ error for $\nabla\tilde{\Phi}$ acting on $\|\nabla\Phi/D\| = O(n^{-1/2})$. Subtracting both sides by the quantity $V/D^2$,

$$\tilde{V} - \frac{V}{D^2} = \mathbb{E}\left[\frac{\nabla L \otimes \nabla L}{\|\nabla L\|^2}\right] - \frac{\nabla\Phi \otimes \nabla\Phi}{D^2} - \left(\frac{\mathbb{E}[\nabla L \otimes \nabla L] - \nabla\Phi \otimes \nabla\Phi}{D^2}\right) + O(n^{-2})$$

$$= \mathbb{E}\left[\frac{\nabla L \otimes \nabla L}{\|\nabla L\|^2}\right] - \frac{\mathbb{E}[\nabla L \otimes \nabla L]}{D^2} + O(n^{-2})$$

using the expansion $V = \mathbb{E}[\nabla L \otimes \nabla L] - \nabla \Phi \otimes \nabla \Phi$. To control the leading term, we conduct a similar analysis as before using the expansion of the function $h(y) = y^{-1}$ around $y = D^2$ which yields,

$$h(y) = \frac{1}{D^2} - \frac{1}{D^4}(y - D^2) + \frac{1}{D^6}(y - D^2)^2 - \frac{1}{\xi^4}(y - D^2)^3$$

for some $\xi \in [y, D^2]$ to express the remainder in Lagrangian form. Substituting this expansion back to the leading term,

$$\mathbb{E}\left[\frac{\nabla L \otimes \nabla L}{\|\nabla L\|^2}\right] - \frac{\mathbb{E}[\nabla L \otimes \nabla L]}{D^2}$$

$$= \mathbb{E}\left[\nabla L \otimes \nabla L \cdot h(\|\nabla L\|^2)\right] - \frac{\mathbb{E}[\nabla L \otimes \nabla L]}{D^2}$$

$$= -\frac{\mathbb{E}[\nabla L \otimes \nabla L \cdot (\|\nabla L\|^2 - D^2)]}{D^4} + \frac{\mathbb{E}[\nabla L \otimes \nabla L \cdot (\|\nabla L\|^2 - D^2)^2]}{D^6} + R(\xi, \nabla L)$$

For the trailing remainder function $R$, one can show that

$$\|R\| \lesssim \mathbb{E}\left[\left|1 - \frac{D^2}{\|\nabla L\|^2}\right|^3\right]$$

which is $O(n^{-2})$ again by Lemma E.4. For the leading two terms, we use Stein's lemma for vector-valued functions (e.g. see (Liu et al., 2016, Lemma 2.2)),

$$\mathbb{E}[Xf(X)^\top] = \mu\mathbb{E}[f(X)^\top] + V\mathbb{E}[\nabla f(X)]$$

for a Gaussian vector $X \sim \mathcal{N}(\mu, V)$ and $\nabla f(X)$ is the Jacobian of the vector-valued function $f$.

- For $\mathbb{E}[\nabla L \otimes \nabla L \cdot (\|\nabla L\|^2 - D^2)]$, apply Stein's Lemma to write

$$\mathbb{E}[\nabla L \otimes \nabla L \cdot (\|\nabla L\|^2 - D^2)] = \nabla \Phi \otimes \mathbb{E}[\nabla L(\|\nabla L\|^2 - D^2)] + V\mathbb{E}[(\|\nabla L\|^2 - D^2)]$$
$$+ 2V\mathbb{E}[\nabla L \otimes \nabla L]$$

$$= 2V^2 - 2(V \nabla \Phi) \otimes \nabla \Phi + O(1) = 2V^2 + O(1)$$

where we obtained the quantity $\mathbb{E}[\nabla L(\|\nabla L\|^2 - D^2)]$ in the previous derivation as $O(1)$ and the other $O(1)$ term is of the same order by symmetry. We note that the entire quantity is of the Frobenius norm order $\|V^2\| = O(\sqrt{n})$ following the spectrum of $V$.

- For $\mathbb{E}[\nabla L \otimes \nabla L \cdot (\|\nabla L\|^2 - D^2)^2]$, we again apply Stein's lemma to write

$$\mathbb{E}[\nabla L \otimes \nabla L \cdot (\|\nabla L\|^2 - D^2)^2] = \nabla \Phi \otimes \mathbb{E}[\nabla L(\|\nabla L\|^2 - D^2)^2] + V\mathbb{E}[(\|\nabla L\|^2 - D^2)^2]$$
$$+ 8V\mathbb{E}[\nabla L \otimes \nabla L \cdot (\|\nabla L\|^2 - D^2)]$$

$$= V \cdot \text{Var}(\|\nabla L\|^2) + 8V \cdot O(\sqrt{n}) + O(n)$$

$$= V \cdot \text{Var}(\|\nabla L\|^2) + O(n)$$

where in the penultimate line we substitute the quantities from the previous step (and derivation of $\nabla \tilde{\Phi}$). We again note that the entire quantity is of the order $O(n^{3/2})$ with respect to the Frobenius norm on the variance quantity and spectrum of $V$.

Putting the full expansion together, we conclude that

$$\tilde{V} = \frac{V}{D^2} - \frac{1}{D^4} \cdot O(n^{1/2}) + \frac{1}{D^6} \cdot O(n^{3/2}) + O(n^{-2}) = \frac{V}{D^2} + O(n^{-3/2}) \qquad \square$$

### C.1.2 Proof of $(\delta_n, \eta_n)$- localizability

*Proof.* We verify $(\delta_n, \eta_n)$-localizability for tensor pca with $\tilde{u}(x) = \sqrt{n}(m(x), r_\perp^2(x))$. We note that verification will then follow immediately for the cases $u(x) = (\sqrt{n}m(x), r_\perp^2(x))$ and $u(x) = (m(x), r_\perp^2(x))$ or any centering of these statistics. We also note item $(1)$ of $(\delta_n, \eta_n)$-localizability has already been verified for the no-preconditioning case (e.g. see (Ben Arous et al., 2024, Section 7)). We move on to $(2)$:

$$\sup_{x \in u^{-1}(E_k)} \|\nabla\tilde{\Phi}\| = \sup_{x \in u^{-1}(E_k)} \sqrt{n}\frac{\|\nabla\Phi(x)\|}{\sqrt{\mathbb{E}\|\nabla L(x,y)\|^2}} + O(1/\sqrt{n})$$

$$= \sup_{x \in u^{-1}(E_k)} \sqrt{n}\frac{\|\nabla\Phi(x)\|}{\sqrt{n\|x\|^{2(k-1)} + \|\Phi(x)\|^2}} + O(1/\sqrt{n})$$

We now note the following bound on $\|\Phi(x)\|$:

$$\|\nabla\Phi(x)\| = \|[-2\lambda km^{k-1} + 2k\|x\|^{2k-2}m]v + k\|x\|^{2k-2}(2x - mv)\|$$

$$\lesssim [\|x\|^{2k-2} + \|x\|^{k-1}]$$

Applying this to the above we get:

$$\sup_{x \in u^{-1}(E_k)} \|\nabla\tilde{\Phi}\| \lesssim \sup_{x \in u^{-1}(E_k)} \sqrt{n}\left[\frac{C_k[\|x\|^{2k-2} + \|x\|^{k-1}]}{\sqrt{n\|x\|^{2k-2} + \|\Phi(x)\|^2}} + O\left(\frac{1}{\sqrt{n}}\right)\right] \leq C_k$$

We now turn to the martingale component:

$$\sup_{x \in u^{-1}(E_k)} \mathbb{E}\|\nabla\tilde{H}(x,y)\|^8 \lesssim n^4 \sup_{x \in u^{-1}(E_k)} \mathbb{E}\left[\left\|\frac{\nabla L(x,y)}{\|\nabla L(x,y)\|}\right\|^8 + \|\nabla\tilde{\Phi}(x)]\|^8 + O(n^{-4})\right]$$

$$\leq C_k n^4$$

We now turn to item $(3)$ of $\delta_n$-localizablity:

$$\sup_{x \in u^{-1}(E_k)} \mathbb{E}\langle\nabla\tilde{H}(x,y), \nabla\tilde{u}_i(x)\rangle^4 = \sup_{x \in u^{-1}(E_k)} \mathbb{E}\langle\frac{\sqrt{n}\nabla L(x,y)}{\|\nabla L(x,y)\|} - \nabla\tilde{\Phi}(x), \nabla\tilde{u}_i(x)\rangle^4$$

$$\lesssim n^2 \sup_{x \in u^{-1}(E_k)} \mathbb{E}\langle\frac{\nabla L(x,y)}{\|\nabla L(x,y)\|}, \nabla\tilde{u}_i(x)\rangle^4$$

$$\leq n^2 \sup_{x \in u^{-1}(E_k)} \sqrt{\mathbb{E}\langle\nabla L(x,y), \nabla\tilde{u}_i(x)\rangle^8 \mathbb{E}\frac{1}{\|\nabla L(x,y)\|^8}}$$

Where in the second line, we use $(a+b)^4 \lesssim a^4 + b^4$ and Jensen's Inequality to remove the $\nabla\tilde{\Phi}$ term at the cost of a constant factor. We now recall that $\nabla L(x,y)$ is n-dimensional Gaussian with mean $O(1)$ and covariance $\Sigma = I_d + \lambda xx^t$, $\|\Sigma\|_{op} = O(1)$, $\|\Sigma\|_F = \Theta(n)$. For both $\sqrt{n}m$ and $\sqrt{n}r^2$ we have that $\langle\nabla L(x,y), \nabla\tilde{u}_i(x)\rangle$ is Gaussian with variance bounded by $C_k n$ and by a technical lemma for Gaussians we have that $\mathbb{E}\frac{1}{\|\nabla L(x,y)\|^8} = O(1/n^4)$ yielding:

$$\max_i \sup_{x \in u^{-1}(E_k)} \mathbb{E}\langle\sqrt{n}\nabla\tilde{H}(x,y), \nabla\tilde{u}_i(x)\rangle^4 = O(n^2)$$

As required. Lastly we consider:

$$\sup_{x \in u^{-1}(E_k)} \mathbb{E}\langle\nabla^2 u_i(x), \nabla\tilde{H}(x,y) \otimes \nabla\tilde{H}(x,y) - \tilde{V}\rangle^2$$

$$= n \sup_{x \in u^{-1}(E_k)} \mathbb{E}\,\mathrm{tr}([I - vv^t](\nabla\tilde{H}(x,y) \otimes \nabla\tilde{H}(x,y) - \tilde{V}))^2$$

$$\lesssim n \sup_{x \in u^{-1}(E_k)} \mathbb{E}\,\mathrm{tr}(\nabla\tilde{H}(x,y) \otimes \nabla\tilde{H}(x,y) - \tilde{V})^2 + \mathbb{E}\mathrm{tr}(vv^\top[\nabla\tilde{H}(x,y) \otimes \nabla\tilde{H}(x,y) - \tilde{V}])^2$$

We now bound the first term above noting that the second term above is bounded by (e.g. using the Von Neumann trace inequality) the first term multiplied by $\|vv^t\|_{\mathrm{op}} = O(1)$:

$$\mathrm{tr}(\nabla\tilde{H}(x,y) \otimes \nabla\tilde{H}(x,y) - \tilde{V}) = \mathrm{tr}\left(\left[\frac{\sqrt{n}\nabla L(x,y)}{\|\nabla L(x,y)\|} - \nabla\tilde{\Phi}(x)\right]^{\otimes 2} - \mathbb{E}\left[\frac{\sqrt{n}\nabla L(x,y)}{\|\nabla L(x,y)\|} - \nabla\tilde{\Phi}\right]^{\otimes 2}\right)$$

$$= \text{tr}\left(\frac{\sqrt{n}\nabla L(x,y)^{\otimes 2}}{\|\nabla L(x,y)\|^2}\right) - 2\text{tr}\left(\frac{\sqrt{n}\nabla L(x,y)}{\|\nabla L(x,y)\|} \otimes \nabla\tilde{\Phi}(x)\right) + \text{tr}(\nabla\tilde{\Phi}(x)^{\otimes 2})$$

$$- \mathbb{E}[\text{tr}\left(\frac{\sqrt{n}\nabla L(x,y)^{\otimes 2}}{\|\nabla L(x,y)\|^2}\right) - 2\text{tr}\left(\frac{\sqrt{n}\nabla L(x,y)}{\|\nabla L(x,y)\|} \otimes \nabla\tilde{\Phi}(x)\right) + \text{tr}(\nabla\tilde{\Phi}(x)^{\otimes 2})]$$

$$= 2\,\text{tr}\left(\frac{\sqrt{n}\nabla L(x,y)}{\|\nabla L(x,y)\|} \otimes \nabla\tilde{\Phi}(x)\right) - 2\mathbb{E}\text{tr}\left(\frac{\sqrt{n}\nabla L(x,y)}{\|\nabla L(x,y)\|} \otimes \nabla\tilde{\Phi}(x)\right)$$

Noting that $\text{tr}(\frac{\nabla L(x,y)^{\otimes 2}}{\|\nabla L(x,y)\|^2}) = 1$ deterministically. We thus get the following bound:

$$\sup_{x\in u^{-1}(E_k)} \mathbb{E}\text{tr}(\nabla\tilde{H}(x,y) \otimes \nabla\tilde{H}(x,y) - \tilde{V})^2 \lesssim \sup_{x\in u^{-1}(E_k)} \mathbb{E}\text{tr}\left(\frac{\sqrt{n}\nabla L(x,y)}{\|\nabla L(x,y)\|} \otimes \nabla\tilde{\Phi}(x)\right)^2$$

$$\leq n^2 \sup_{x\in u^{-1}(E_k)} \|\nabla\tilde{\Phi}(x)\|^2$$

$$\leq C_k\, n$$

where in the first line we use the fact $(a-b)^2 \lesssim a^2 + b^2$ and Jensen's inequality. In the second line we use Cauchy Schwartz to bound the trace (which is here equal to an inner product) and in the last line we use item (2) of $(\delta_n, \eta_n)$-localizability above. Which yields the desired:

$$\max_i \sup_{x\in u^{-1}(E_k)} \mathbb{E}\langle\nabla^2 u_i(x), n\nabla\tilde{H}(x,y) \otimes \nabla\tilde{H}(x,y) - \tilde{V}\rangle^2 = o(n^3)$$

$\square$

We conclude this subsection by proving the limiting dynamics of Proposition 3.1 for SGD-M and SGD-U.

*Proof of Proposition 3.1.* Following Theorem 2.3 and the discussion following it, we find that the dynamics of SGD-M follow those of online SGD up to the additional $\beta$-dependent constants. With this, our analysis immediately follows from (Ben Arous et al., 2024, Section 7) where we take the mappings,

$$f \mapsto \frac{1}{1-\beta}f, \quad g \mapsto \frac{1}{(1-\beta)^2}g$$

for $\beta \in (0,1)$. For example, by (Ben Arous et al., 2024, Proposition 3.1), the limiting generators for online SGD are given by

$$f_m = -2\lambda k m^{k-1} + 2kR^{2k-2}m, \quad f_{r^2} = 4r^2 kR^{2k-2}$$

and $g_m = 0$, $g_{r^2} = 4c_\delta kR^{2k-2}$ for the correctors. The dynamics of SGD-M immediately following from the rescaled mappings we discussed.

We now turn our attention to SGD-U and recall that $\nabla m = v$ and $\nabla r^2 = 2(x - mv)$ and note that $\|\nabla m\| = 1$ and $\|\nabla r^2\| \leq 2\|x\|$. Substituting all of these into Lemma C.1 yields the pre-limiting drifts

$$\tilde{f}_m = \frac{\sqrt{n}}{D}\langle\nabla\Phi, \nabla m\rangle + O(n^{-1}\|\nabla m\|) = \sqrt{\frac{n}{\text{tr}(V) + \|\nabla\Phi\|^2}}\, f_m + o(1)$$

$$= \sqrt{\frac{n}{4nkR^{2k-2} + 4k(k-1)R^{2k-4}}}(-2\lambda k m^{k-1} + 2kR^{2k-2}m) + o(1)$$

$$= \frac{\sqrt{k}}{R^{k-1} + o(1)}(R^{2k-2}m - \lambda m^{k-1}) + o(1)$$

$$\tilde{f}_{r^2} = \frac{\sqrt{n}}{D}\langle\nabla\Phi, \nabla r^2\rangle + O(n^{-1}\|\nabla r^2\|) = \frac{1}{2\sqrt{k}R^{k-1} + o(1)}\, f_{r^2} + o(1)$$

$$= \frac{4r^2 kR^{2k-2}}{2\sqrt{k}R^{k-1} + o(1)} + o(1) = \frac{2r^2\sqrt{k}R^{k-1}}{1 + o(1)} + o(1)$$

Under the scaling that $n \to \infty$, we have the limiting functions

$$\tilde{f}_m = \sqrt{k}R^{k-1}m - \lambda\sqrt{k}\left(\frac{m}{R}\right)^{k-1}, \quad \tilde{f}_{r^2} = 2r^2\sqrt{k}R^{k-1}$$

For the second-order generators, recall that $\nabla^2 m = 0$ and $\nabla^2 r^2 = 2(I - vv^\top)$ and note that $\|\nabla r^2\| \sim \sqrt{n}$. Substituting all of these into Lemma C.2 yields the limiting corrector term for $r^2$,

$$\tilde{g}_{r^2} = \frac{c_\delta}{n} \cdot \frac{n}{2D^2}\langle V, \nabla r^2\rangle + O\left(n^{-3/2}\|\nabla^2 r^2\|\right)$$

$$= \frac{n}{4nkR^{2k-2} + 4k(k-1)R^{2k-4}} \cdot 4c_\delta kR^{2k-2} + o(1) = \frac{4c_\delta kR^{2k-2}}{4kR^{2k-2} + o(1)} + o(1) \xrightarrow{n\to\infty} c_\delta$$

To conclude, we determine the volatility matrix using our established approximations

$$\delta JVJ^\top = \frac{c_\delta}{n} \cdot nJ\left(\frac{V}{D^2} + O(n^{-3/2})\right)J^\top = c_\delta\frac{JVJ^\top}{D^2} + O(n^{-3/2}\|J\|_{\mathrm{op}}^2)$$

The error term is thus negligible as $\|J\|_{\mathrm{op}} = O(\sqrt{n})$ by $\delta$-localizability (i.e. the norms of the summary statistics satisfy $\|\nabla u_i\| = O(\sqrt{n})$). Recalling the volatility matrix for Tensor PCA under online SGD, we have the following:

$$\frac{n}{D^2}JVJ^\top = \frac{1}{4kR^{2k-2} + o(1)}\begin{bmatrix} 4k(k-1)m^2R^{2k-4} + 4kR^{2k-2} & 4k(k-1)m(R^2 - m)R^{2k-4} \\ 4k(k-1)m(R^2 - m)R^{2k-4} & 4k(k-1)(R^2 - m)^2R^{2k-4} \end{bmatrix}$$

$$= \begin{bmatrix} (k-1)\xi^2 + 1 & (k-1)\xi(R - \xi) \\ (k-1)\xi(R - \xi) & (k-1)(R - \xi)^2 \end{bmatrix}$$

where we set the direction cosine $\xi = m/R$. The volatility matrix evidently vanishes for $\delta = c_\delta/n$. $\qquad\square$

## C.2 PROOF OF PROPOSITIONS 3.2 AND 3.3

The proof of Proposition 3.2 follows directly from the case of online SGD in Ben Arous et al. (2024) and Theorem 2.3. In particular the equivalence of SGD and SGD-M after appropriate time and step-size rescaling. Below we prove Proposition 3.3.

*Proof.* We first determine the fixed points for the case when $k = 2$. In particular, we solve the system

$$m\sqrt{k}R(\lambda - R^2) = 0, \quad 2\sqrt{k}r^2R = c_\delta$$

When $m = 0$, it follows that $R^2 = r^2$ so that we obtain

$$r^3 = \frac{c_\delta}{2\sqrt{2}} \implies r^2 = \left(\frac{c_\delta}{2\sqrt{2}}\right)^{2/3} = \frac{c_\delta^{2/3}}{2}$$

On the other hand when $m \neq 0$, we have that $R^2 = \lambda$ which implies that

$$r^2 = \frac{c_\delta}{2\sqrt{2\lambda}} \implies m^2 = \lambda - \frac{c_\delta}{2\sqrt{2\lambda}}$$

for fixed $\lambda > 0$. In particular, the solutions exist if and only if

$$\lambda \geq \frac{c_\delta^{2/3}}{2} = \lambda_c(2)$$

which is the critical $\lambda$ for the case of preconditioning and $k = 2$. To treat the general $k \geq 3$, the system under consideration reads

$$\lambda\left(\frac{m}{R}\right)^{k-1} = R^{k-1}m, \quad c_\delta = 2R^{k-1}\sqrt{k}r^2$$

where rearranging the equation involving $r^2$ yields,

$$r^2 = \frac{c_\delta}{2\sqrt{k}R^{k-1}}$$

Introduce the direction cosine $\xi := m/R \in [-1, 1]$ so that $m = \xi R$ and $r^2 = R^2(1 - \xi^2)$. This in turn allows us to isolate the equation above in terms of $R^2$ by writing,

$$(1 - \xi^2)R^{k+1} = \frac{c_\delta}{2\sqrt{k}}.$$

In particular, from the first equation we see that

$$\lambda \xi^{k-1} = R^k \xi \quad \Longrightarrow \quad \xi = 0 \text{ or } R^k = \lambda \xi^{k-2}$$

implying that all equilibria are obtained by solving the second equation together with either of the two conditions. Hereafter, we set $A := c_\delta/(2\sqrt{k})$.

First, if $\xi = 0$ (i.e. $m = 0$), then we obtain

$$R^2 = r^2 = \left(\frac{c_\delta}{2\sqrt{k}}\right)^{2/(k+1)}$$

On the other hand, for $\xi \neq 0$ we have

$$R^{k+1} = \frac{A}{1 - \xi^2}, \quad R^k = \lambda \xi^{k-2} \quad \Longrightarrow \quad \frac{A^k}{(1 - \xi^2)^k} = \lambda^{k+1} \xi^{(k-2)(k+1)}$$

This yields the equivalent condition with respect to $\xi$ as,

$$F(\xi) := \xi^{(k-2)(k+1)}(1 - \xi^2)^k = \frac{A^k}{\lambda^{k+1}}, \quad \xi \in [0, 1).$$

To conclude our analysis, we record the following facts about $F$:

1. $F(0) = F(1) = 0$ and $F(\xi) > 0$ for $\xi \in (0, 1)$.

2. $F$ has a unique critical point on $(0, 1)$, which is a global maximum at

$$\xi_*^2 = \frac{(k - 2)(k + 1)}{(k - 1)(k + 2)} \in (0, 1).$$

   This can be seen by considering the $\log F(\xi)$ so that

$$\frac{F'(\xi)}{F(\xi)} = \frac{(k - 2)(k + 1)}{\xi} - \frac{2k\xi}{1 - \xi^2}$$

   Solving for $F'(\xi) = 0$ will recover the critical point above.

3. The maximum value is

$$F_{\max} := F(\xi_*) = \frac{\left((k - 2)(k + 1)\right)^{\frac{(k-2)(k+1)}{2}}(2k)^k}{\left((k - 1)(k + 2)\right)^{\frac{(k-2)(k+1)}{2} + k}}.$$

Therefore, the solutions for $F(\xi)$ only exist so long as

$$F(\xi) = \frac{A^k}{\lambda^{k+1}} \leq F_{\max} \implies \lambda_{\mathrm{crit}}(k, c_\delta) = \left[\frac{\left(\frac{c_\delta}{2\sqrt{k}}\right)^k ((k - 1)(k + 2))^{\frac{(k-2)(k+1)}{2} + k}}{(2k)^k ((k - 2)(k + 1))^{\frac{(k-2)(k+1)}{2}}}\right]^{1/(k+1)} \qquad \square$$

## C.3 DIFFUSIVE LIMITS FOR TENSOR PCA: PROOF OF PROPOSITION 3.4

*Proof.* Applying Theorem 2.3 in light of 2.1, we simply check asymptotic closability, noting that $(\delta_n, \eta_n)$-localizability was already verified for this setting in the proof of Proposition 3.1. We check asymptotic closability which also follows easily from the same proof. The drifts under the new coordinates are written as

$$\langle \nabla \tilde{\Phi}, \nabla \tilde{u}_1 \rangle = \frac{n}{D} \langle \nabla \Phi, \nabla m \rangle + O(n^{-1/2}) = \sqrt{n} R^{k-1} m - \lambda \sqrt{n} \left(\frac{m}{R}\right)^{k-1}$$

$$= \left(r^2 + (\tilde{u}_1^2/n)\right)^{k-1} \tilde{u}_1 - \lambda n^{-\frac{k-2}{2}} \left(\frac{\tilde{u}_1}{\sqrt{r^2 + (\tilde{u}_1^2/n)}}\right)^{k-1}$$

$$\langle \nabla \tilde{\Phi}, \nabla r^2 \rangle = 2r^2 \left(r^2 + (\tilde{u}_1^2/n)\right)^{\frac{k-1}{2}}$$

Taking limits as $n \to \infty$, as long as $\lambda$ is fixed in $n$, we obtain the rescaled drift

$$\tilde{f}_{\tilde{u}_1} = \begin{cases} \tilde{u}_2^{k-1}\tilde{u}_1 - \lambda \left(\dfrac{\tilde{u}_1}{\sqrt{\tilde{u}_2}}\right)^{k-1} & k = 2 \\ \tilde{u}_2^{k-1}\tilde{u}_1 & k \geq 3 \end{cases}, \qquad \tilde{f}_{\tilde{u}_2} = 2\tilde{u}_2^{(k+1)/2}$$

The volatility matrix (under the new coordinates) is given by the limit

$$\tilde{J}\tilde{V}\tilde{J}^\top = \begin{bmatrix} (k-1)(\tilde{u}_1^2/\tilde{u}_2) + 1 & 0 \\ 0 & 0 \end{bmatrix}$$

where the only surviving entry after the limit is $\Sigma_{11} = c_\delta(k-1)(\tilde{u}_1^2/\tilde{u}_2) + c_\delta$. $\qquad \square$

## D PROOFS FOR SINGLE-INDEX MODELS

We start with SGD-M below. The limiting generators were first determined in (Rangriz, 2025, Theorem 3.1, Corollary 3.2). We provide a slightly different derivation of the result for completeness. Refer to (Rangriz, 2025, Lemma 3.5) to show that the problem satisfies the $\delta$-localizability and closability assumptions as they also apply to SGD-M.

### D.1 PROOF OF PROPOSITION 3.5

*Proof for SGD-M Dynamics.* We first establish some preliminaries to simplify subsequent arguments and notation. For the direction vector $v$, we write $x$ according to the decomposition

$$x = \langle x, v \rangle v + (I - vv^\top)x = mv + bu$$

where $u = \nabla r^2 / \|\nabla r^2\| = (x - mv)/\|x - mv\|$ is an unit vector orthogonal to $v$, i.e. $\langle u, v \rangle = 0$ and $b = \|x - mv\| = \|\nabla r^2\|/2$.

By rotational invariance, any standard Gaussian vector $g \sim \mathcal{N}(0, I)$ admits the decomposition

$$a = a_1 v + a_2 u + w, \quad a_1 = \langle v, g \rangle, a_2 = \langle u, g \rangle$$

where $a_1, a_2$ are independent Gaussian variables from $\mathcal{N}(0,1)$ and $w := g - a_1 v - a_2 u$ is a centered Gaussian vector independent from $a_1, a_2$ and orthogonal to $\mathrm{Span}\{u, v\}$ with covariance $\mathbb{E}[ww^\top] = I - uu^\top - vv^\top$.

We define the scalar projections $s := \langle x, a \rangle = ma_1 + ra_2$ (as $\langle x, w \rangle = 0$ by construction) and $t := \langle v, a \rangle = a_1$. Let $\Delta := f(s) - f(t) + \varepsilon$ denote the residual error.

To determine the expected loss $\nabla \Phi = \mathbb{E}[\nabla L]$, note that by the chain rule

$$\nabla L = 2\left(f(s) - f(t) + \varepsilon\right) f'(s) \nabla s = 2\Delta f'(s)a$$

Substituting the original decomposition $a = a_1 v + a_2 u + w$ and rearranging terms yields,

$$\nabla_x L = 2\Delta f'(s)(a_1 v + a_2 u + w) = (2a_1 \Delta f'(s))v + (2a_2 \Delta f'(s))u + 2\Delta f'(s)w$$

In particular, noting that $w$ is independent of $a_1, a_2, \epsilon$ we find that

$$\nabla \Phi = 2\mathbb{E}[a_1 \Delta f'(s)]v + 2\mathbb{E}[a_2 \Delta f'(s)]u$$

Now to determine the covariance matrix $\mathrm{Cov}(\nabla H)$, we first show that the component of $\nabla L$ lying on $\mathrm{span}\{v, u\}$ and the remainder are uncorrelated to simplify our analysis. This follows by noting

$$\mathrm{Cov}((2a_1 \Delta f'(s))v + (2a_2 \Delta f'(s))u, 2\Delta f'(s)w) = 4\mathbb{E}[\Delta^2 [f'(s)]^2 (a_1 v + a_2 u)w] = 0$$

due to independence of $w$ from $\{a_1, a_2\}$. Thus, we have the orthogonal decomposition for the covariance matrix

$$
\begin{aligned}
V = \mathrm{Cov}(\nabla L) &= 4\mathrm{Cov}\left(\Delta f'(s)\left[(a_1 v + a_2 u) + w\right]\right) \\
&= 4\mathrm{Cov}\left(\Delta f'(s)(a_1 v + a_2 u)\right) + 4\mathrm{Cov}\left(\Delta f'(s)w\right) \\
&= 4\mathrm{Var}[\Delta f'(s)a_1]vv^\top + 4\mathrm{Var}[\Delta f'(s)a_2]uu^\top + 4\mathrm{Cov}(\Delta f'(s)a_1, \Delta f'(s)a_2)[uv^\top + vu^\top] \\
&\quad + 4\mathbb{E}[\Delta^2[f'(s)]^2](I - uu^\top - vv^\top)
\end{aligned}
$$

We now compute each of the generators using the quantities obtained above. In particular, the first order generators are given by

$$
f_m = \langle \nabla \Phi, \nabla m \rangle = 2\mathbb{E}[a_1 \Delta f'(s)] = 2\mathbb{E}[a_1 f'(ma_1 + ra_2)\left(f(ma_1 + ra_2) - f(a_1)\right)]
$$

$$
f_{r^2} = \langle \nabla \Phi, \nabla r^2 \rangle = 4r\mathbb{E}[a_2 \Delta f'(s)] = 4\mathbb{E}[a_1 f'(ma_1 + ra_2)\left(f(ma_1 + ra_2) - f(a_1)\right)]
$$

where we also simplify the expectation using the fact that $\epsilon$ are independent zero-mean errors. The second-order generators are $g_m = 0$ and

$$
\frac{c_\delta}{2n}\langle V, \nabla^2 r^2 \rangle = \frac{c_\delta}{n}\left(4\mathrm{Var}[\Delta f'(s)a_2] + 4\mathbb{E}[\Delta^2[f'(s)]^2](n-2)\right)
$$

$$
\overset{n\to\infty}{\to} 4c_\delta \mathbb{E}\left[[f'(ma_1 + ra_2)]^2[(f(ma_1 + ra_2) - f(a_1))^2 + \sigma^2]\right] = g_{r^2}
$$

where we again use the independence of $\epsilon$ and the fact that $\mathbb{E}[\epsilon^2] = \sigma^2$ by assumption. Finally, we determine the pre-limit for the volatility matrix as

$$
JVJ^\top = \begin{bmatrix} 4\mathrm{Var}[\Delta f'(s)a_1] & 8r\,\mathrm{Cov}(\Delta f'(s)a_1, \Delta f'(s)a_2) \\ 8r\,\mathrm{Cov}(\Delta f'(s)a_1, \Delta f'(s)a_2) & 16r^2\mathrm{Var}[\Delta f'(s)a_2] \end{bmatrix}
$$

which evidently vanishes as $n \to \infty$. Recall that we set $s := \langle x, g \rangle = ma_1 + ra_2$ and $\Delta := f(s) - f(t) + \varepsilon$. $\qquad\square$

We now prove the remainder of the proposition for SGD-U. We note that verifying $(\delta_n, \eta_n)$-localizability is a similar exercise as to the case of Tensor PCA in Section C.1.2. We thus cover the key observations below. Again, we consider here $\tilde{u}(x) = \sqrt{n}u(x) = \sqrt{n}(m(x), r_\perp^2(x))$ which implies all cases of interest (i.e. $m$ and $r^2$ up to a rescaling of $\sqrt{n}$ with appropriate centering).

*Proof for SGD-U $(\delta_n, \eta_n)$-localizability.* Item (1) is the same as the SGD-M case above. For item (2), the preconditioned gradient is:

$$
\begin{aligned}
\eta \nabla L &= \frac{\sqrt{n}}{2|\Delta \sigma'(s)|\|a\|}(2\Delta \sigma'(s)a) \\
&= \sqrt{n}\,\mathrm{sgn}(\Delta \sigma'(s))\left[\frac{a_1 v + a_2 u}{\|a\|} + \frac{w}{\|a\|}\right]
\end{aligned}
$$

Which implies that $\|\tilde{\Phi}(x)\| = O(1)$ given the expectation of each fraction above is $O(1/\sqrt{n})$. The above also trivially gives $\|\nabla \tilde{H}\|^8 = O(\delta_n^{-4})$, concluding item (2). For item (3) we once again (as in the tensor PCA example) have:

$$
\sup_{x \in u^{-1}(E_k)} \mathbb{E}\langle \nabla \tilde{H}(x,y), \nabla \tilde{u}_i(x) \rangle^4 \le n^2 \sup_{x \in u^{-1}(E_k)} \sqrt{\mathbb{E}\langle \nabla L(x,y), \sqrt{n}\nabla u_i(x) \rangle^8 \mathbb{E}\frac{1}{\|\nabla L(x,y)\|^8}}
$$

We remind the reader that $\nabla L = 2\Delta f'(s)a$ and $\nabla u(x) = (v, 2(x - mv))$, hence $\langle \nabla L(x,y), \sqrt{n}\nabla u_i(x) \rangle = O(\sqrt{n})$. Additionally, $\mathbb{E}\|\nabla L(x,y)\|^{-1} = O(1/\sqrt{n})$ which yields:

$$
\sup_{x \in u^{-1}(E_k)} \mathbb{E}\langle \nabla \tilde{H}(x,y), \nabla u_i(x) \rangle^4 = O(n^2) = O(\delta^{-2})
$$

The last part of item (3) follows identically to the Tensor PCA example. $\qquad\square$

*Proof for SGD-U asymptotic closability.* Recall that the gradient of the loss function is given by $\nabla L(x, Y) = 2\Delta\sigma'(s)a$ follow the established notation in the online SGD case. The norm of the gradient is $\|\nabla L\| = 2|\Delta\sigma'(s)|\|a\|$.

The preconditioned gradient $G(x, Y)$ is:

$$G(x, Y) = \eta\nabla L = \frac{\sqrt{n}}{2|\Delta\sigma'(s)|\|a\|}(2\Delta\sigma'(s)a) = \sqrt{n}\,\text{sgn}(\Delta\sigma'(s))\frac{a}{\|a\|}$$

To simplify notation, we write $S(x, Y) = \text{sgn}(\Delta\sigma'(s))$ and $\hat{a} = a/\|a\|$ so that

$$G(x, Y) = \sqrt{n}S\hat{a}$$

The expected gradient $\nabla\tilde{\Phi}(x) = \mathbb{E}[G(x, Y)] = -\sqrt{n}\mathbb{E}[S\hat{a}]$ can be further decomposed as

$$\nabla\tilde{\Phi}(x) = \sqrt{n}\mathbb{E}\left[S\frac{a_1v + a_2u + w}{\|a\|}\right] = \sqrt{n}\left(\mathbb{E}\left[S\frac{a_1}{\|a\|}\right]v + \mathbb{E}\left[S\frac{a_2}{\|a\|}\right]u\right)$$

To see why the last component vanishes in expectation, recall that the law of $w$ is spherical and symmetric about zero. Furthermore, for fixed $a_1, a_2$,

$$\frac{w}{\|a\|} = \frac{w}{\sqrt{a_1^2 + a_2^2 + \|w\|^2}}$$

is an odd function of $w$. By conditioning on $a_1, a_2, \epsilon$ we conclude that

$$\mathbb{E}\left[S\frac{w}{\|a\|}\right] = \mathbb{E}\left[S\mathbb{E}\left[\frac{w}{\|a\|}\ \middle|\ a_1, a_2, \epsilon\right]\right] = 0$$

Now consider the covariance matrix is $\tilde{V}(x) = \text{Cov}(G(x, Y)) = \mathbb{E}[GG^\top] - \nabla\tilde{\Phi}\nabla\tilde{\Phi}^\top$. The second moment matrix $\mathbb{E}[GG^\top]$ can be deduced as

$$\mathbb{E}[GG^\top] = \mathbb{E}[nS^2\hat{a}\hat{a}^\top] = n\mathbb{E}[\hat{a}\hat{a}^\top].$$

where $S^2 = 1$ almost surely. Since $a$ is an isotropic Gaussian vector, $\hat{a}$ is uniformly distributed on the sphere. It follows that $\mathbb{E}[\hat{a}\hat{a}^\top] = I/n$ (see, for instance (Vershynin, 2018, Proposition 3.3.8) for the construction). It follows then that the covariance matrix is:

$$\tilde{V}(x) = I - \nabla\tilde{\Phi}(x) \otimes \nabla\tilde{\Phi}(x)$$

We now turn our attention to the limiting generators. First, we see that $\nabla\tilde{\Phi}$ admits an asymptotic simplification due to the behaviour of $\|a\|$. In particular, we see that for the coefficient of the basis vector $v$,

$$\left|\mathbb{E}\left[Sa_1\frac{\sqrt{n}}{\|a\|} - Sa_1\right]\right| \leq \sqrt{\mathbb{E}(S^2a_1^2)\mathbb{E}\left(\frac{\sqrt{n}}{\|a\|} - 1\right)^2} = \sqrt{\mathbb{E}\left(\frac{\sqrt{n}}{\|a\|} - 1\right)^2}$$

by Cauchy-Schwartz where $S^2 = 1$ almost surely and $\mathbb{E}[a_1^2] = 1$. By Lemma E.3 (assuming $n \geq 9$) and Jensen's inequality, we have

$$\mathbb{E}\left[\frac{n}{\|a\|^2}\right] \leq \frac{n}{n-2} \to 1 \quad \text{and} \quad \mathbb{E}\left[\frac{\sqrt{n}}{\|a\|}\right] \leq \sqrt{\frac{n}{n-2}} \to 1$$

which shows that $\sqrt{n}\,\mathbb{E}[Sa_1/\|a\|] = \mathbb{E}[Sa_1] + o(1)$. Similarly, we can show that $\sqrt{n}\,\mathbb{E}[Sa_2/\|a\|] = \mathbb{E}[Sa_2] + o(1)$. A useful observation is that $\|\nabla\tilde{\Phi}\| = O(1)$ with respect to $n$ as each of the coefficients for the unit vectors $v, u$ are bounded.

With this, it follows that the first-order generators for $\nabla m = v$ and $\nabla r^2 = 2ru$ are

$$\langle\nabla\tilde{\Phi}, \nabla m\rangle = \mathbb{E}[Sa_1] + o(\|v\|) \overset{n\to\infty}{\to} \mathbb{E}[a_1 \cdot \text{sgn}(\sigma'(s)(\sigma(s) - \sigma(a_1) + \epsilon))] = \tilde{f}_m$$

$$\langle\nabla\tilde{\Phi}, \nabla r^2\rangle = 2r\mathbb{E}[Sa_2] + o(\|u\|) \overset{n\to\infty}{\to} 2r\mathbb{E}[a_2 \cdot \text{sgn}(\sigma'(s)(\sigma(s) - \sigma(a_1) + \epsilon))] = \tilde{f}_{r^2}$$

For the second-order generator, again we have that $g_m = 0$ and

$$\frac{c_\delta}{2n}\langle\tilde{V}, \nabla^2 r^2\rangle = \frac{c_\delta}{n}\left((n-1) + \langle\nabla\tilde{\Phi} \otimes \nabla\tilde{\Phi}, I - vv^\top\rangle\right) \overset{n\to\infty}{\to} c_\delta = g_{r^2}$$

since $|\langle \nabla \tilde{\Phi} \otimes \nabla \tilde{\Phi}, I - vv^\top \rangle| \leq \|I - vv^\top\|_{\text{op}} \|\nabla \tilde{\Phi}\|^2 = O(1)$. It remains to compute the volatility matrix for the preconditioned covariance matrix.

$$J\tilde{V}J^\top = \begin{bmatrix} 1 - (\mathbb{E}[Sa_1])^2 + o(1) & -2r\mathbb{E}[Sa_1]\mathbb{E}[Sa_2] + o(1) \\ -2r\mathbb{E}[Sa_1]\mathbb{E}[Sa_2] + o(1) & 1 - 4r^2(\mathbb{E}[Sa_2])^2 + o(1) \end{bmatrix}$$

which evidently vanishes as $n \to \infty$ once we incorporate the additional factor of $\delta = c_\delta/n$. Recall that we define

$$S = \text{sgn}(\sigma'(ma_1 + ra_2)(\sigma(ma_1 + ra_2) - \sigma(a_1) + \epsilon))$$

and $s = ma_1 + ra_2$. $\qquad \square$

## D.2 FIXED POINT ANALYSIS FOR SGD-M (MONOMIAL)

In this section we provide the proof for Propositions 3.7 and 3.8. Before doing so, we prove a more general result and obtain the dynamics for any monomial link $f(x) = x^k$ with $k \geq 1$. To simplify notation, recall that we are given the dynamics for $(m, r_\perp^2)$ by:

$$\frac{\mathrm{d}m}{\mathrm{d}t} = \frac{-2}{1-\beta}E_m, \qquad \frac{\mathrm{d}r_\perp^2}{\mathrm{d}t} = \frac{-4}{1-\beta}\left(E_{r,1} - C'E_{r,2}\right),$$

where $C' = \frac{c_\delta}{1-\beta}$ and the expectations are defined as:

$$E_m = \mathbb{E}\left[a_1 f'(Z)(f(Z) - f(a_1))\right],$$
$$E_{r,1} = \mathbb{E}\left[a_2 r_\perp f'(Z)(f(Z) - f(a_1))\right],$$
$$E_{r,2} = \mathbb{E}\left[f'(Z)^2\left((f(Z) - f(a_1))^2 + \sigma^2\right)\right].$$

Here $Z = a_1 m + a_2 r_\perp$, and $a_1, a_2 \sim \mathcal{N}(0, 1)$ are independent. By Stein's Lemma we may further rewrite the expectations as

$$E_{r,1} = r_\perp\mathbb{E}[a_2 f'(Z)(f(Z) - f(a_1))] = r_\perp\mathbb{E}\left[\frac{\partial}{\partial a_2}\{f'(Z)(f(Z) - f(a_1))\}\right]$$
$$= r_\perp\mathbb{E}[r_\perp f''(Z)(f(Z) - f(a_1)) + r_\perp f'(Z)^2] = r_\perp^2 P$$

$$E_m = \mathbb{E}[a_1 f'(Z)(f(Z) - f(a_1))] = \mathbb{E}\left[\frac{\partial}{\partial a_1}\{f'(Z)(f(Z) - f(a_1))\}\right]$$
$$= \mathbb{E}[m f''(Z)(f(Z) - f(a_1)) + f'(Z)(m f'(Z) - f'(a_1))] = mP - Q$$

which yields us the reduced dynamics

$$\frac{\mathrm{d}m}{\mathrm{d}t} = \frac{-2}{1-\beta}(mP - Q), \qquad \frac{\mathrm{d}r_\perp^2}{\mathrm{d}t} = \frac{-4}{1-\beta}(r_\perp^2 P - C'E_{r,2}),$$

We set $f(x) = x^k$. We define the joint moments $M_{p,q}(m, r_\perp^2) = \mathbb{E}[Z^p a_1^q]$. These are polynomials in $m$ and $r_\perp$ given by the Binomial expansion:

$$M_{p,q} = \mathbb{E}\left[\sum_{j=0}^{p}\binom{p}{j}m^j r_\perp^{p-j}a_1^{j+q}a_2^{p-j}\right] = \sum_{j=0}^{p}\binom{p}{j}m^j r_\perp^{p-j}\mu_{j+q}\mu_{p-j},$$

where $\mu_n$ is the $n$-th moment of a standard Gaussian. Substituting $f(x) = x^k$ into $P, Q, E_{r,2}$:

$$P = \mathbb{E}[k(k-1)Z^{k-2}(Z^k - a_1^k)] + \mathbb{E}[k^2 Z^{2k-2}] = k(2k-1)M_{2k-2,0} - k(k-1)M_{k-2,k}.$$

$$Q = \mathbb{E}[kZ^{k-1} \cdot ka_1^{k-1}] = k^2 M_{k-1,k-1}.$$

$$E_{r,2} = k^2\mathbb{E}[Z^{2k-2}((Z^k - a_1^k)^2 + \sigma^2)] = k^2(M_{4k-2,0} - 2M_{3k-2,k} + M_{2k-2,2k} + \sigma^2 M_{2k-2,0}).$$

Following this notation, the fixed points to the system satisfy $mP - Q = 0$ and $r_\perp^2 P - C'E_{r,2} = 0$. We necessarily have that any fixed point must satisfy $r_\perp^2 > 0$ under the construction that $\sigma^2 > 0$, $k \geq 1$ and $c_\delta > 0$ since they imply that $E_{r,2} > 0$ and $C' > 0$. The global optimum is $(1, 0)$, however it is not necessarily a fixed point in the presence of noise. We now prove the two results of interest.

*Proof of Proposition 3.7.* For $f(x) = x^2$, we have that the quantities in the dynamics reduce to

$$P = 2(4-1)\mathbb{E}[(a_1 m + a_2 r)^2] - 2(1)\mathbb{E}[a_1^2] = 6R^2 - 2$$

$$Q = 4\mathbb{E}[(a_1 m + a_2 r)a_1] = 4m$$

$$E_{r,2} = 4\left(\mathbb{E}[(a_1 m + a_2 r)^6] - 2\mathbb{E}[(a_1 m + a_2 r)^4 a_1^2] + \mathbb{E}[(a_1 m + a_2 r)^2 a_1^4] + \sigma^2 \mathbb{E}[(a_1 m + a_2 r)^2]\right)$$

$$= 4\left(15R^6 - 30m^4 - 36m^2 r^2 - 6r^4 + 15m^2 + 3r^2 + \sigma^2 R^2\right)$$

where we use the fact that $a_1 m + a_2 r \sim \mathcal{N}(0, R^2)$ and apply Wick's lemma for computing the higher-order joint moments. To determine the fixed points, note that they must satisfy $6m(R^2 - 1) = 0$ which implies $m = 0$ or $R^2 = 1$.

In the case of $m = 0$, we have $R^2 = r^2$ and $(m, r^2) = (0, 0)$ is a trivial fixed point. For $r^2 > 0$, we find that fixed points exist only if there are roots to the quadratic equation with respect to $r^2$,

$$6r^2 - 2 = \frac{4c_\delta}{1 - \beta}\left(15r^4 - 6r^2 + 3 + \sigma^2\right)$$

For $R^2 = 1$, we solve the equation using $m^2 + r^2 = 1$,

$$r^2 = \frac{c_\delta}{1 - \beta}\left(12r^2 + \sigma^2\right) \implies r_*^2 = \frac{\sigma^2}{\frac{1-\beta}{c_\delta} - 12} = \frac{c_\delta \sigma^2}{1 - \beta - 12c_\delta}$$

with the corresponding $m_*^2 = 1 - r_*^2$. Finally, under the constraint that $r^2 > 0$, it follows that this fixed point exists only if $c_\delta < (1 - \beta)/12$ which completes the proof. □

*Proof of Proposition 3.8.* We analyze the local stability of the dynamics near $(m, r^2) = (1, 0)$ in the regime where $\sigma^2 \to 0$. We do so by examining the Jacobian $J$. Let $F_m = dm/dt$ and $F_r = dr^2/dt$ denote the time derivatives of both statistics. One can show that $J(1, 0)$ is upper-triangular since $\partial_m F_r|_{(1,0)}$ depends on $\partial_m E_{r,2}$ which has terms depending on products of $f(Z) - f(a_1)$. At $(1, 0)$, $Z = a_1$ so the derivative is zero. Thus it is sufficient to look at the diagonal elements (eigenvalues).

For $\lambda_m = \partial_m F_m|_{(1,0)}$, we have

$$\lambda_m = \frac{-2}{1 - \beta}\partial_m E_m|_{(1,0)} = \frac{-2}{1 - \beta}\mathbb{E}[a_1\{f''(Z)a_1(f(Z) - f(a_1)) + f'(Z)f'(Z)a_1\}]\Big|_{(1,0)}$$

At $(1, 0)$, the expectation is strictly positive so that $\lambda_m < 0$ and we see that the dynamics are stable along the $m$ direction.

For $\lambda_r = \partial_{r_\perp^2} F_r|_{(1,0)}$, we have

$$\lambda_r = \frac{-4}{1 - \beta}\left(P(1, 0) - C'\partial_{r_\perp^2} E_{r,2}|_{(1,0)}\right)$$

Evaluating each term independently, we see that $P(1, 0) = \mathbb{E}[f'(a_1)^2] = k^2 \mu_{2k-2}$ for $f(x) = x^k$.

On the other hand, we analyze the expansion of $E_{r,2}(1, r_\perp^2)$ near $r_\perp = 0$. In particular, using $f(Z) - f(a_1) = f'(a_1)a_2 r_\perp + O(r_\perp^2)$:

$$E_{r,2}(1, r_\perp^2) = \mathbb{E}\left[f'(Z)^2(f(Z) - f(a_1))^2\right]$$
$$= \mathbb{E}\left[(f'(a_1)^2 + O(r_\perp))(f'(a_1)^2 a_2^2 r_\perp^2 + O(r_\perp^3))\right]$$
$$= r_\perp^2 \mathbb{E}[f'(a_1)^4]\mathbb{E}[a_2^2] + O(r_\perp^3).$$

so that $\partial_{r_\perp^2} E_{r,2}|_{(1,0)} = \mathbb{E}[f'(a_1)^4] = k^4 \mu_{4k-4}$ for $f(x) = x^k$.

Stability requires $\lambda_r < 0$ which yields the condition

$$k^2 \mu_{2k-2} - \frac{c_\delta}{1 - \beta}k^4 \mu_{4k-4} > 0 \iff c_\delta < (1 - \beta)\frac{1}{k^2}\frac{(2k-3)!!}{(4k-5)!!}$$

by substituting the closed-form moments of the standard Gaussian. We adopt the convention $(-1)!! = 1$ for convenience. □

If we wish to obtain the *decay rate* for the upper bound, we first see that

$$\frac{(2k-3)!!}{(4k-5)!!} = \frac{(2k-2)!}{2^{k-1}(k-1)!} \cdot \frac{2^{2k-2}(2k-2)!}{(4k-4)!} = 2^{k-1}\frac{((2k-2)!)^2}{(k-1)!(4k-4)!}$$

where we use the fact that $2k-3$ and $4k-5$ are odd numbers allowing us to substitute the closed-form expression for the double factorials. Apply the Stirling approximation $m! \sim \sqrt{2\pi m}(m/e)^m$ with $m = k-1$ yo get,

$$2^{k-1}\frac{((2(k-1))!)^2}{(k-1)!(4(k-1))!} \sim \left(\frac{e}{8(k-1)}\right)^{k-1}$$

which implies that the upper bound of $c_\delta$ is of the order $k^{-k-2}$ for sufficiently large degree $k$.

### D.3 DIFFUSIVE LIMITS FOR SGD-M

We present a more general version of Proposition 3.10 below.

**Theorem D.1.** Suppose the link function $f \in C^2$ satisfies $\mathbb{E}_Z[f'(Z)] = 0$ for $Z \sim \mathcal{N}(0,1)$. Then for SGD-M, the rescaled statistics $\tilde{\boldsymbol{u}}_n(t) = (\tilde{m}, r^2) = (\sqrt{n}m, r^2)$ converges as $n \to \infty$ to the solution of the following SDE,

$$d\tilde{m} = -\frac{2}{1-\beta}\tilde{m}\mathbb{E}\left[a_1^2\left(f''(ra_2)(f(ra_2)-f(a_1)) + f'(ra_2)^2\right)\right]dt$$

$$+ \frac{1}{1-\beta}\sqrt{4c_\delta\mathbb{E}\left[a_1^2 f'(ra_2)^2\left((f(ra_2)-f(a_1))^2 + \sigma^2\right)\right]}\,dB_t$$

$$dr^2 = -\frac{4}{1-\beta}\left[r\mathbb{E}\left[a_2 f'(ra_2)(f(ra_2)-f(a_1))\right]\right.$$

$$\left. - \frac{c_\delta}{1-\beta}\mathbb{E}\left[f'(ra_2)^2\left((f(ra_2)-f(a_1))^2 + \sigma^2\right)\right]\right]dt \qquad (21)$$

where $a_1, a_2 \sim \mathcal{N}(0,1)$ independently and $B_t$ is a standard one-dimensional Brownian motion.

*Proof.* From Proposition 3.5, we rewrite the drift components for the dynamics associated with $\tilde{u}_2 = r^2$ as

$$\langle \nabla\Phi, \nabla r^2 \rangle = -4r\,\mathbb{E}\left[a_2 f'\left(a_1(\tilde{u}_1/\sqrt{n}) + a_2 r\right)\left(f\left(a_1(\tilde{u}_1/\sqrt{n}) + a_2 r\right) - f(a_1)\right)\right]$$

$$\frac{c_\delta}{2n}\langle V, \nabla^2 r^2 \rangle = 4c_\delta\,\mathbb{E}\left[f'\left(a_1(\tilde{u}_1/\sqrt{n}) + a_2 r\right)^2\left((f\left(a_1(\tilde{u}_1/\sqrt{n}) + a_2 r\right) - f(a_1))^2 + \sigma^2\right)\right]$$

Taking the limit as $n \to \infty$ recovers the drift components for $r^2$. For the dynamics of $\tilde{u}_1 = \sqrt{n}m$, we linearize the drift about $m = 0$,

$$F_m(m,r) = \mathbb{E}[a_1 f'(a_1 m + a_2 r)(f(a_1 m + a_2 r) - f(a_1))]$$

$$= \mathbb{E}[a_1 f'(a_2 r)(f(a_2 r) - f(a_1))] + m\partial_m F(0,r) + o(m)$$

By independence of $a_1, a_2$, the leading term reduces to

$$\mathbb{E}[a_1 f'(a_2 r)f(a_2 r)] - \mathbb{E}[a_1 f'(a_2 r)f(a_1)] = -\mathbb{E}[a_1 f(a_1)]\mathbb{E}[f'(a_2 r)]$$

By Stein's lemma, we have that $\mathbb{E}[a_1 f(a_1)] = \mathbb{E}[f'(a_1)]$ which is zero by assumption. For the first-order term, the partial with respect to $m$ is given by

$$\partial_m F(m,r) = \mathbb{E}[a_1^2 f''(a_1 m + a_2 r)(f(a_1 m + a_2 r) - f(a_1)) + f'(a_1 m + a_2 r)^2]$$

Thus under the rescaling of $\tilde{u}_1 = \sqrt{n}m$, we have that

$$\langle \nabla\Phi, \nabla\tilde{u}_1 \rangle = \sqrt{n}\left[\frac{\tilde{u}_1}{\sqrt{n}}\mathbb{E}[a_1^2(f''(a_2 r)(f(a_2 r) - f(a_1)) + f'(a_2 r)^2)] + o(n^{-1/2})\right]$$

which recovers the limiting drift for $\tilde{u}_1$ as $n \to \infty$. For the diffusion term, recall the limiting volatility matrix from the proof of Proposition 3.5,

$$JVJ^\top = \begin{bmatrix} 4\mathrm{Var}[\Delta f'(s)a_1] & 8r\,\mathrm{Cov}(\Delta f'(s)a_1, \Delta f'(s)a_2) \\ 8r\,\mathrm{Cov}(\Delta f'(s)a_1, \Delta f'(s)a_2) & 16r^2\mathrm{Var}[\Delta f'(s)a_2] \end{bmatrix}$$

where we set $s := \langle x, g \rangle = ma_1 + ra_2$ and $\Delta := f(s) - f(a_1) + \varepsilon$. Evidently, under the rescaling only $\Sigma_{11}$ survives the $n \to \infty$ limit as reduces to

$$\Sigma_{11}(\tilde{u}_1, r) = 4\text{Var}\left[ (f(ra_2) - f(a_1) + \varepsilon)\, f'(ra_2)a_1 \right]$$

To simplify the variance, note that

$$\mathbb{E}\left[ (f(ra_2) - f(a_1) + \varepsilon)\, f'(ra_2)a_1 \right] = -\mathbb{E}[a_1 f(a_1)]\mathbb{E}[f'(ra_2)] = 0$$

where the reduction follows by independence of $a_1, a_2, \epsilon$ followed by our assumption that $\mathbb{E}[a_1 f(a_1)] = \mathbb{E}[f'(a_1)] = 0$ by Stein's lemma. Thus the limiting volatility coefficient as $n \to \infty$ is

$$\Sigma_{11}(\tilde{u}_1, r) = 4\mathbb{E}\left[ a_1^2 f'(ra_2)^2 \left( (f(ra_2) - f(a_1))^2 + \sigma^2 \right) \right]$$

which completes the proof. $\qquad\square$

The condition $\mathbb{E}_Z[f'(Z)] = 0$ is to ensure that the fixed point for which we are considering the diffusive limit exists along $m = 0$. This holds, for example, if $f$ is an even function such as in phase retrieval where $f(x) = x^2$.

*Proof of Proposition 3.10.* With Theorem D.1, we compute each of the coefficients using $f(x) = x^2$, $f'(x) = 2x$, and $f''(x) = 2$,

$$2\mathbb{E}\left[ a_1^2 \left( f''(ra_2)(f(ra_2) - f(a_1)) + f'(ra_2)^2 \right) \right] = 2\mathbb{E}[a_1^2(6r^2a_2^2 - 2a_1^2)] = 12(r^2 - 1)$$

$$4r\mathbb{E}\left[ a_2 f'(ra_2)(f(ra_2) - f(a_1)) \right] = 8r^2\mathbb{E}[a_2^2(r^2a_2^2 - a_1^2)] = 8r^2(3r^2 - 1)$$

and with $\sigma \to 0$,

$$4\mathbb{E}\left[ f'(ra_2)^2 \left( (f(ra_2) - f(a_1))^2 + \sigma^2 \right) \right] = 16r^2\mathbb{E}[a_2^2(r^2a_2^2 - a_1^2)^2] + 16r^2\sigma^2$$
$$= 16r^2(15r^4 - 6r^2 + 3 + \sigma^2)$$

$$4\mathbb{E}\left[ a_1^2 f'(ra_2)^2 \left( (f(ra_2) - f(a_1))^2 + \sigma^2 \right) \right] = 16r^2\mathbb{E}[a_1^2a_2^2(r^2a_2^2 - a_1^2)^2] + 16r^2\sigma^2$$
$$= 16r^2(15r^4 - 18r^2 + 15 + \sigma^2)$$

Substituting the quantities recovers the diffusive dynamics in equation 3.10. $\qquad\square$

## D.4 DYNAMICS FOR SGD-U (STRICTLY INCREASING LINK FUNCTIONS)

If $f$ is strictly monotonically increasing, then $f'(s) > 0$ almost everywhere (a.e.) and $\text{sgn}(f(t) - f(s)) = \text{sgn}(t - s)$.

**Lemma D.2.** Let $(X, Y)$ be jointly centered Gaussian variables. Then

$$\mathbb{E}[X\text{sgn}(Y)] = \sqrt{\frac{2}{\pi}} \frac{\text{Cov}(X, Y)}{\sqrt{\text{Var}(Y)}}$$

*Proof.* Consider the random variable $W = X - aY$ with constant $a = \text{Cov}(X, Y)/\text{Var}(Y)$. Since $(X, Y)$ is centered, we have $\mathbb{E}W = 0$. Moreover, $\text{Cov}(W, Y) = 0$ which implies that $W$ is independent of $Y$ as they are jointly Gaussian. By the towering property and independence, it follows that

$$\mathbb{E}\left[ X \mid Y \right] = \mathbb{E}[aY + R \mid Y] = aY = \frac{\text{Cov}(X, Y)}{\text{Var}(Y)}Y$$

Returning to the original quantity $\mathbb{E}[X\text{sgn}(Y)]$, we have again by towering that

$$\mathbb{E}[X\,\text{sgn}(Y)] = \mathbb{E}\left[ \mathbb{E}[X\,\text{sgn}(Y)] \mid Y \right] = \frac{\text{Cov}(X, Y)}{\text{Var}(Y)}\mathbb{E}|Y|$$

To complete the proof, write $Y = \text{Var}(Y) \cdot Z$ where $Z \sim \mathcal{N}(0, 1)$ and use the fact that $\mathbb{E}|Z| = \sqrt{2/\pi}$. $\qquad\square$

From here, we collect the following quantities

$$\text{Cov}\,(a_1, (1-m)a_1 - ra_2) = 1 - m, \quad \text{Cov}\,(a_2, (1-m)a_1 - ra_2) = -r,$$

$$\text{Var}\,((1-m)a_1 - ra_2) = (1-m)^2 + r^2$$

so that by Lemma D.2 we conclude that

$$\lim_{\sigma \to 0} \lim_{n \to \infty} \langle \nabla \tilde{\Phi}, \nabla m \rangle = \mathbb{E}[a_1 \cdot \text{sgn}\,((1-m)a_1 - ra_2)] = \sqrt{\frac{2}{\pi}} \frac{1-m}{\sqrt{(1-m)^2 + r^2}},$$

$$\lim_{\sigma \to 0} \lim_{n \to \infty} \langle \nabla \tilde{\Phi}, \nabla r^2 \rangle = 2r\,\mathbb{E}[a_2 \cdot \text{sgn}\,((1-m)a_1 - ra_2)] = -\sqrt{\frac{2}{\pi}} \frac{2r^2}{\sqrt{(1-m)^2 + r^2}}$$

## D.5 DYNAMICS FOR SGD-U (EVEN POWERED MONOMIALS)

To complement Proposition 3.9, we provide additional analysis for even-powered monomials (i.e. $f(x) = x^{2k}$ for $k \geq 1$) next. We first define the auxillary functions,

$$F(x, y) = \frac{x - 1}{\sqrt{(x-1)^2 + y^2}} + \frac{x + 1}{\sqrt{(x+1)^2 + y^2}} - \frac{x}{\sqrt{x^2 + y^2}}$$

$$G(x, y) = \frac{1}{\sqrt{(x-1)^2 + y^2}} + \frac{1}{\sqrt{(x+1)^2 + y^2}} - \frac{1}{\sqrt{x^2 + y^2}}$$

We show a similar universality result where the dynamics are again invariant with respect to the monomial's degree.

**Proposition D.3.** If $f$ is an even power polynomial (e.g. $f(x) = x^{2k}$ for $k \geq 1$), then with $\eta = \sqrt{n}/\|\nabla L\|$ we obtain the following universal dynamics,

$$\mathrm{d}m = -\frac{2}{\sqrt{2\pi}} F(m, r)\,\mathrm{d}t, \quad \mathrm{d}r^2 = -\frac{4r^2}{\sqrt{2\pi}} G(m, r) + c_\delta\,\mathrm{d}t$$

Of interest to us is a more refined analysis of the fixed points that approach the population optimum $(1, 0)$. As we cannot determine closed-form solutions for the system of equations in Proposition D.3, we instead obtain the following asymptotic expansion of fixed points in the regime where $r^2$ approaches zero.

**Proposition D.4.** The asymmetric fixed points $(m, r^2)$ of the dynamics in Proposition D.3 satisfy $F(m, r) = 0$ and $\frac{4r^2}{\sqrt{2\pi}} G(m, r) = c_\delta$, where $r = \sqrt{r^2}$. More precisely, we write

1. For $0 < c_\delta \leq 4/\sqrt{10\pi}$, these fixed points satisfy $|m| \in (\frac{1}{2}, 1)$ and $r^2 \in (0, 1]$.

2. Fix $0 < \gamma \leq 0.2$. Extend $c$ by $c(0) := 0$ and define

$$c(r) := \frac{4r^2}{\sqrt{2\pi}} G(m(r), r), \qquad C_\gamma := \max_{0 \leq r \leq \gamma} c(r).$$

   Then $C_\gamma > 0$, and for every $0 < c_\delta \leq C_\gamma$ there exists an asymmetric fixed point with $r \in (0, \gamma]$ and $m = \pm m(r)$. In particular, $1 - r^3 < m(r) < 1$. Moreover, as $c_\delta \downarrow 0$ (equivalently $k := c_\delta \sqrt{2\pi}/4 \downarrow 0$), the solutions obey

$$r = k + \frac{1}{2}k^2 + \frac{1}{2}k^3 + O(k^4), \qquad m = \pm \left( 1 - \frac{3}{8}r^3 + O(r^5) \right).$$

The first item in Proposition D.4 provides the existence of a threshold for $c_\delta$ such that for $c_\delta$ smaller than this threshold, we obtain fixed points away from $m = 0$. In particular, we find this threshold to be $4/\sqrt{10\pi} \approx 0.71$, which is larger than, for example, the $1/12$ threshold observed for phase retrieval in Proposition 3.7. We suspect a similar phenomenon to occur for more general polynomials.

The second item states that for $c_\delta$ satisfying another threshold constraint with upper bound $C_\gamma$, the system converges to a fixed point within some neighbourhood of the optimum $(1, 0)$. In particular, we have that for $\gamma \leq 0.2$, we can compute $C_\gamma \approx 0.286$ for the precise asymptotic expressions for $(m, r^2)$ to hold. Even if we did not take $\gamma$ in this range, we still have a larger range of admissible step-sizes for SGD-U for which we attain a desirable fixed point whereas online SGD is still at the equator (i.e. $1/12 < c_\delta < 4/\sqrt{10\pi}$ in the case of phase retrieval).

### D.5.1 PROOF OF PROPOSITION D.3

Let $f(x) = x^{2k}$ for $k \geq 1$. Applying the identity $a^k - b^k = (a - b) \sum_{r=0}^{\ell-1} a^{\ell-1-r} b^r$, we have that

$$t^{2k} - s^{2k} = (t - s)(t + s) \sum_{r=0}^{k-1} t^{2(k-1-r)} s^{2r}$$

where the series is a polynomial with strictly positive components for $t, s \neq 0$. Thus we can reduce the quantity inside the expectation again to get

$$\mathrm{sgn}(t^{2k} - s^{2k}) = \mathrm{sgn}((t - s)(t + s))$$

for $k \geq 1$, thus the following result is universal for any even-powered monomial since we have

$$S(s, t) = \mathrm{sgn}\left(f'(s)(f(t) - f(s))\right) = \mathrm{sgn}\left(s(t - s)(t + s)\right)$$

Recall that $t = a_1 = \langle a, v \rangle$ and $s = ma_1 + ra_2 = \langle a, mv + ru \rangle$ so that $s, t - s, t + s$ are linear in $(a_1, a_2)$. If we write $S$ as a function of $(a_1, a_2)$, we see that $S(a_1, a_2) = S(\rho a_1, \rho a_2)$ for any $\rho > 0$, thus it is a homogeneous function of degree zero. In particular, if we write the joiny vector $(a_1, a_2)$ in terms of polar coordinates $(a_1, a_2) = (\rho \cos\theta, \rho \sin\theta)$, we see that

$$S(a_1, a_2) = S(\rho\cos\theta, \rho\sin\theta) = S(\theta)$$

so that $S$ only depends on the angle $\theta$.

**Lemma D.5.** The inner expectations in the drift terms simplfy to

$$\mathbb{E}[a_1 S] = \frac{1}{\sqrt{8\pi}} \int_0^{2\pi} \cos\theta \, S(\theta) \, d\theta, \qquad \mathbb{E}[a_2 S] = \frac{1}{\sqrt{8\pi}} \int_0^{2\pi} \sin\theta \, S(\theta) \, d\theta.$$

*Proof.* We consider $\mathbb{E}[a_1 S]$ and the other integral follows by symmetry. Substituting the joint density of $(a_1, a_2)$ in polar coordinates (recall the Jacobian is the radius $\rho$) and noting that $S$ is bounded (to invoke Fubini-Tonelli), we have that

$$\mathbb{E}[a_1 S] = \frac{1}{2\pi} \int_{\mathbb{R}^2} a_1 S(a_1, a_2) \, e^{-(a_1^2 + a_2^2)/2} \, da_1 \, da_2$$

$$= \frac{1}{2\pi} \int_0^{2\pi} \int_0^\infty (\rho\cos\theta) S(\theta) \, e^{-\rho^2/2} \, \rho \, d\rho \, d\theta$$

$$= \frac{1}{2\pi} \left( \int_0^\infty \rho^2 e^{-\rho^2/2} \, d\rho \right) \left( \int_0^{2\pi} \cos\theta \, S(\theta) \, d\theta \right).$$

The first integral simplifies by the standard $\Gamma$-formula to $\sqrt{\pi/2}$. Thus the integrals reduce to

$$\mathbb{E}[a_1 S] = \frac{1}{2\pi} \sqrt{\frac{\pi}{2}} \int_0^{2\pi} \cos\theta \, S(\theta) \, d\theta = \frac{1}{\sqrt{8\pi}} \int_0^{2\pi} \cos\theta \, S(\theta) \, d\theta$$

as desired. $\square$

To further simplify the integrals, we consider the behaviour of $S(\theta)$. In particular we see when the sign changes as one of the three linear factors vanishes. These boundaries are solutions to the equations

$$\begin{cases} s = 0 & \Leftrightarrow m\cos\theta + r\sin\theta = 0, \\ t - s = 0 & \Leftrightarrow (1 - m)\cos\theta - r\sin\theta = 0, \\ t + s = 0 & \Leftrightarrow (1 + m)\cos\theta + r\sin\theta = 0. \end{cases}$$

and parameterize the auxillary tangent quantities with $\alpha_1, \alpha_2, \alpha_3 \in (0, \frac{\pi}{2})$ by,

$$\tan \alpha_1 = \frac{m}{r}, \quad \tan \alpha_2 = \frac{1-m}{r}, \quad \tan \alpha_3 = \frac{1+m}{r},$$

The solutions to the three systems gives two antipodal angles in $[0, 2\pi)$ (by also accounting for the quadrant they lie in to ensure the signs align with the system):

$$s = 0 \iff \theta \in \{\pi - \alpha_1, \ 2\pi - \alpha_1\},$$
$$t - s = 0 \iff \theta \in \{\alpha_2, \ \pi + \alpha_2\},$$
$$t + s = 0 \iff \theta \in \{\pi - \alpha_3, \ 2\pi - \alpha_3\}.$$

For simplicity we assume $0 < m < 1$ (we have previously established $r > 0$), however the same arguments hold for $-1 < m < 0$ in the case of even-powered monomials as only the ordering of angles will change, not their values. With this, ordering the angles yields

$$0 < \underbrace{\theta_0}_{=\alpha_2} < \underbrace{\theta_1}_{=\pi-\alpha_3} < \underbrace{\theta_2}_{=\pi-\alpha_1} < \underbrace{\theta_3}_{=\pi+\alpha_2} < \underbrace{\theta_4}_{=2\pi-\alpha_3} < \underbrace{\theta_5}_{=2\pi-\alpha_1} < 2\pi,$$

using the observation that $0 < \alpha_2 < \frac{\pi}{2}, 0 < \alpha_1 < \alpha_3 < \frac{\pi}{2}$ since $0 < \alpha_2 < \frac{\pi}{2}, 0 < \alpha_1 < \alpha_3 < \frac{\pi}{2}$ (by monotonicity of $\tan x$ for $x \in (0, \pi/2)$.

To determine the boundary behaviour, we see that at $\theta = 0^+$, we have $a_1 > 0, a_2 = 0^+$, hence $s > 0, t - s > 0, t + s > 0$, so $S = +1$ for small $\theta > 0$. Crossing a single boundary changes the sign of exactly one factor, hence flipping the overall sign as we have an odd number of quantities to consider. This exhibits a behaviour where the signs alternate across the intervals

$$(0, \theta_0), \ (\theta_0, \theta_1), \ (\theta_1, \theta_2), \ (\theta_2, \theta_3), \ (\theta_3, \theta_4), \ (\theta_4, \theta_5), \ (\theta_5, 2\pi).$$

The boundary sets themselves have Lebesgue measure-zero, and thus are not considered when computing the integral.

To compute the integrals using the alternating sign pattern (and simplifying the notation to illustrate the argument), we first see that

$$\int_0^{2\pi} \cos\theta \, S(\theta) \, d\theta = \left( \int_0^{\theta_0} - \int_{\theta_0}^{\theta_1} + \int_{\theta_1}^{\theta_2} - \int_{\theta_2}^{\theta_3} + \int_{\theta_3}^{\theta_4} - \int_{\theta_4}^{\theta_5} + \int_{\theta_5}^{2\pi} \right) \cos\theta \, d\theta$$

$$= \left[\sin\theta\right]_0^{\theta_0} - \left[\sin\theta\right]_{\theta_0}^{\theta_1} + \cdots + \left[\sin\theta\right]_{\theta_5}^{2\pi}$$

$$= 2\left(\sin\theta_0 - \sin\theta_1 + \sin\theta_2 - \sin\theta_3 + \sin\theta_4 - \sin\theta_5\right)$$

Substituting the values for $\theta_0$ with $i = 1, \ldots, 5$ yields the concrete integrals

$$\int_0^{2\pi} \cos\theta \, S(\theta) \, d\theta = 4\left(\sin\alpha_1 + \sin\alpha_2 - \sin\alpha_3\right)$$

$$\int_0^{2\pi} \sin\theta \, S(\theta) \, d\theta = 4\left(\cos\alpha_1 - \cos\alpha_2 - \cos\alpha_3\right)$$

Writing out the trigonometric ratioes in terms of $(m, r^2)$, we conclude that

$$\lim_{\sigma \to 0} \lim_{n \to \infty} \langle \nabla \tilde{\Phi}, \nabla m \rangle = \frac{2}{\sqrt{2\pi}} \left[ \frac{m-1}{\sqrt{(m-1)^2 + r^2}} + \frac{m+1}{\sqrt{(m+1)^2 + r^2}} - \frac{m}{\sqrt{m^2 + r^2}} \right]$$

$$\lim_{\sigma \to 0} \lim_{n \to \infty} \langle \nabla \tilde{\Phi}, \nabla r^2 \rangle = \frac{4r^2}{\sqrt{2\pi}} \left[ \frac{1}{\sqrt{(m-1)^2 + r^2}} + \frac{1}{\sqrt{(m+1)^2 + r^2}} - \frac{1}{\sqrt{m^2 + r^2}} \right]$$

which recovers the first-order drifts as desired.

### D.5.2 PROOF OF PROPOSITION D.4

We prove this in several steps, starting with the existence of fixed points and respective global bounds. The proof only requires elementary arguments from calculus.

**Lemma D.6.** For any fixed $r \in (0, 1]$, there exists a unique $m(r) \in (\frac{1}{2}, 1)$ such that $F(m(r), r) = 0$.

*Proof.* We analyze the function $m \mapsto F(m, r)$ on the interval $[\frac{1}{2}, 1]$. Starting with $\partial_m F$, we find that

$$\partial_m F(m, r) = \frac{r^2}{((m-1)^2 + r^2)^{3/2}} + \frac{r^2}{((m+1)^2 + r^2)^{3/2}} - \frac{r^2}{(m^2 + r^2)^{3/2}}$$

For $m \geq 1/2$, we have $(m-1)^2 \leq m^2$ which implies that $\partial_m F(m, r) > 0$. At the endpoints, we find that $F(1, r) > 0$ and $F(\frac{1}{2}, r) < 0$ for any $r^2 < 27/20$. By assumption that $r \in (0, 1]$, the existence of a unique root $m(\rho) \in (\frac{1}{2}, 1)$ follows by the Intermediate Value Theorem (IVT). $\square$

**Lemma D.7.** The function $c(r)$ given by

$$c(r) := \frac{4r^2}{\sqrt{2\pi}} G(m(r), r), \quad r \in (0, 1]$$

is continuous on $(0, 1]$, satisfies $c(1) \geq 4/\sqrt{10\pi}$, and $\lim_{r \downarrow 0} c(r) = 0$.

*Proof.* By Lemma D.6, we have that for any fixed $r_0 \in (0, 1]$, there exists some $m_0 = m(r_0)$ such that $F(m(r_0), r_0) = 0$ with $\partial_m F(m(r_0), r_0) > 0$. By the Implicit Function Theorem (IFT), it follows that $m(r)$ is $C^1$ on $(0, 1]$, thus $c(r)$ is continuous.

On the other hand, for $m(r) \in (1/2, 1)$ we have $(m-1)^2 < m^2$ which implies that

$$G(m(r), r) > \frac{1}{\sqrt{(m+1)^2 + r^2}} > \frac{1}{\sqrt{4 + r^2}} \implies c(r) > \frac{4r^2}{\sqrt{2\pi}\sqrt{4 + r^2}}$$

At the boundary $r = 1$, we have $c(1) > 4/\sqrt{10\pi}$. Finally at the limit $r \to 0$, it follows by the Squeeze Theorem and non-negativity of $c(r)$ that

$$0 \leq c(r) \leq \frac{4r^2}{\sqrt{2\pi}} \cdot \frac{1}{r} \to 0$$

as desired. $\square$

*Proof of Part (1).* Let $0 < c_\delta \leq 4/\sqrt{10\pi}$. By Lemma D.7, $c(1) \geq c_\delta$ and $\lim_{r \downarrow 0} c(r) = 0$. By IVT, there exists $r_* \in (0, 1]$ such that $c(r_*) = c_\delta$ where the corresponding $m_* = m(r_*)$ satisfying $m_* \in (\frac{1}{2}, 1)$ follows by Lemma D.6. $\square$

To prove (2), we first analyze the behaviour when $r$ is small, focusing on the regime $r \in (0, r_0]$ where $r_0 = 0.2$. Then we connect this asymptotic behaviour with the specific choice of $c_\delta$ as stated. The arguments remain elementary, albeit tedious.

**Lemma D.8.** For $r \in (0, 0.2]$, the unique root $m(r)$ satisfies $1 - r^3 < m(r) < 1$.

*Proof.* We analyze $F(m, r)$ on the interval $I = [1 - r^3, 1]$. We first note that the function

$$h(s) = \left(1 + \frac{s}{4}\right)^{-1/2} - (1 + s)^{-1/2}$$

is concave on the interval $s \in (0, 1]$ since $h''(s) \leq 0$. In particular we have that $h(s) \leq h(0) + h'(0)s$ where $h(0) = 0$ and $h'(0) = 3/8$. Thus, we obtain the upper bound $F(1, r) = h(r^2) \leq \frac{3}{8}r^2$.

On the interval $m \in I$, we have $|m - 1| \leq r^3$ so that $\partial_m F$ is bounded below by

$$\partial_m F(m, r) \geq \frac{1}{r(1 + r^4)^{3/2}} - r^2$$

by bounding the first term in $\partial_m F$, dropping the second term, and noting that for the third term that

$$m^2 + r^2 \geq (1 - r^3) + r^2 = 1 + r^2(1 - 2r + r^4) \geq 1$$

for $r \leq 0.2$. Thus for $r \leq 0.2$ we can establish that $\partial_m F(m, r) \geq 1/(2r) - r^2$. To conclude, by the Mean Value Theorem for some $\xi \in I$ we have

$$F(1 - r^3, r) = F(1, r) - \partial_m F(\xi, r) \cdot r^3$$

where we substitute the previous bounds to get that

$$F(1 - r^3, r) \leq \frac{3}{8}r^2 - \left(\frac{1}{2r} - r^2\right)r^3 = -\frac{1}{8}r^2 + r^5 < 0 \quad \text{for } r \leq 0.2$$

Thus since $F(1 - r^3, r) < 0$ and $F(1, r) > 0$, the conclusion follows by IVT. □

**Lemma D.9.** As $r \downarrow 0$,

$$m(r) = 1 - \frac{3}{8}r^3 + O(r^5), \qquad c(r) = \frac{4}{\sqrt{2\pi}}\left(r - \frac{1}{2}r^2 + O(r^4)\right).$$

Consequently, with $k := c(r)\sqrt{2\pi}/4$, the inverse admits

$$r(k) = k + \frac{1}{2}k^2 + \frac{1}{2}k^3 + O(k^4).$$

*Proof.* We write $\delta := 1 - m(r)$ where $F(m(r), r) = 0$ by Lemma D.7. By Lemma D.8, for $r \in (0, 0.2]$ one has

$$1 - r^3 < m(r) < 1 \implies 0 \leq \delta \leq r^3 = O(r^3),$$

so that on the interval $I(r) := [1 - r^3, 1]$, we obtain $F_m(m, r) \geq 1/(2r)$ for $r \leq 0.2$. Using a similar MVT argument as the previous lemma, we write

$$F(1, r) - F(1 - \delta, r) = F(1, r) = F_m(\xi, r)\delta \implies \delta = \frac{F(1, r)}{F_m(\xi, r)}$$

for some $\xi \in (1 - \delta, 1) \subset I(r)$. Taking the expansion of $F(1, r)$ as $r \to 0$, we have

$$F(1, r) = \left(1 + \frac{r^2}{4}\right)^{-1/2} - \left(1 + r^2\right)^{-1/2} = \frac{3}{8}r^2 + O(r^4)$$

Next, we similarly expand $F_m(\xi, r)$ noting that $|\xi - 1| \leq \delta = O(r^3)$. For the first term, we have

$$\frac{r^2}{((\xi - 1)^2 + r^2)^{3/2}} = \frac{r^2}{r^3\left(1 + O(r^4)\right)^{3/2}} = \frac{1}{r} + O(r^3)$$

while the remaining two terms in $F_m(\xi, r)$ are $O(r^2)$ uniformly in $\xi \in I(r)$:

$$0 \leq \frac{r^2}{((\xi + 1)^2 + r^2)^{3/2}} \leq r^2, \qquad \left|-\frac{r^2}{(\xi^2 + r^2)^{3/2}}\right| \leq r^2 \quad (\text{since } \xi^2 + r^2 \geq 1).$$

Putting all of the terms together yields the asymptotic expansion

$$F_m(\xi, r) = \frac{1}{r} + O(r^2) \qquad \text{uniformly for } \xi \in I(r). \tag{22}$$

where substituting it back to the quotient representation for $\delta$ yields,

$$\delta = \left(\frac{3}{8}r^2 + O(r^4)\right)(r + O(r^4)) = \frac{3}{8}r^3 + O(r^5) \implies m(r) = 1 - \delta = 1 - \frac{3}{8}r^3 + O(r^5)$$

Next to obtain the expansion of $c(r)$, we first expand $G$ at $m = 1$:

$$G(1, r) = \frac{1}{r} + \frac{1}{\sqrt{4 + r^2}} - \frac{1}{\sqrt{1 + r^2}} = \frac{1}{r} - \frac{1}{2} + O(r^2)$$

By MVT, for some $\zeta \in (1 - \delta, 1) \subset I(r)$ with $\delta$ as defined before gives

$$G(m(r), r) - G(1, r) = \partial_m G(\zeta, r)\,(m(r) - 1)$$

We can show that $\partial_m G(\zeta, r) = O(1)$ uniformly on $I(r)$ by observing that

$$\left| \frac{m - 1}{((m - 1)^2 + r^2)^{3/2}} \right| \leq \frac{r^3}{r^3} = 1, \quad \frac{m + 1}{((m + 1)^2 + r^2)^{3/2}} \leq \frac{1}{(m + 1)^2} \leq 1$$

$$\frac{m}{(m^2 + r^2)^{3/2}} \leq \frac{1}{m^2} \leq 2$$

where we also recall that $m \geq 1 - r^3 \geq \frac{1}{2}$. Hence $|\partial_m G| \leq 4$ uniformly and thus

$$G(m(r), r) = G(1, r) + O(\delta) = \frac{1}{r} - \frac{1}{2} + O(r^2) \implies c(r) = \frac{4}{\sqrt{2\pi}}\left( r - \frac{1}{2}r^2 + O(r^4) \right)$$

To show the final claim about inversion, let $k := c(r)\sqrt{2\pi}/4$ and note that $c'(r) > 0$ as $r \to 0$. Following our expansion for $c(r)$, we write

$$k = r - \frac{1}{2}r^2 + O(r^4)$$

and revert the series up to the leading terms, say $r = r(k) = k + a_2 k^2 + a_3 k^3 + O(k^4)$. To do so, we simply substitute and match powers to obtain that

$$k = (k + a_2 k^2 + a_3 k^3) - \frac{1}{2}(k + a_2 k^2)^2 + O(k^4) = k + \left( a_2 - \frac{1}{2} \right) k^2 + (a_3 - a_2)\,k^3 + O(k^4),$$

which implies that $a_2 = a_3 = \frac{1}{2}$ which gives

$$r(k) = k + \frac{1}{2}k^2 + \frac{1}{2}k^3 + O(k^4). \quad \square$$

*Proof of (2).* Let $0 < \gamma \leq 0.2$. By Lemma D.6, D.7, and the extension of $c(0) := 0$, $c$ is continuous on the compact interval $[0, \gamma]$, hence attains a maximum $C_\gamma = \max_{[0,\gamma]} c(r)$.

By Lemma D.9, we have the asymptotic expansion as $r \to 0$,

$$c(r) = \frac{4}{\sqrt{2\pi}}\left( r - \frac{1}{2}r^2 + O(r^4) \right)$$

which again verifies that $c(r) > 0$ for sufficiently small $r > 0$, in particular $C_\gamma > 0$.

Fix $c_\delta \in (0, C_\gamma]$ and choose $r_\gamma \in [0, \gamma]$ with $c(r_\gamma) = C_\gamma$. Since $c$ is continuous and $c(0) = 0$, by IVT there exists $r_* \in (0, r_\gamma) \subset (0, \gamma]$ such that $c(r_*) = c_\delta$. Set $m_* := m(r_*)$, then by construction

$$F(m_*, r_*) = 0, \qquad \frac{4r_*^2}{\sqrt{2\pi}}\,G(m_*, r_*) = c(r_*) = c_\delta,$$

so $(\pm m_*, r_*^2)$ are fixed points. Furthermore, since $r_* \leq \gamma \leq 0.2$, Lemma D.8 implies $1 - r_*^3 < m_* < 1$.

Finally, because $c(r) \to 0$ as $r \downarrow 0$ it follows that $c_\delta \downarrow 0$ asymptotically as well. By Lemma D.9, we obtain the expansions,

$$m(r) = 1 - \frac{3}{8}r^3 + O(r^5), \qquad c(r) = \frac{4}{\sqrt{2\pi}}\left( r - \frac{1}{2}r^2 + O(r^4) \right),$$

which the local inversion

$$r(k) = k + \frac{1}{2}k^2 + \frac{1}{2}k^3 + O(k^4), \quad k = \frac{\sqrt{2\pi}}{4}\,c_\delta,$$

which together yield the claimed asymptotics for $(m_*, r_*)$ as $c_\delta \to 0$. $\qquad \square$

# E  TECHNICAL PROOFS OF AUXILLARY RESULTS

## E.1  NORMS OF GAUSSIAN VECTORS

**Lemma E.1.** Let $X \sim \mathcal{N}(\mu, \Sigma)$ be an $n$-dimensional Gaussian vector with $\Sigma \succ 0$ and $\|\Sigma\|_{\mathrm{op}} < \infty$ (i.e. $X$ is non-degenerate). If $\|\mu\| = O(\sqrt{n})$, then the cumulants $\kappa_p$ of $\|X\|^2$ are also of the order $O(n)$ for each fixed $p \geq 1$.

*Proof.* Noting that $\|X\|^2 = X^\top X$ is a quadratic form, the cumulants can be determined exactly, for example see (Magnus, 1986, Lemma 2):

$$\kappa_p = 2^{p-1}(p-1)! \left\{ \mathrm{tr}(\Sigma^p) + p\langle \mu, \Sigma^{p-1}\mu\rangle \right\}$$

Using the assumption that $\|\mu\| = O(\sqrt{n})$ and $\mathrm{tr}(\Sigma^k) = O(n)$ for any $k \geq 1$ completes the proof by assumptions on the spectrum. $\qquad\square$

**Lemma E.2.** Let $X \sim \mathcal{N}(\mu, \Sigma)$ be an non-degenerate $n$-dimensional Gaussian vector. If $n > 2k$, then $\mathbb{E}\|X\|^{-2k} < \infty$.

*Proof.* Denote $\gamma_X$ as the corresponding density of $X$. It suffices to demonstrate integrability near the origin since,

$$\mathbb{E}[\|X\|^{-2k} \, \mathbb{1}_{\{\|X\| \geq r\}}] = \int_{\|x\| \geq r} \|x\|^{-2k} \, \gamma_X(x)\, \mathrm{d}x \leq \frac{1}{r^{2k}} < \infty, \quad \text{for } r > 0$$

To this end, consider the restricted expectation over the open ball $\mathbb{B}_r(0)$ for $r > 0$. By continuity and positivity of $\gamma_X$ on $\mathbb{B}_r(0)$,

$$\int_{\mathbb{B}_r(0)} \|x\|^{-2k} \, \gamma_X(x)\, \mathrm{d}x \leq \|\gamma_X\|_\infty \int_{\mathbb{B}_r(0)} \|x\|^{-2k} \, \mathrm{d}x$$

As $\|x\|$ is a radial function, by the *coarea formula* (e.g. see (Stromberg, 2015, pg. 370)) the remaining integral becomes

$$\int_{\mathbb{B}_r(0)} \|x\|^{-2k} \, \mathrm{d}x = |\mathbb{S}^{n-1}| \int_0^r \rho^{n-1-2k} \, \mathrm{d}\rho$$

where $|\mathbb{S}^{n-1}|$ is the surface measure of the $(n-1)$-dimensional unit sphere. Noting that the integral is finite only if $n > 2k$ completes the proof. $\qquad\square$

**Lemma E.3.** Let $X \sim \mathcal{N}(\mu, \Sigma)$ in $\mathbb{R}^n$ with $\Sigma \succ 0$. Then for any integer $k \geq 1$ with $n > 2k$,

$$\mathbb{E}\|X\|^{-2k} \leq \lambda_{\min}(\Sigma)^{-k} \left( \frac{1}{n - 2k} \right)^k$$

In particular, if $\lambda_{\min}(\Sigma) \geq m > 0$ uniformly in $n$, then $\mathbb{E}\|X\|^{-2k} = O(n^{-k})$.

*Proof.* Let $Z \sim \mathcal{N}(0, I_n)$ be a standard Gaussian vector. We first determine a comparison inequality involving $\mathbb{E}\|X\|^{-2k}$ for $\mu = 0$, and show that the upper bound is still applicable $\mu \neq 0$ by Anderson's inequality Anderson (1955). In particular, we have that for the centered Gaussian density $\gamma_\Sigma$ and a origin-symmetric, convex set $K \subset \mathbb{R}^n$

$$\int_K \gamma_\Sigma(x - \mu)\, \mathrm{d}x = \int_{K+\mu} \gamma_\Sigma(x)\, \mathrm{d}x \leq \int_K \gamma_\Sigma(x)\, \mathrm{d}x \tag{23}$$

Proceeding with the argument, if $\mu = 0$, we write $X = \Sigma^{1/2}Z$ and obtain the comparison inequality

$$\sqrt{\lambda_{\min}(\Sigma)}\|Z\| \leq \|X\| \leq \sqrt{\lambda_{\max}(\Sigma)}\|Z\|.$$

Raising both sides to the power $2k$ for $k \geq 1$ and applying expectations yields

$$\lambda_{\max}(\Sigma)^{-k}\mathbb{E}\|Z\|^{-2k} \leq \mathbb{E}\|X\|^{-2k} \leq \lambda_{\min}(\Sigma)^{-k}\mathbb{E}\|Z\|^{-2k} \tag{24}$$

On the other hand if $\mu \neq 0$, let $Y \sim \mathcal{N}(0, \Sigma)$ be a centered version of $X$ and denote $\gamma_\Sigma(y)$ as the Gaussian density of $Y$. Note that the density of $X$ is $\gamma_\Sigma(x - \mu)$. Then we write

$$\mathbb{E}[\|X\|^{-2k}] = \int_{\mathbb{R}^n} \|x\|^{-2k} \gamma_\Sigma(x - \mu) \, dx = \int_{\mathbb{R}^n} \int_0^\infty \mathbb{1}_{\{x:\|x\| < t^{-1/(2k)}\}} \, \gamma_\Sigma(x - \mu) \, dt \, dx$$

by "layer-cake" representation. Note that the inner integral is over the centered Euclidean ball

$$B(0, t^{-1/(2k)}) = \{x : \|x\| < t^{-1/(2k)}\}$$

which is symmetric and convex. Furthermore, by verified integrability in Lemma E.2 for $n > 2k$, we invoke Fubini-Tonelli to write

$$\mathbb{E}[\|X\|^{-2k}] = \int_0^\infty \int_{\mathbb{R}^n} \mathbb{1}_{B(0, t^{-1/(2k)})} \, \gamma_\Sigma(x - \mu) \, dx \, dt = \int_0^\infty \int_{B(0, t^{-1/(2k)})} \gamma_\Sigma(x - \mu) \, dx \, dt$$

By equation 23, the inner integral can be bounded (and further simplified by reversing the layer-cake arguments above),

$$\mathbb{E}[\|X\|^{-2k}] = \int_0^\infty \int_{B(0, t^{-1/(2k)})} \gamma_\Sigma(x - \mu) \, dx \, dt \leq \int_0^\infty \int_{B(0, t^{-1/(2k)})} \gamma_\Sigma(x) \, dx \, dt = \mathbb{E}[\|Y\|^{-2k}]$$

Thus the upper bound in equation 24 still holds as the analysis applies to the centered $Y$.

It remains to determine the asymptotic rate for $\mathbb{E}[\|Z\|^{-2k}]$. Since $\|Z\|^2$ follows a $\chi_n^2$-distribution and $n > 2k$ with $k \geq 1$, the inverse moments are given by Soch et al. (2023)

$$\mathbb{E}[\|Z\|^{-2k}] = 2^{-k} \frac{\Gamma\left(\frac{n}{2} - k\right)}{\Gamma(\frac{n}{2})} = 2^{-k} \prod_{i=1}^k \frac{2}{n - 2i} \leq \left(\frac{1}{n - 2k}\right)^k$$

Combining this bound with equation 24 completes the proof. $\square$

**Lemma E.4.** Consider an $n$-dimensional non-degenerate Gaussian vector $X \sim \mathcal{N}(\mu, \Sigma)$ such that $\text{tr}(\Sigma) = \Theta(n)$ and $\|\mu\| = o(n)$. Then for $n \geq 9$,

$$\mathbb{E}\left[\left(1 - \frac{\mathbb{E}[\|X\|^2]}{\|X\|^2}\right)^3\right] = O(n^{-2}).$$

with $\mathbb{E}[\|X\|^2] = \text{tr}(\Sigma) + \|\mu\|^2$.

*Proof.* To simplify notation, put $Q = \|X\|^2$ and $a = \mathbb{E}Q = \text{tr}(\Sigma) + \|\mu\|^2$ and write $\delta = (Q - a)/a$, so $(1 - a/Q)^3 = (\delta/(1 + \delta))^3$. To control the expectation around the singularity $\delta = -1$, split the expectation on the events

$$A = \{\delta \geq -1/2\}, \qquad B = \{\delta < -1/2\} = \{Q \leq a/2\}.$$

where we slightly abuse notation to incorporate the null event $\{Q = a/2\}$ in $B$. For $x \geq -1/2$, we consider the bound

$$\left|\left(\frac{x}{1 + x}\right)^3 - x^3\right| = |x^3| \cdot |(1 + x)^{-3} - 1| \leq |x|^3 \cdot \frac{3|x|}{(1 - 1/2)^4} = 48|x|^4$$

by the mean-value theorem applied to $t \mapsto (1 + t)^{-3}$. Thus using the expansion above, we write the expectation over $A$ as

$$\mathbb{E}\left[\left(\frac{\delta}{1 + \delta}\right)^3 \mathbb{1}_A\right] = \mathbb{E}[\delta^3 \mathbb{1}_A] + O\left(\mathbb{E}[|\delta|^4 \mathbb{1}_A]\right)$$

Now $\mathbb{E}\delta^3 = \mu_3(Q)/a^3$ and $\mathbb{E}|\delta|^4 = \mu_4(Q)/a^4$. By Lemma E.1, we can determine the order of the central moments through the cumulants,

$$\mu_3(Q) = \kappa_3(Q) = O(n), \qquad \mu_4(Q) = \kappa_4(Q) + 3\kappa_2(Q)^2 = O(n^2).$$

Since $a \geq cn$ for some $c > 0$ by assumption, we conclude for the first term in the partitioned expectation that

$$\mathbb{E}\left[\left(\frac{\delta}{1+\delta}\right)^3 \mathbf{1}_A\right] = \frac{O(n)}{(cn)^3} + O\left(\frac{O(n^2)}{(cn)^4}\right) = O(n^{-2})$$

It remains to control the tail event $B$. When $Q \leq a/2$, it follows that $Q \leq a$ and $Q - a \leq 0$ so that

$$\left|\left(1 - \frac{a}{Q}\right)^3\right| = \left|\frac{Q-a}{Q}\right|^3 \leq \left(\frac{a}{Q}\right)^3.$$

where the last inequality follows as $Q \geq 0$ by definition. By Lemma E.2 with $n \geq 9$ and Hölder's inequality with exponents $p = 4$, $q = 4/3$,

$$\mathbb{E}\left[\left(\frac{a}{Q}\right)^3 \mathbf{1}_B\right] \leq a^3 \left(\mathbb{E}Q^{-4}\right)^{3/4} \mathbb{P}(B)^{1/4}.$$

We bound each factor individually. First, note that $a^3 \leq (\operatorname{tr}\Sigma + \|\mu\|^2)^3 = O(n^3)$. Second, we have by Lemma E.3 that $\mathbb{E}Q^{-4} = O(n^{-4})$ so that $(\mathbb{E}Q^{-4})^{3/4} + O(n^{-3})$.

To bound the final term $\mathbb{P}(B) = \mathbb{P}(Q \leq a/2)$, we use a left-tail Chernoff argument

$$\mathbb{P}(Q \leq (1-\varepsilon)a) \leq \exp\left(t(1-\varepsilon)a + K_Q(-t)\right), \quad \text{for any } t > 0, \, \varepsilon \in (0,1).$$

where the cumulant generating function $K_Q(-t)$ is given by,

$$K_Q(-t) = -\frac{1}{2}\log\det(I + 2t\Sigma) - t\mu^\top(I + 2t\Sigma)^{-1}\mu$$

Expanding the log determinant and noting that $\log(1+x) \geq x - x^2/2$ for $x \geq 0$, we obtain the upper bound on the cgf

$$K_Q(-t) \leq -\frac{1}{2}\log\det(I + 2t\Sigma) = -\frac{1}{2}\sum_{i=1}^n \log(1 + 2t\lambda_i) \leq -t\operatorname{tr}(\Sigma) + t^2\operatorname{tr}(\Sigma^2)$$

as the trailing term is a quadratic form over a positive definite matrix. Applying the cgf bound on the tail probability, we have that

$$\exp\left(t(1-\varepsilon)a + K_Q(-t)\right) \leq \exp\left(-\varepsilon t \operatorname{tr}(\Sigma) + t^2\operatorname{tr}(\Sigma^2) + t(1-\varepsilon)\|\mu\|^2\right)$$

recalling that $a = \|\mu\|^2 + \operatorname{tr}(\Sigma)$. To obtain the desired decay rate, set $t = \varepsilon\operatorname{tr}(\Sigma)/(2\operatorname{tr}(\Sigma^2))$ hence,

$$-\varepsilon t \operatorname{tr}(\Sigma) + t^2\operatorname{tr}(\Sigma^2) = -\frac{\varepsilon^2(\operatorname{tr}\Sigma)^2}{4\operatorname{tr}(\Sigma^2)} \leq -\frac{\varepsilon^2}{4C}\operatorname{tr}(\Sigma) \leq -c'n$$

for an absolute constant $c > 0$ where the penultimate inequality follows by the von-Neumann trace inequality

$$\operatorname{tr}(\Sigma^2) \leq \|\Sigma\|_{\mathrm{op}}\operatorname{tr}(\Sigma) \leq C\operatorname{tr}(\Sigma)$$

using $\|\Sigma\|_{\mathrm{op}} \leq C$ and $\operatorname{tr}(\Sigma) \geq c'n$. Finally, noting that the remaining term satisfies $t(1-\varepsilon)\|\mu\|^2 = O(1)$, we obtain the desired exponential decay $\mathbb{P}(B) \leq Ke^{-c'n}$ for some constant $K < \infty$ independent of $n$. Combining, $\mathbb{E}[((a/Q) - 1)^3 \mathbf{1}_B] \leq O(e^{-c'n})$ so that altogether,

$$\mathbb{E}\left[\left(1 - \frac{a}{Q}\right)^3\right] = \mathbb{E}\left[\left(\frac{\delta}{1+\delta}\right)^3 \mathbf{1}_A\right] + \mathbb{E}\left[\left(\frac{\delta}{1+\delta}\right)^3 \mathbf{1}_B\right] = O(n^{-2}),$$

as claimed. $\qquad\square$

