# OpenReview forum: "High-dimensional limit theorems for SGD: Momentum and Adaptive Step-sizes"
_ICLR.cc/2026/Conference — ICLR 2026 Poster_

### Official Review · Reviewer_WtAq · 2025-10-28

**Soundness:** 4
**Presentation:** 2
**Contribution:** 3
**Rating:** 6
**Confidence:** 1

**Summary:**

This paper extends the high-dimensional limit theory of Ben Arous et al. (2022, 2024) to analyze Stochastic Gradient Descent (SGD) with Polyak momentum (SGD-M) and adaptive step-size variants (including a normalized-gradient variant, denoted SGD-U). The authors derive the asymptotic stochastic differential equations governing the evolution of summary statistics of SGD-M and adaptive SGD, under suitable regularity assumptions (delta-localizability and delta-closability). The theory establishes an equivalence between SGD-M and online SGD up to a time rescaling, and provides rigorous justification for the stabilizing role of gradient normalization in high-dimensional regimes. These results are illustrated on two canonical problems: spiked tensor PCA and single-index models, supported by both analytical fixed-point analyses and simulations.

**Strengths:**

I would like to note that I am not a specialist in stochastic differential equations or high-dimensional diffusion limits. While I have done my best to assess the work carefully, my understanding of some of the deeper technical aspects is limited.

1. The paper presents a non-trivial and technically rigorous generalization of existing high-dimensional diffusion limit results to momentum and adaptive-step-size algorithms. Extending the practical dynamics framework of Ben Arous et al. to handle additional algorithmic components is a clear and meaningful step forward. The identification of the equivalence between SGD-M and SGD via time rescaling (Theorem 2.3 and Remark 2.4) is elegant and provides theoretical insight on known empirical observations.

2. The analysis of normalized-gradient SGD (SGD-U) offers a rigorous interpretation of empirical stabilization mechanisms (e.g., gradient clipping, normalization), bridging stochastic process theory and optimization practice.

**Weaknesses:**

1. The paper is extremely technical and assumes strong familiarity with high-dimensional probability, weak convergence, and diffusion processes. While this is common in this line of work, some readers may find it challenging to connect the formal definitions. For instance, the concepts of delta-localizability and delta-closability are poorly described

2. I am not entirely sure about this claim, but it seems he localizability and closability conditions are mathematically strong (requiring smoothness up to third derivatives and bounded high-order moments). It seems to me that this condition is basically assuming the algorithm’s stochastic trajectories converge to an SDE, which compresses both randomness and nonlinearity into a self-contained stochastic differential equation. These may hold for the idealized models considered but might limit its applicability to more realistic scenarios (where similar behavior of SGDM and SGD have been observed, like in deep learning).

3. I might have missed this part, but it seems the authors does not provide examples of problem that satisfies delta-localizability and delta-closability. Moreover, it seems these are never explicitly verified for either of the two worked examples (spiked tensor PCA or single-index models).

**Questions:**

1. Is it possible to verify, even heuristically, that the paper's assumption are satisfied in one of the examples, or provide sufficient conditions under which they hold?

---

> ### Author Response · Authors · 2025-11-19
>
> To reviewer WtAq,
>
> Thank you for taking the time to give our paper a thoughtful review. We hope to address some of your concerns in what follows.
>
> ---
>
> **Weaknesses:** You have mentioned that the delta-localizability and asymptotic closability conditions are not well described or motivated. Similar conditions appeared in [1] along with some discussion in terms of intuition. Given the discussion in [1], we opted not to include it here for the sake of brevity. However, you have brought to our attention that we did not make this clear for readers who are not already familiar with [1]. To fix this, we have added a brief discussion after these definitions to point the reader to the discussion in [1] and highlight key points.
>
> We would also like to note here that these conditions are substantially weaker than those commonly used in the literature (e.g., $L$-smoothness or $K$-lipschitz conditions) and can be easily verified for broad classes of models commonly studied in the learning theory community. (We have now pointed this out in our revision.) In particular, these are conditions on the moments of directional derivatives of the loss only in a fixed number of directions (specifically the directions of summary statistics). Furthermore, we do not ask that these moments are bounded, but only that they do not diverge too rapidly. Another notable example which falls in our framework which we do not discuss in our work is supervised classification of the Gaussian XOR problem via a two layer network. In [1], the authors demonstrate that the localizability and closability conditions are satisfied with a two layer neural network with ReLU activations, and a similar argument applies to our conditions (which are essentially theirs + one additional term, see also Question below).
>
> We hope that this brings some clarity. We are happy to include more discussion if you feel it is warranted.
>
> ---
>
> **Question:**
> Please note that we did indeed verify these conditions for both the examples we worked out in the paper. The conditions are verified in the appendices as part of the proofs for tensor PCA (see Appendix C.1) and single index models (see Appendix D.1). Most of these conditions were already verified in other cited works (for example [1] and [2]), and the additional condition for SGD-M follows by the same argument. We also show that these conditions still hold for SGD-U and direct the reader to the proofs section in the appendix for both Tensor PCA and single index models.
>
> [1] Gerard Ben Arous, Reza Gheissari, and Aukosh Jagannath. High-dimensional limit theorems for SGD: Effective dynamics and critical scaling. Communications on Pure and Applied Mathematics, 77(3):2030–2080, 2024
>
> [2] Parsa Rangriz. High-dimensional scaling limits of online stochastic gradient descent in single-index models. University of Waterloo Thesis, 2025.

---

### Official Review · Reviewer_L1BR · 2025-10-31

**Soundness:** 2
**Presentation:** 1
**Contribution:** 3
**Rating:** 2
**Confidence:** 3

**Summary:**

This paper extends the high-dimensional effective dynamics of summary statistics framework of online SGD in [1] to online SGD with Polyak momentum (SGD-M) and online SGD with adaptive step-sizes. Based on these results, the authors looked at two applications: the spiked tensor PCA and single index models. In these applications, the authors

(1) consider a specific online SGD with adaptive step-sizes defined by normalizing gradient, referred as SGD-U;

(2) derive the ballistic limits of the summary statistics of SGD-M and SGD-U and study the fixed points in different regimes;

(3) derive the diffusive limits of the summary statistics of SGD-U at the equator in spiked tensor PCA.

[1] Gerard Ben Arous, Reza Gheissari, and Aukosh Jagannath. *High-dimensional limit theorems for
SGD: Effective dynamics and critical scaling*. Communications on Pure and Applied Mathemat-
ics, 77(3):2030–2080, 2024.

**Strengths:**

1. The paper provides a **valuable generalization** of high-dimensional diffusion limits to include both momentum and adaptive step-size variants of SGD. This extension broadens the understanding of optimization dynamics beyond standard online SGD, offering a unified framework for several widely used algorithms.

2. The main theorems and assumptions are rigorously stated and internally consistent.

**Weaknesses:**

1. The manuscript does not clearly highlight its key novelty—the extension from online SGD to momentum and adaptive-step methods. The relation to existing online SGD results is underdeveloped theoretically, which makes the contribution appear incremental despite being substantial.

2. The effective limits of SGD-U is difficult to follow. Neither a general limiting theorem or a comparison to SGD/SGD-M is provided.

3. The paper’s structure could better emphasize the main theoretical message. Sections on motivation, scaling intuition, and comparison to prior results are brief, while technical proofs dominate early sections, making it hard to identify the central takeaway.

**Questions:**

1. How does $\beta$ affect the $\delta_n$-localizable condition defined in Definition 2.1?

2. in line 322-323, the coefficient of the diffusion term is incorrect;

3. Why do you scaling by $\sqrt{n}$ in SGD-U? As explained in the paper, if $\lVert \nabla L_n \rVert=O(\sqrt{n})$, wouldn't the adaptive step-size of constant order, which contradicts to that we want the step-size tend to zero?

4. When is there a diffusive limit in the single index model example?

5. Please double check for typos. Some that I noticed are listed below:

    (1) $\mu$ in line095 should be $P_n$;

    (2) $\nabla L$ in equation (1) should be $\nabla L_n$;

    (3) in line 230: $R^2$ should be $m^2+r_\perp^2$;

    (4) $r$ and $r_\perp$ are the same. Please be consistent in notations.

---

> ### Author Response · Authors · 2025-11-19
>
> To reviewer L1BR,
>
> Firstly we would like to thank you for your time spent reviewing our work. We appreciate that you believe our work provides a substantial and valuable contribution to the current literature. With that said, we have interpreted your concerns to be primarily centered around presentation. We address these below.
>
> **Regarding Weakness 1:** We have revised our Introduction and Section 1.1 (Contributions). We added additional citations which we feel we missed in the original draft. Furthermore, we have tried to do a better job of highlighting where our work stands in the greater literature and make our contributions clear so that it is more apparent to the reader how substantial our contributions are. We hope that this addresses your concern. Please let us know if you feel there are specific works that we are failing to mention.
>
> **Regarding Weakness 2:** We have included a section in the appendix to more clearly address how we are able to rigorously derive the effective dynamics for SGD with (Markov) adaptive step-sizes, of which SGD-U is one example. We hope that this will clear up any confusion for future readers. As the equations are abstract, it is difficult to make clear, general comparisons between SGD-U and SGD/SGD-M. Instead, we decided to demonstrate these differences by comparing the algorithms in concrete examples.
>
> The primary discussion of Section 3 is concerned with highlighting how SGD-U can significantly alter the training dynamics relative to SGD-M. In particular, we observe that SGD-U is able to converge to desirable solutions with relative ease in scenarios where the training dynamics under SGD-M are unstable. This is consistent with observations and motivating heuristics that are discussed in the empirical literature.
>
> **Regarding Weakness 3:** We acknowledge that the introductory and motivation sections are brief.
> We revised Section 1.1 (Contributions) to clearly state the main takeaway of our results and examples in less technical language. To the best of our knowledge, there are no proofs included in the main body of the paper.
> We have elected to follow the style common in mathematics papers, and as such a significant portion of the paper is devoted to the technical set-up so that the results can be stated in a clear and mathematically rigorous fashion.
> If you have further concrete suggestions, please let us know.
>
> **Regarding your questions:**
>
> 1. As you pointed out, we adjusted the definition of $\delta_n$-localizability to exclude $\beta$ in the tuple. To avoid confusion, we revised the defining term to "augmented"-localizability to clearly distinguish this from similar definition(s) for online SGD.
>
> 2. Thank you for pointing this out. This was a typo in the constant factor within the coefficient(s) of the SDE. This is fixed in the revision (see the followup in our response to Question 5 below). This does not affect any of results or conclusions.
>
> 3. To clarify here, the step-size for the algorithm under SGD-U is $\delta_n \eta_n = \delta_n \frac{\sqrt{n}}{\|\nabla L_n\|}$. The adaptive component of the step-size is indeed of constant order with respect to the scaling, however the $\delta_n$ is the component of the step-size that ensures the limit follows as we take this part to zero. It is important that $\eta_n$ is constant order with respect to $n$ so that the limiting dynamics are non-trivial (otherwise, $\delta_n \eta_n$ will scale faster than $\delta_n$).
>
> 4. One can study a diffusive limit for the Single Index model at appropriate fixed points in a manner similar to that of Tensor PCA. This part is consistent with [1], where the diffusive limits are studied for a handful of different problems. We originally did not include any diffusive limits for Single Index Models as we felt that the main insights and comparisons between SGD-U and SGD-M were already captured by their ballistic dynamics. For completeness, we included a diffusive example for the single index model (see Section 3.2.2) simply to demonstrate that our extended framework still allows one to study such diffusive limits.
>
> 5. Thank you for catching all these typos. We have fixed the errors you pointed out as well as additional typos we came across in this revision. We want to clarify again that none of these typos affect our results or conclusions.
>
> [1] Gerard Ben Arous, Reza Gheissari, and Aukosh Jagannath. High-dimensional limit theorems for SGD: Effective dynamics and critical scaling. Communications on Pure and Applied Mathematics, 77(3):2030–2080, 2024

---

> > ### Comment · Reviewer_L1BR · 2025-11-20
> >
> > I appreciate the revisions made by the authors. The updates clearly highlight the theoretical contributions and substantially improve the readability of the paper.
> >
> > In the newly added Section 1.1 on contributions, it may help readers if the authors include explicit references to the corresponding theorems or propositions when describing each contribution.
> >
> > Overall, the manuscript has improved significantly with these changes. I will increase my evaluation score to 6.

---

### Official Review · Reviewer_17tv · 2025-11-04

**Soundness:** 3
**Presentation:** 3
**Contribution:** 3
**Rating:** 6
**Confidence:** 3

**Summary:**

The paper extends the main ideas of high-dimensional scaling limit for Stochastic Gradient Descent presented in Ben Arous et al. (2024) to two important variants: SGD with momentum and SGD with adaptive step-sizes. The paper provides a solid theory, and extensive experiments verify the results and justify the need for the exploration of these ideas.

**Strengths:**

The paper is well written and well-organized with clear contributions. I enjoy reading the motivation of the main ideas, and in particular, i find the writing of section 2 on the main result of the paper very informative. Theorem 3.2 is the main theoretical contribution of the work, with the examples in Section 3 to justify the need for the theoretical result.

The proofs for the major steps I checked seem correct, and the theoretical statements are reasonable.

**Weaknesses:**

I do not have to point out a specific weakness. I like this work and I believe it is worth being accepted to ICLR.

Some questions:
Typically, the HIGH-DIMENSIONAL LIMIT THEOREMS of an algorithm are purely theoretical results. In this work, the authors show that SGD-M will amplify high-dimensional effects, potentially degrading performance relative to online SGD. Can they comment on what else can be an interesting follow-up to these ideas? What other algorithms might have a similar outcome?

I believe having such remarks and paragraphs at the end of the paper would be particularly valuable, in a section, let's say, named Conclusion, “Remarks and Future impact”.

**Questions:**

See the Weaknesses part.

---

> ### Author Response · Authors · 2025-11-19
>
> To Reviewer 17tv,
>
> We thank you for your time spent reviewing and for your remarks on our work. Regarding your first question: While SGD-M nominally amplifies high-dimensional effects, the equivalence between online SGD and SGD-M (achieved by appropriate stepsize adjustment) suggests that there are no new phenomena in this regime. On the other hand, we show that one can indeed improve upon online SGD by making use of adaptive stepsizes (SGD-U). We illustrate, for example, how using SGD-U can actually reduce the influence of the same high-dimensional effects. See Sections 3.1 and 3.2.
>
> In line with your suggestion, we believe that it would be very interesting to study further adaptive methods using these techniques. Ultimately we hope this will improve our understanding of pre-conditioning and inform use and design of optimizers. We have included a new "Concluding Remarks" section per your suggestion, where we highlight this point.

---

### Author Response · Authors · 2025-11-19

To the Reviewers,

Thank you for your time in reviewing our submission. We have resubmitted a revision of our paper with all of the requested changes. For your convenience, we have explicitly made all of our revisions in blue.

We address each of the reviewers' concerns below.

---

### Meta-Review · Area_Chair_kN6S · 2026-01-21

**Summary:**

This paper develops novel scaling limits for SGD with momentum and normalized SGD, explaining when these methods behave like standard SGD and when normalization can improve stability. Initial concerns about clarity and novelty were resolved by the rebuttal, and the reviwers agree that the contribution is valuable.

**Reviewer Concerns:**

All issues were addressed: novelty, clarity of exposition, relation to prior work

**Reviewer Scores:**

NA

---

### Decision · Program_Chairs · 2026-01-26

Accept (Poster)